# On the Inherent Privacy Properties of Discrete Denoising Diffusion Models

**Rongzhe Wei**  *rongzhe.wei@gatech.edu*
*ECE, Georgia Institute of Technology*

**Eleonora Kreačić**  *eleonora.kreacic@jpmchase.com*
*J.P. Morgan AI Research*

**Haoyu Wang**  *haoyu.wang@gatech.edu*
*ECE, Georgia Institute of Technology*

**Haoteng Yin**  *yinht@purdue.edu*
*CS, Purude University*

**Eli Chien**  *ichien6@gatech.edu*
*ECE, Georgia Institute of Technology*

**Vamsi K. Potluru**  *vamsi.k.potluru@jpmchase.com*
*J.P. Morgan AI Research*

**Pan Li**  *panli@gatech.edu*
*ECE, Georgia Institute of Technology*

**Reviewed on OpenReview:** *https://openreview.net/forum?id=UuU6C6CUoF*

## Abstract

Privacy concerns have led to a surge in the creation of synthetic datasets, with diffusion models emerging as a promising avenue. Although prior studies have performed empirical evaluations on these models, there has been a gap in providing a mathematical characterization of their privacy-preserving capabilities. To address this, we present the pioneering theoretical exploration of the privacy preservation inherent in *discrete diffusion models* (DDMs) for discrete dataset generation. Focusing on per-instance differential privacy (pDP), our framework elucidates the potential privacy leakage for each data point in a given training dataset, offering insights into how the privacy loss of each point correlates with the dataset's distribution. Our bounds also show that training with $s$-sized data points leads to a surge in privacy leakage from $(\epsilon, \mathcal{O}(\frac{1}{s^2\epsilon}))$-pDP to $(\epsilon, \mathcal{O}(\frac{1}{s\epsilon}))$-pDP of the DDM during the transition from the pure noise to the synthetic clean data phase, and a faster decay in diffusion coefficients amplifies the privacy guarantee. Finally, we empirically verify our theoretical findings on both synthetic and real-world datasets.

## 1 Introduction

Discrete tabular or graph datasets with categorical attributes are prevalent in many privacy-sensitive domains (Vatsalan et al., 2013; Pourhabibi et al., 2020; Li et al., 2021; Shwartz-Ziv & Armon, 2022; Borisov et al., 2022), including finance (Clements et al., 2020; Wang et al., 2021; Potluru et al., 2024), e-commerce (Ahmed et al., 2017; Zhang et al., 2019), and medicine (Duvenaud et al., 2015; Schork, 2015; Ulmer et al., 2020). For instance, medical researchers often collect patient data, such as race, gender, and medical conditions, in a discrete tabular form. However, using and sharing data in these domains carry the risk of revealing personal information (Abay et al., 2019). Studies have shown that it is possible to re-identify individuals

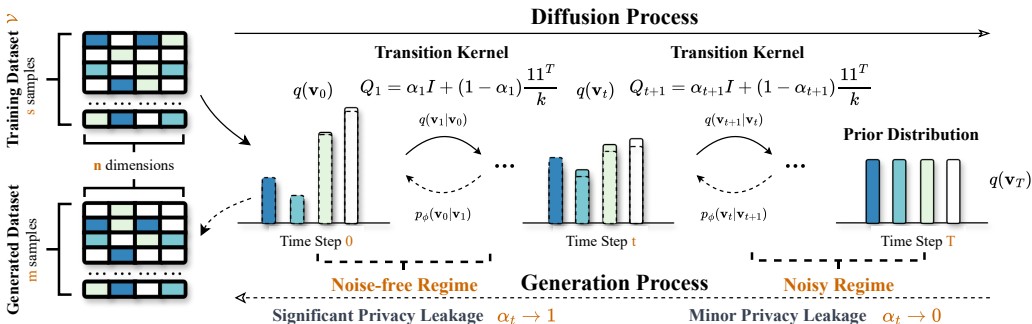

Figure 1: **An Illustration of Discrete Diffusion Models (DDMs).**

in supposedly de-identified healthcare data (McGuire & Gibbs, 2006; El Emam et al., 2011). To address these types of concerns, publishing synthetic datasets with privacy guarantees has been proposed as a way to protect sensitive information and to reduce the risk of privacy leakage (Choi et al., 2017; Patel et al., 2018; Tucker et al., 2020; DuMont Schütte et al., 2021).

Previous research has explored discrete synthetic database releasing methods (Zhou et al., 2009b; Blum et al., 2013; Li et al., 2023). Many of these methods employ data anonymization techniques (Sweeney, 2002; Li et al., 2006; Liu & Terzi, 2008; Lu et al., 2012) or focus on private statistics/statistical models (Sala et al., 2011; Jorgensen et al., 2016; Balog et al., 2018; Harder et al., 2021). In the former category, k-anonymization (Sweeney, 2002) directly works on anonymizing categorical features but it can be vulnerable to the attackers with background knowledge (Machanavajjhala et al., 2007). Alternatively, methods using private statistics or models concentrate on sharing specific private statistics (Harder et al., 2021) or privatizing model parameters (Hardt et al., 2012; Zhang et al., 2017). However, these techniques can sometimes misrepresent the original distribution or reduce sample quality by adding noise directly to model parameters.

Neural network (NN)-based generative models have been leveraged in various domains on account of their ability in learning underlying distributions (Austin et al., 2021). Recently, discrete diffusion models (DDMs) (Hoogeboom et al., 2021; Austin et al., 2021; Campbell et al., 2022; Gu et al., 2022; Vignac et al., 2022), as a typical representative of diffusion models (DMs), have emerged as a powerful class of generative models for discrete data and demonstrate great potential to generate samples with striking performance (Haefeli et al., 2022; Zheng et al., 2023). DDMs are latent variable generative models that employ both a forward and reverse Markov process (See Fig. 1). In the forward diffusion process, each discrete sample is gradually corrupted with dimension-wise independent noise. This is often implemented through the use of progressive transition kernels, which yields not only high fidelity-diversity trade-offs but also robust training objectives (Dhariwal & Nichol, 2021). On the other hand, the reverse process learns denoising neural networks that aim to predict the noise and reconstruct the original sample. Despite the impressive performance of DDMs, it is still unclear whether DDMs trained on sensitive datasets can be safely used to generate synthetic samples.

Efforts have empirically examined the privacy implications of DMs. While previous literature suggests that DMs generate synthetic training data to address privacy concerns (Jahanian et al., 2021; Carr, 2022), recent studies have shown that DMs may not be suitable for releasing private synthetic data. Specifically, Wu et al. (2022); Hu & Pang (2023) conduct membership inference attacks on DMs for text-to-image tasks and demonstrate that membership inference poses a severe threat in diffusion-based generation. Besides, studies show that DMs can memorize training samples (Somepalli et al., 2022; Carlini et al., 2023). Although there exist practical observations for privacy properties of DMs, there is limited research aimed at mathematically characterizing the privacy guarantees of data generated by DMs. Moreover, understanding privacy guarantees may guide practitioners to determine whether additional mechanisms, such as DP-SGD (Abadi et al., 2016), PATE (Papernot et al., 2016), should be incorporated to meet practical privacy requirements.

Differential privacy (DP) (Dwork et al., 2006; 2014), the most commonly used *algorithm-centric* framework to characterize the privacy guarantee of an algorithm, is derived from the worst-case dataset. However, in the

context of synthetic data sharing, the characterization of privacy leakage is about the synthetic dataset (the algorithm output) rather than the generative model (the algorithm itself), and the learned data distribution to generate synthetic data strongly depends on the empirical distribution of the data points used for training. Therefore, a privacy guarantee that may incorporate *the distributional characteristics of data points* in the given training dataset may offer a far more accurate privacy characterization than the worst-case analysis. Such data-dependent analysis may help practitioners learn which data points in the training dataset tend to introduce privacy leakage concerns in the generation process and thus design the relevant protection strategy.

In this paper, we take the first step to analyze the privacy guarantees of DDMs for a fixed training dataset. Specifically, we leverage the data-dependent privacy framework termed per-instance differential privacy (pDP), which is defined upon an instance in a fixed training dataset as outlined by (Wang, 2019). The analysis of pDP allows for a fine-grained characterization of the potential privacy leakage of each data point in the training set. This offers data curators a better understanding of the sensitivity of training data.

Our analysis considers a DDM trained on $s$ samples and generates $m$ samples, and we keep track of the privacy leakage in each generation step. We prove that as the data generation step transits from $t = T$ (noisy regime) to $t = 0$ (noise-free regime), the privacy leakage increases from $(\epsilon, \mathcal{O}(\frac{m}{s^2 \epsilon (1-e^{-\epsilon})}))$-pDP to $(\epsilon, \mathcal{O}(\frac{m}{s \epsilon (1-e^{-\epsilon})}))$-pDP where the data-dependent term is hidden in the big-$\mathcal{O}$ notation. Consequently, the final few generation steps ($\alpha_t \to 1$ in Fig. 1) dominate the main privacy leakage in DDMs. Further, our analysis demonstrates that the privacy bound $\mathcal{O}(1/s)$ is tight when $m = 1$, emphasizing the inherent weak privacy guarantee of DDMs. Moreover, faster decay in diffusion coefficients yields better privacy preservation. Both synthetic and real dataset evaluations validate our theoretical findings.

For the data-dependent part, we develop a practical algorithm to estimate the privacy leakage of each data point in real-world datasets according to our pDP bounds. We evaluate the data-dependent part by removing the most sensitive data points (according to our data-dependent privacy parameters) from the dataset to train a DDM, and then evaluating the ML models trained based on the synthetic dataset generated by the DDM. Interestingly, we observe that the ML models obtained after a part of data removal can even outperform others without such data removal. We attribute this to the fact that the removed data points are likely outliers which may be actually not good for ML models to learn from. This illustrates another potentially valuable usage of our data-dependent analysis.

To avoid any confusion, we provide several important explanations for considering pDP in our work. pDP, tailored to the training set, offers data curators a more accurate and fine-grained estimation of the potential privacy leakage of each data point, compared to DP which studies the worst case and keeps agnostic to the dataset (Wang, 2019). However, it is crucial to understand that pDP is not a replacement for DP. Direct application of data-dependent sensitivity for noise addition is not permissible for ensuring privacy, as the added noise may leak private information due to its data dependency. Data-dependent methods such as smooth sensitivity (Nissim et al., 2007) and propose-test-release (Dwork & Lei, 2009) may be employed, while they are beyond the scope of this paper. Our analysis is to provide insights into the inherent privacy afforded by DDMs, and to guide data curators in assessing the privacy risks associated with different parts of the dataset. We are not to develop an algorithm to match a certain privacy budget as the goal. Given this purpose, pDP is a more suitable metric than DP. In practice, the pDP assessment is expected to be kept confidential and used by the data curators to understand the dataset and evaluate the potential privacy leakage if one uses DDMs to generate synthetic datasets.

## 1.1 More Related Work

A significant amount of research has been conducted on the subject of publishing privacy sensitive data (Ji et al., 2014; Baraheem & Yao, 2022). As of now, traditional non-deep learning techniques for preserving privacy while generating discrete data can be broadly classified into two categories: **(a) Data anonymization-based approaches.** These methods employ a variety of techniques to directly sanitize data to prevent easy re-identification (Abay et al., 2019). One most popular framework is termed k-anonymity (Sweeney, 2002) that requires each record is indistinguishable from at least $k - 1$ other records with respect to certain identifying attributes. Several extensions of this framework have been proposed in (LeFevre et al., 2005; Aggarwal et al., 2005; Machanavajjhala et al., 2007; Li et al., 2006; Truta & Vinay, 2006; Machanavajjhala et al., 2008;

Liu & Terzi, 2008; Liang & Samavi, 2020). However, these methods are typically prone to various privacy attacks (Machanavajjhala et al., 2007). **(b) Methods based on statistical models or private statistics.** Barak et al. (2007) employed Fourier decomposition and prior knowledge to release low-dimensional data projections. Zhou et al. (2009b) proposed a database compression procedure based on low-rank random affine transformations and publish low-dimensional data. Other works along this line include (Liu et al., 2005; Zhou et al., 2009a; Ding et al., 2011; Cormode et al., 2011; Kenthapadi et al., 2012; Cormode et al., 2012). Note that these works can work for both discrete and continuous data. Furthermore, Balog et al. (2018) introduced a framework employing kernel mean embeddings (Smola et al., 2007) in Reproducing Kernel Hilbert Space, and ensuring privacy by using synthetic data approximations to enable safe data release. Nevertheless, these methods usually suffer from poorly generated sample qualities. With regard to this, establishing NN-based private models is a promising way to enhance sample qualities due to the great expressive power of deep networks.

Hitherto, there are studies on NN-based private models but few analyze the inherent privacy of the model itself. In (Lin et al., 2021), it was shown that a vanilla GAN trained on $s$ samples inherently satisfies a weak $(\epsilon, \mathcal{O}(\frac{m}{s\epsilon}))$-DP guarantee when releasing $m$ samples. In this work, our results demonstrate that DDMs provide weak privacy guarantees in the same order as GANs. But note that Lin et al. (2021) did not provide a data-dependent bound. Their bounds are in the order form and cannot be explicitly computed from data curator's side for a given training dataset. Because of such weak inherent privacy there were efforts to bring additional privacy techniques into the model, such as DP-SGD (Abadi et al., 2016). Xie et al. (2018) proposed DPGAN that integrates modified DP-SGD in WGAN to ensure privacy for GAN-generated samples. Dockhorn et al. (2022) applied DP-SGD to privatize model parameters in continuous DMs for image data without analyzing the inherent privacy of DMs. Recently, Ghalebikesabi et al. (2023) have showed that fine-tuning a pre-trained diffusion model with DP-SGD can generate verifiable private synthetic data for the dataset used for fine-tuning.

## 2 Preliminaries

We start by introducing notations and concepts for analysis. Let $[n] = \{1, 2, ..., n\}$ and $\mathcal{X}^n$ represent an $n$-dimensional discrete space with each dimension having $k$ categories, i.e. $\mathcal{X}^n := \mathcal{X}_1 \times \cdots \times \mathcal{X}_n$ with $\mathcal{X}_i = [k], i \in [n]$. We assume that training datasets $\mathcal{V}$ reside in $\mathcal{X}^n$, implying samples are vector-valued data of $n$ entries, each from one of the $k$ categories. Although we assume consistent categories across columns, our analysis can account for datasets with varied category counts using the maximum category count.

**Per-instance Differential Privacy.** DP (Dwork et al., 2006; 2014) is a de-facto standard to quantify privacy leakage. We adapt DP definition for specific adjacent datasets, introducing per-instance DP:

**Definition 1** (($\epsilon, \delta$)-Per-instance Differential Privacy (pDP) (Wang, 2019)). Let $\mathcal{V}_0$ be a training dataset, $\mathbf{v}^* \in \mathcal{V}_0$ be a fixed point and $\mathcal{M}$ be a randomized mechanism. Define adjacent dataset $\mathcal{V}_1 = \mathcal{V}_0 \backslash \{\mathbf{v}^*\}$. We say $\mathcal{M}$ satisfies ($\epsilon, \delta$)-pDP with respect to ($\mathcal{V}_0, \mathbf{v}^*$) if for all measurable set $\mathcal{O} \subset range(\mathcal{M})$, $\{i, j\} = \{0, 1\}$:

$$\mathcal{P}(\mathcal{M}(\mathcal{V}_i) \in \mathcal{O}) \leq e^\epsilon \mathcal{P}(\mathcal{M}(\mathcal{V}_j) \in \mathcal{O}) + \delta. \tag{1}$$

It is important to highlight that pDP is uniquely defined for a specific dataset-data point pair. This capability is crucial for understanding the privacy leakage of the given dataset, as elaborated in Sec. 4. Additionally, by taking the supremum over all conceivable datasets $\mathcal{V}_0$ and points $\mathbf{v}^*$, we can obtain DP from pDP when considering model releasing scenario (Theorem E.1). A more comprehensive discussion of the DP guarantees associated with DDMs is provided in Appendix. E.

**Discrete Diffusion Models.** DDMs (Hoogeboom et al., 2021; Austin et al., 2021; Vignac et al., 2022; Haefeli et al., 2022) are diffusion models that can generate categorical data. Let $\mathbf{v}_t$ denote the data random variable at time t. The forward process involves gradually corrupting data with the noising Markov chain $q$, according to $q(\mathbf{v}_{1:T}|\mathbf{v}_0) = \prod_{t=1}^{T} q(\mathbf{v}_t|\mathbf{v}_{t-1})$, where $\mathbf{v}_{1:T} = \mathbf{v}_1, \mathbf{v}_2, ..., \mathbf{v}_T$. On the other hand, the reverse process, $p_\phi(\mathbf{v}_{0:T}) = p(\mathbf{v}_T) \prod_{t=1}^{T} p_\phi(\mathbf{v}_{t-1}|\mathbf{v}_t)$, gradually reconstructs the datasets starting from a prior $p(\mathbf{v}_T)$. The denoising neural network (NN) learns $p_\phi(\mathbf{v}_{t-1}|\mathbf{v}_t)$ by optimizing the ELBO, which comprises three loss terms: the reconstruction term ($L_r$), the prior term ($L_p$), and the denoising term ($L_t$), represented in the

following equation (Ho et al., 2020):

$$\underbrace{\mathbb{E}_{q(\mathbf{v}_1|\mathbf{v}_0)}[\log p_\phi(\mathbf{v}_0|\mathbf{v}_1)]}_{\text{Reconstruction Term } L_r} - \underbrace{D_{\text{KL}}(q(\mathbf{v}_T|\mathbf{v}_0)\|p_\phi(\mathbf{v}_T))}_{\text{Prior Term } L_p} - \sum_{t=2}^{T}\underbrace{\mathbb{E}_{q(\mathbf{v}_t|\mathbf{v}_0)}[D_{\text{KL}}(q(\mathbf{v}_{t-1}|\mathbf{v}_t,\mathbf{v}_0)\|p_\phi(\mathbf{v}_{t-1}|\mathbf{v}_t))]}_{\text{Denoising Term } L_t}. \quad (2)$$

Specifically, the **forward process** can be described by a series of transition kernels $\{Q_t^i\}_{t\in[T],i\in[n]}$ where for any entry $\mathbf{v}^i$, $[Q_t^i]_{lh} = q(\mathbf{v}_t^i = h|\mathbf{v}_{t-1}^i = l)$ represent the probability of a jump from category $l$ to $h$ on the $i$-th entry at time $t$. Since for each entry $i$ the number of categories is the same, we can rely on the same transition kernels for all dimensions and use $Q_t$ instead of $Q_t^i$. Let $\overline{Q}_t = Q_1 Q_2 ... Q_t$ denote the accumulative transition matrix from time 1 to time t. We use a uniform prior distribution $p(\mathbf{v}_T)$. The corresponding doubly stochastic matrices is determined by a series of important parameters termed **diffusion coefficients** $(\{\alpha_t, t \in [T]|\alpha_t \in (0,1)\})$ which control the transition rate from original distribution to uniform measure. Specifically, define $Q_t = \alpha_t I + (1 - \alpha_t)\frac{\mathbb{1}\mathbb{1}^T}{k}$ and then $\bar{Q}_t = \bar{\alpha}_t I + (1 - \bar{\alpha}_t)\frac{\mathbb{1}\mathbb{1}^T}{k}$ where $\bar{\alpha}_t = \prod_{i=1}^{t}\alpha_t$. In the **reverse process**, denoising networks are leveraged to predict $p_\phi(\mathbf{v}_{t-1}|\mathbf{v}_t)$ in hope of approximating $q(\mathbf{v}_{t-1}|\mathbf{v}_t,\mathbf{v}_0)$. In practice, instead of directly predicting $p_\phi(\mathbf{v}_{t-1}|\mathbf{v}_t)$, denoising networks are learned to predict a clean data $\mathbf{v}_0$ at time 0 with a noisy $\mathbf{v}_t$ as input, i.e. $p_\phi(\mathbf{v}_0|\mathbf{v}_t)$. To train the denoising network, one needs to sample noisy points from $q(\mathbf{v}_t|\mathbf{v}_0)$, and feed them into the denoising network $\phi_t$ and obtain $p_\phi(\mathbf{v}_0|\mathbf{v}_t)$. Specifically, we adopt

$$L_{\text{train}} = D_{\text{KL}}(q(\mathbf{v}_0|\mathbf{v}_t)\|p_\phi(\mathbf{v}_0|\mathbf{v}_t)) = \frac{1}{|\mathcal{V}|}\sum_{\mathbf{v}_0\in\mathcal{V}}\mathbb{E}_{\mathbf{v}_t\sim q(\mathbf{v}_t|\mathbf{v}_0)}\left[\sum_{i=1}^{n}L_{\text{CE}}(\mathbf{v}_0^i, p_\phi(\mathbf{v}_0^i|\mathbf{v}_t))\right] \quad (3)$$

This loss serves as the basis for our later sufficient training Assumption 1. In the generation process, we need to bridge the connection of $p_\phi(\mathbf{v}_{t-1}|\mathbf{v}_t)$ and $p_\phi(\mathbf{v}_0|\mathbf{v}_t)$, which in practice depends on a dimension-wise conditional independence condition (Vignac et al., 2022):

$$p_\phi(\mathbf{v}_{t-1}|\mathbf{v}_t) = \prod_{i\in[n]} p_\phi(\mathbf{v}_{t-1}^i|\mathbf{v}_t) = \prod_{i\in[n]}\sum_{l\in\mathcal{X}_i} q(\mathbf{v}_{t-1}^i|\mathbf{v}_t, \mathbf{v}_0^i = l)p_\phi(\mathbf{v}_0^i = l|\mathbf{v}_t). \quad (4)$$

**Other Notations.** Given two samples $\mathbf{v}$ and $\tilde{\mathbf{v}}$, let $\bar{\omega}(\mathbf{v}, \tilde{\mathbf{v}})$ represent the count of differing entries, i.e., $\bar{\omega}(\mathbf{v}, \tilde{\mathbf{v}}) = \#\{i|\mathbf{v}^i \neq \tilde{\mathbf{v}}^i, i \in [n]\}$. For $\eta \in [n]$ and $\mathbf{v} \in \mathcal{V}_1$, define $N_\eta(\mathbf{v}) = |\{\mathbf{v}' \in \mathcal{V}_1 : \bar{\omega}(\mathbf{v}, \mathbf{v}') \leq \eta\}|$ and $\mathcal{V}_1^{i|l} = \{\mathbf{v} \in \mathcal{V}_1|\mathbf{v}^i = l\}$ the set of data points with a fixed-valued entry. We use $\mathcal{D}_{\text{KL}}(\cdot\|\cdot)$ and $\|\cdot\|_{TV}$ for KL-divergence and total variation. Let $\mu_t^+ = \frac{1+(k-1)\alpha_t}{k}$ and $\mu_t^- = \frac{1-\alpha_t}{k}$ represent one-step transition probabilities to the same and different states respectively at time $t$ while $\bar{\mu}_t^+ = \frac{1+(k-1)\bar{\alpha}_t}{k}$ and $\bar{\mu}_t^- = \frac{1-\bar{\alpha}_t}{k}$ are the accumulated transition probabilities. Transition probability ratios are defined as $R_t = \frac{\mu_t^+}{\mu_t^-}$ and $\bar{R}_t = \frac{\bar{\mu}_t^+}{\bar{\mu}_t^-}$. A larger ratio indicates a higher likelihood of maintaining the same feature category in the diffusion process. Moreover, define $(\cdot)_+ = \max\{\cdot, 0\}$.

## 3 Main Results

### 3.1 Inherent Privacy Guarantees of DDMs

First, we define the mechanism under analysis. Let $\mathcal{M}_t(\mathcal{V}; m)$ represent the mechanism where, for an input dataset $\mathcal{V}$, it outputs $m$ samples generated at time $t$ using the DDM's generation process. Specifically, $\mathcal{M}_0(\mathcal{V}; m)$ signifies the final generated dataset by DDM. In the paper, we focus on the behavior of $\mathcal{M}_t$ in the generation process. Below, we outline the assumptions:

**Assumption 1** (Sufficient training of $\phi$). Given dataset $\mathcal{V}$, let $\mathbf{v}_0$ denote the predicted random variables at time 0. Let $\phi$ denote denoising NNs trained on dataset $\mathcal{V}$. We say Assumption 1 is satisfied if there exist small constants $\gamma_t > 0$ such that $\forall \mathbf{v}_t \in \mathcal{X}^n$:

$$\mathcal{D}_{\text{KL}}(q(\mathbf{v}_0^i|\mathbf{v}_t)\|p_\phi(\mathbf{v}_0^i|\mathbf{v}_t)) \leq \gamma_t, \forall i \in [n], \forall t \in [T]. \quad (5)$$

**Assumption 2 (Gap between Forward and Backward Diffusion Paths).** Given dataset $\mathcal{V}$, let $\mathbf{v}_t$ denote the random variable sampled from intermediate distributions at time t in both the forward process (following $q(\mathbf{v}_t)$) and backward process (following $p_\phi(\mathbf{v}_t)$). We say the Assumption 2 is satisfied if there exists small positive constant $\tilde{\gamma}_t \ll 2$ such that

$$\|q(\mathbf{v}_t) - p_\phi(\mathbf{v}_t)\|_{\mathrm{TV}} \le \tilde{\gamma}_t, \forall t \in [T]. \tag{6}$$

Assumption 1 states that denoising networks, when trained using the loss function in Eq. (3), can effectively infer clean data from intermediate noisy data distributions. Given a sufficiently expressive model, we expect $\gamma_t$ to be small. Assumption 2 asserts that diffusion and generation paths are close, which is a reasonable assumption due to the recent analysis (Campbell et al., 2022). However, one cannot use Eq. (6) to derive privacy bound directly as closeness in total variation does not imply DP in general though the reverse could be true (Bassily et al., 2016).

With above assumptions, we investigate the flow of privacy leakage along generation process. Our analysis centers around the inherent privacy guarantees of DDM-generated samples at specific release step, denoted as $T_{\mathrm{rl}}$. Later, we will show that our privacy bound is tight when generating a single sample ($m = 1$).

**Theorem 1 (Inherent pDP Guarantees for DDMs).** *Given a dataset $\mathcal{V}_0$ with size $|\mathcal{V}_0| = s + 1$ and a data point $\mathbf{v}^* \in \mathcal{V}_0$ to be protected, denote $\mathcal{V}_1$ such that $\mathcal{V}_1 = \mathcal{V}_0 \backslash \{\mathbf{v}^*\}$. Assume the denoising networks trained on $\mathcal{V}_0$ and $\mathcal{V}_1$ satisfy Assumption 1 and Assumption 2. Given a specific time step $T_{rl}$, the mechanism $\mathcal{M}_{T_{rl}}(\cdot; m)$ satisfies $(\epsilon, \delta)$-pDP with respect to $(\mathcal{V}_0, \mathbf{v}^*)$ such that given $\epsilon$,*

$$\delta(\mathcal{V}_0, \mathbf{v}^*) \le m \left[ \underbrace{\sum_{t=T_{rl}}^{T} \min\left\{ \frac{4N_{(1+c_t^*)\eta_t}(\mathbf{v}^*)}{s}, 1 \right\} \cdot \frac{n}{s^{\psi_t}} + \frac{n(1 - \frac{1}{\bar{R}_{t-1}})}{s^2}}_{\text{Main Privacy Term}} + \underbrace{\mathcal{O}\left( \sqrt{\gamma_t} + \tilde{\gamma}_t \right)}_{\text{Error Term}} \right] / (\epsilon(1 - e^{-\epsilon})). \tag{7}$$

*where $\psi_t, \eta_t, c_t^*$ are **data-dependent quantities** determined by $\mathbf{v}^*$ and $\mathcal{V}_1$. Define a similarity measure $Sim(\mathbf{v}^*, \mathcal{V}) = \sum_{\mathbf{v} \in \mathcal{V}} \bar{R}_t^{-\bar{\omega}(\mathbf{v}, \mathbf{v}^*)}$. Then, $\psi_t, \eta_t, c_t^*$ follow*

$$\frac{n}{s^{\psi_t}} = \frac{(\overline{\alpha}_{t-1} - \overline{\alpha}_t)/(k\bar{\mu}_t^+ \bar{\mu}_t^-)}{1 + Sim(\mathbf{v}^*, \mathcal{V}_1)} \cdot \sum_{i=1}^{n} \log\left( 1 + \frac{\bar{R}_{t-1}^2 - 1}{\bar{R}_{t-1}^2 Sim(\mathbf{v}^*, \mathcal{V}_1^{i|\mathbf{v}^{*i}}) + Sim(\mathbf{v}^*, \mathcal{V}_1) + 1} \right). \tag{8}$$

*And, $\eta_t, c_t^*$ are the smallest $\eta_t \in \{1, 2, ..., n\}$, $c_t^* \in \{0, \frac{1}{\eta_t}, \frac{2}{\eta_t}, ..., \frac{n-\eta_t}{\eta_t}\}$ which satisfy*

$$\eta_t \ge \frac{\log \vartheta(\eta_t)}{\log \frac{1}{n(1-\bar{\mu}_t^+)}} + \left( \frac{\log\left( \vartheta(\eta_t) \frac{\overline{\alpha}_{t-1} - \overline{\alpha}_t}{k\bar{\mu}_t^+ \bar{\mu}_t^-} \cdot s^{\psi_t} \right)}{2 \log \bar{R}_t} - 2 \right)_+, \quad c_t^* \ge \frac{\frac{1}{\eta_t} \log \vartheta((1 + c_t^*)\eta_t) + \frac{3}{2}}{\log \frac{1}{\bar{\mu}_t^-} - 1}. \tag{9}$$

*where $\vartheta(\eta) = (s - N_\eta(\mathbf{v}^*))/N_\eta(\mathbf{v}^*)$ that represents the ratio between the numbers of points outside the $\eta$-ball and inside it.*

Theorem 1 quantifies the privacy leakage of a specific point $\mathbf{v}^*$ in training set $\mathcal{V}_0$. The privacy bound comprises a main privacy term that represents the inherent pDP guarantees for DDMs, highlighting the data-dependent nature of our bound, and an error term stemming from denoising network training and path discrepancies. Those data-dependent quantities are complex to maintain a tight measurement for a dataset-data point pair. Next, we will further explain these quantities.

First, as the generation process forms a Markov chain where the transition probability $p_\phi(\mathbf{v}^{(t-1)}|\mathbf{v}^{(t)})$ is learned from training, each generation step will leak some information from the training dataset. It can be shown that the majority of such leakage, represented in the pDP bound (in the appendix) follows

$$\mathbb{E}_{\mathbf{v} \sim p_\phi(\mathbf{v}_{t|0} = \mathbf{v})} d^{(t)}(\mathbf{v}) \tag{10}$$

where let $\mathbf{v}_{t|\lambda}$ represents the random variable of the generated data at time $t$ of the generation process when the diffusion model gets trained over the dataset $\mathcal{V}_\lambda$, $\lambda \in \{0, 1\}$ and $d^{(t)}(\mathbf{v}) = \sum_{\lambda \in \{0,1\}} \mathcal{D}_{\mathrm{KL}}(p_\phi(\mathbf{v}_{t-1|\lambda}|\mathbf{v}_{t|\lambda} =$

$\mathbf{v})\|p_\phi(\mathbf{v}_{t-1|\bar{\lambda}}|\mathbf{v}_{t|\bar{\lambda}} = \mathbf{v}))$ with $\bar{\lambda} = 1 - \lambda$, which characterizes a symmetric distance between two conditional distributions characterized by the learned diffusion model. Essentially, the three data-dependent quantities $\psi_t, \eta_t, c_t^*$ are to bound Eq. (10).

**Quantity** $\psi_t$: As shown in Fig. 2, $\frac{n}{s^{\psi_t}}$ quantifies $\max_\mathbf{v} d^{(t)}(\mathbf{v})$ where the maximum is achieved at the removed point $\mathbf{v} = \mathbf{v}^*$ (green in Fig. 2 ). A closer inspection reveals that $\psi_t$ depends on the terms $\mathrm{Sim}(\mathbf{v}^*, \mathcal{V}_1)$ and $\mathrm{Sim}(\mathbf{v}^*, \mathcal{V}_1^{i|\mathbf{v}^{*i}})$. By the definition of $\bar{\omega}$, these terms assess how $\mathbf{v}^*$ aligns with the remaining points in $\mathcal{V}_1$.

**Evolution of** $\psi_t$. During the generation phase, as $t$ progresses from $T$ to 1, the values of $\frac{1}{s^{\psi_t}}$ increase from $\mathcal{O}_s(\frac{1}{s^2})$ to $\mathcal{O}_s(1)$. This implies that the potential privacy risk escalates as the data generation process evolves from a noisy regime to a noise-free regime.

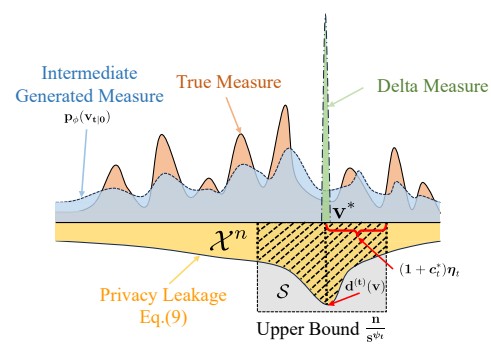

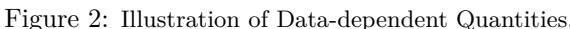

Figure 2: Illustration of Data-dependent Quantities.

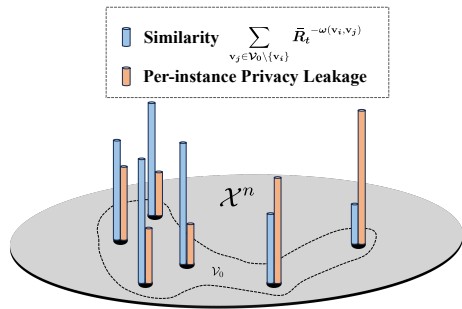

Figure 3: Illustration of the correlation between dataset similarity $(\mathrm{Sim}(\mathbf{v}_i, \mathcal{V}_0\backslash\{\mathbf{v}_i\}), \forall \mathbf{v}_i \in \mathcal{V}_0)$ and pDP Leakage.

**Quantities** $\eta_t$ **and** $c_t^*$: It is evident that the intermediate generated measure $p_\phi(\mathbf{v}_{t|0})$ (blue in Fig. 2) diverges from the delta measure on the most sensitive point $\delta_{\mathbf{v}=\mathbf{v}^*}$ (green). Therefore, the actual privacy leakage characterized by $d^{(t)}(\mathbf{v})$ (yellow) averaged over the measure $p_\phi(\mathbf{v}_{t|0})$ is much less than its maximum. To provide a tight characterization of such, the two quantities $\eta_t$ and $c_t^*$ are introduced to define a local region $\mathcal{S} = \{\mathbf{v}' \in \mathcal{X}^n : \bar{\omega}(\mathbf{v}, \mathbf{v}') \le (1 + c_t^*)\eta_t\}$ centered on vulnerable point $\mathbf{v}^*$, within which the privacy leakage can be bounded by the sum of (a) $p_\phi(\mathbf{v}_{t|0} \in \mathcal{S}) \max_{\mathbf{v} \in \mathcal{S}} d^{(t)}(\mathbf{v})$ with a small $p_\phi(\mathbf{v}_{t|0} \in \mathcal{S})$ and (b) $p_\phi(\mathbf{v}_{t|0} \notin \mathcal{S}) \max_{\mathbf{v} \notin \mathcal{S}} d^{(t)}(\mathbf{v})$ with a small $\max_{\mathbf{v} \notin \mathcal{S}} d^{(t)}(\mathbf{v})$. $(\eta_t, c_t^*)$ shown in Eq. (9) are chosen to properly balance these two parts. $\eta_t$ and $c_t^*$ always exist: Note that when $\eta_t = n$ or $c_t^* = n/\eta_t - 1$, the right-hand side of either inequality in Equation (9) approaches $-\infty$ ($\log 0$). In fact, both of the RHS's of the two inequalities decrease w.r.t. $\eta_t$ and $c_t^*$. So, in practice, the smallest $\eta_t$ and $c_t^*$ can be found via binary search given the dataset $\mathcal{V}_0$ and $\mathbf{v}^*$.

**Evolution of** $(1 + c_t^*)\eta_t$. For each time step $t$, the smallest value of $(1 + c_t^*)\eta_t$ is chosen as the radius. As $t$ progresses from $T$ to 1, the value of $(1 + c_t^*)\eta_t$ monotonically decreases. When $\bar{\alpha}_t$ approaches 1 for smaller $t$ values, $(1 + c_t^*)\eta_t$ tends to zero, i.e., $\mathcal{S}$ only includes $\mathbf{v}^*$. The reason is that, as smaller $t$ values, different data points are less mixed with others (because of less noise added in the forward process), the privacy leakage of $\mathbf{v}^*$ becomes more concentrated around the changes of the likelihoods of the generated data points that look like $\mathbf{v}^*$, thus calling for a decrease of the radius. To consider the impact on the bound in Eq. (7), the number of data points in this region $N_{(1+c_t^*)\eta_t}(\mathbf{v}^*)$ will decrease from $s$ to 1 as $t$ changes from $T$ to 1.

**Discussion on Theorem 1.** Based on the previous discussion, as $t$ decreases from $T$ to 1, $N_{(1+c^*)\eta_t}(\mathbf{v}^*)/s$ changes from $\mathcal{O}_s(1)$ to $\mathcal{O}_s(1/s)$, and $1/s^{\psi_t}$ changes from $\mathcal{O}_s(\frac{1}{s^2})$ to $\mathcal{O}_s(1)$. Consequently, the privacy leakage for each-step DDM-generated samples gradually increases from $\mathcal{O}_s(\frac{m}{s^2})/[\epsilon(1 - e^{-\epsilon})]$ to $\mathcal{O}_s(\frac{m}{s})/[\epsilon(1 - e^{-\epsilon})]$.

This implies a natural utility-privacy tradeoff for the data generated by DDMs. In practice, to guarantee the data quality, we often release the data in the noise-free side ($t = 0$), where only a weak privacy guarantee of approximately $(\epsilon, \mathcal{O}_s(m/s)/[\epsilon(1 - e^{-\epsilon})])$ can be achieved. To enhance data privacy, we may expect to release the data generated with a larger step $t \ge 1$.

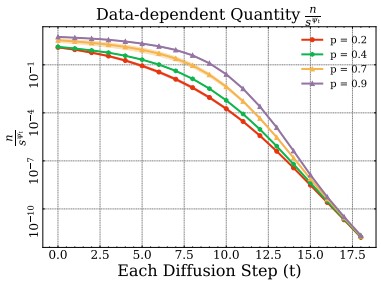 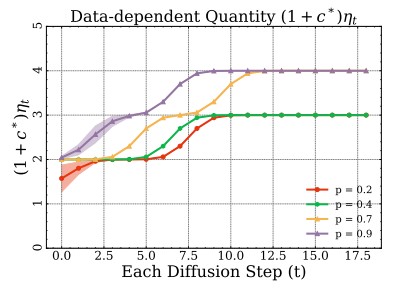 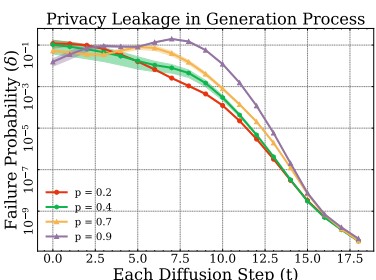

Figure 4: pDP Leakage in Eq. (7): **LEFT:** Characterization of $\frac{n}{s^{\psi_t}}$. **MIDDLE:** Characterization of $(1 + c_t^*)\eta_t$. **RIGHT:** Characterization of Privacy Leakage (Main Privacy Term). **Experimental Setup:** Given specific DDM design $k = 5, n = 5, T = 20, \epsilon = 10$ trained on dataset with $s = 1000$ following the distribution in Sec. 3.3 with parameter $p$. Fix $\mathbf{v}^*$ where each column has a non-majority category. Results are based on 5 times independent tests.

This result also reveals that the inherent privacy guarantees of releasing data generated by DDMs is weak ($\propto \mathcal{O}(m/s)$), in the same order of guarantees for GAN-generated samples (Lin et al., 2021). This characterization also matches many recent empirical studies that have shown concerns on privacy leakage due to publishing data generated by DMs (Hu & Pang, 2023; Carlini et al., 2023; Dockhorn et al., 2022). While privacy budgets for all data points maintain the same order in relation to the sample size, the contacts can differ markedly across data points. Intuitively, a data point $\mathbf{v}^* \in \mathcal{V}_0$ with less similarity with the other data points tends to have higher privacy leakage. This is indicated by Eq. (8), where a smaller similarity $\sum_{\mathbf{v} \in \mathcal{V}_0 \setminus \{\mathbf{v}^*\}} \bar{R}_t^{-\omega(\mathbf{v}^*, \mathbf{v})}$, leads to a larger pDP leakage (as illustrated in Fig. 3).

The upper bound for $t = 1$ with a dependence on the dataset size $\mathcal{O}_s(\frac{1}{s})$ demonstrates a weak privacy guarantee given by DDMs, as this is on par with the privacy implication of the Strawman approach that uniformly at random samples from the original dataset and publishes the samples. However, we emphasize that this is not due to a loose analysis, as the following will provide a lower bound of such privacy leakage due to DDM generation in the same order. Beyond this, our upper bound is valuable as it elucidates that releasing a dataset at an earlier stage (with a larger $t$ step) generated by DDMs could potentially strengthen the privacy guarantee to $\Omega_s(\frac{1}{s^2})$. Moreover, our upper bound specifies the influence of diffusion coefficients and dataset distributions on the privacy bounds, which the Strawman approach by publishing the samples from the original dataset cannot tell. More details will be discussed in Sec. 3.2.

**Tightness of Privacy Bound w.r.t Sample Size.** In Theorem 1, the privacy parameter of $\delta$ scales as $\mathcal{O}_s(\frac{1}{s})$ with sample size. Here we establish a lower bound for $\delta$ with respect to the sample size by evaluating the worst-case scenario and show that $\mathcal{O}(1/s)$ is the optimal bound that DDM can achieve inherently for $m = 1$. For illustrative purposes, consider the case where $n = 2$ with two distinct categories. Define adjacent datasets: $\mathcal{V}_0 = \{\underbrace{[0,0]^T, ...., [0,0]^T}_{s-1}, [1,1]^T, [1,1]^T\}$ and $\mathcal{V}_1 = \mathcal{V}_0 \setminus \{[1,1]^T\}$.

**Theorem 2** (**Lower Bound on Inherent pDP Guarantees for DDMs**)**.** *Assume the denoising networks are perfectly trained. Given a diffusion model architecture design (Sigmoid Schedule $\alpha_t = \frac{Sigmoid(3) - Sigmoid(\frac{3t}{T})}{Sigmoid(3) - 0.5}, T = 10$), there exist an adjacent dataset $\mathcal{V}_0, \mathcal{V}_1 = \mathcal{V}_0 \setminus \{\mathbf{v}^*\}$ with feature dimension $n = 2$ such that the mechanism $\mathcal{M}_0(\cdot; 1)$ does not satisfies $(0.04, \delta)$-pDP with respect to $(\mathcal{V}_0, \mathbf{v}^*)$ for any $\delta < \frac{1}{6s}$.*

Regarding lower bounds on $(\epsilon, \delta)$-pDP under general model configurations, including diffusion schedules and diffusion steps, please refer to Appendix D.

**Discussion on the Inherent DP Guarantees for DDMs.** As mentioned in Sec. 2, DP guarantees can be obtained from taking all conceivable datasets $\mathcal{V}_0$ and points $\mathbf{v}^*$. Therefore, by considering the worst case adjacent dataset pair, we derive the DP privacy bound for DDMs:

**Theorem 3** (**Inherent DP Guarantee for DDMs (Informal)**)**.** *Given any adjacent datasets $\mathcal{V}_0, \mathcal{V}_1$. Assume the denoising networks trained on $\mathcal{V}_0$ and $\mathcal{V}_1$ satisfy Assumption 1 and Assumption 2. Given a specific*

time step $T_{rl}$, the mechanism $\mathcal{M}_{T_{rl}}(\cdot; m)$ satisfies $(\boldsymbol{\epsilon, \delta})$-**differential privacy** such that

$$\delta \leq m \left[ \underbrace{\sum_{t=T_{rl}}^{T} \min\left\{ \frac{4N_{(1+\tilde{c}_t^*)\tilde{\eta}_t}}{s}, 1 \right\} \cdot \frac{n}{s^{\Psi_t}} + \frac{n\varrho_t}{s^2}}_{\textit{Main Privacy Term}} + \underbrace{\mathcal{O}\left( \sqrt{\gamma_t} + \tilde{\gamma}_t \right)}_{\textit{Error Term}} \right] / (\epsilon(1 - e^{-\epsilon})) \tag{11}$$

where $\Psi_t, \tilde{c}_t^*, \tilde{\eta}_t, \varrho_t$ are quantities that **only depend on the diffusion coefficients**, and $\tilde{c}_t^*, \tilde{\eta}_t$ meet certain radius selection criteria (refer to Eq. (50) and Eq. (49) in Appendix E). As t progresses from $T$ to $1$, the value of $\frac{1}{s^{\Psi_t}}$ increases from $\mathcal{O}_s\left(\frac{1}{s^2}\right)$ to $\mathcal{O}s(1)$, while $N_{\eta_t'}/s$ monotonically decreases from $\mathcal{O}_s(1)$ to $\mathcal{O}_s\left(\frac{1}{s}\right)$.

As indicated by Theorem 3, akin to findings from pDP, during the generation process, privacy leakage intensifies from $\mathcal{O}_s\left(\frac{m}{s^2}\right)/[\epsilon(1 - e^{-\epsilon})]$ to $\mathcal{O}_s\left(\frac{m}{s}\right)/[\epsilon(1 - e^{-\epsilon})]$. The formal presentation and analysis of DP guarantees for DDMs are detailed in Appendix E.

### 3.2 Impact of DDM Coefficients and Dataset Distributions on the Privacy Bound

**Influence of Diffusion Coefficients.** The privacy term is largely influenced by the proximity between $\mathbf{v}^*$ and $\mathcal{V}_1$. As time $t$ progresses, this similarity is governed by the transition ratio $\tilde{R}_t$. A faster rate of diffusion coefficients going to zero boosts this ratio, enhancing the privacy guarantee. Experiments in Sec. 4 validate this observation.

**Impact of Dataset Distribution.** We find that $\psi_t$ has a major effect on the privacy bound. $\psi_t$ is influenced by the similarity between the additional point $\mathbf{v}^*$ and $\mathcal{V}_0 \backslash \{\mathbf{v}^*\}$. If $\mathbf{v}^*$ is far away from (close to) the rest points in $\mathcal{V}_0$, then $\text{Sim}(\mathcal{V}_0 \backslash \{\mathbf{v}^*\}, \mathbf{v}^*, t)$ becomes small (large) and the corresponding term $s^{-\psi_t}$ become large (small), which indicates weaker (stronger) protection of $\mathbf{v}^*$. *This indicates that points with notably low $Sim(\mathcal{V}_0 \backslash \{\mathbf{v}^*\}, \mathbf{v}^*, t)$ are probably sensitive points in the dataset.*

### 3.3 Characterizing Data-dependent Quantities under Simple Distributions

Here, we consider the training dataset sampled from some specific distributions to further illustrate the data-dependent quantities.

Consider a distribution such that each column independently takes value $l \in [k]$ with probability $p$ $(p \geq \frac{1}{k})$ and any other $k - 1$ categories with probability $\frac{1-p}{k-1}$. Let $\mathbf{v}^*$ take non-majority category $((\mathbf{v}^*)^i \neq l)$ along all $n$ columns (termed **non-majority points**, which thus tends to have higher privacy leakage) and the rest points in $\mathcal{V}_0 \backslash \{\mathbf{v}^*\}$ are sampled from the distribution. We have the following characterization (For detailed explanations and proofs, please refer to Appendix F).

- $\frac{1}{s^{\psi_t}}$. For a sufficiently large $s$ (detailed in appendix), with high probability, $\frac{1}{s^{\psi_t-2}} \to \frac{(\overline{\alpha}_{t-1} - \overline{\alpha}_t)/(k\bar{\mu}_t^+ \bar{\mu}_t^-)}{\bar{R}_{t-1}^2 \cdot \tau_t^{2n-1} \cdot \frac{1-p}{k-1} + \tau_t^{2n}}$, where $\tau_t := \frac{1-p}{k-1} + \frac{\bar{\mu}_t^-}{\bar{\mu}_t^+}(1 - \frac{1-p}{k-1})$. In the noisy regime (a large t, $\frac{\bar{\mu}_t^-}{\bar{\mu}_t^+} \to 1$), $\tau_t \to 1$, $\frac{1}{s^{\psi_t}} = \mathcal{O}_s(\frac{1}{s^2})$. For distribution characterized by larger skewness, i.e., larger $p$, we have smaller $\tau_t$ result in larger $\frac{1}{s^{\psi_t}}$. Fig. 4 (LEFT) precisely matches the above conclusions.

- $\eta_t, c_t^*$. For a sufficiently large $s$ (detailed in appendix), a sufficient condition for $\eta_t$ and $c_t^*$ to satisfy Eq. (9) is

$$\eta_t \geq n - \left( \frac{n - \log(s\sqrt{\frac{\overline{\alpha}_{t-1} - \overline{\alpha}_t}{k\bar{\mu}_t^+ \bar{\mu}_t^-}})/\log \frac{\bar{\mu}_t^+}{\bar{\mu}_t^-}}{2\log \frac{k-1}{1-p}/\log(\max\{\frac{1}{n\bar{\mu}_t^-}, 1\}) + 1} \right)_+, \quad c_t^* \geq \frac{\frac{n-\eta_t}{\eta_t}\log \frac{k-1}{1-p} - \log \frac{1}{2e}}{\log \frac{k-1}{1-p} + \log \frac{1}{e\bar{\mu}_t^-}}. \tag{12}$$

In the noise free regime ($\alpha_t \to 1$), $\eta_t \to 0$, while in the noise full regime ($\alpha_t \to 0$), $\eta_t \to n$. From noise free regime to noisy regime, $\bar{\mu}_t$ increases, $c_t^* \to \frac{n-\eta_t}{\eta_t}$. Furthermore, as we rise in the skewness ($p$) of the distribution, the R.H.S of Eq. 12 monotonically increases, and results in larger values for $\eta_t$ and $c_t^*$. Fig. 4 (MIDDLE) matches the above conclusions.

### 3.4 The Algorithm for Evaluating Privacy Bound in Eq. (7) on a given Dataset

In practical situations, when data curators release synthetic data, it is crucial to assess the privacy safeguards of the mechanism trained on a specific dataset. This ensures the synthetic data upholds privacy and the confidentiality of the training data's sensitive information. To this end, we introduce Algorithm 1 (paired with Algorithm 2), to compute the privacy bound, enabling direct per-instance privacy leakage calculation for DDM-generated datasets given particular training sets. Specifically, for each $\mathbf{v}^*$, we determine $\psi_t$, $\eta_t$, and $c_t^*$ to compute $\delta(\mathbf{v}^*, \mathcal{V}_0)$ using Eq. (7). Using this algorithm, data curator can have better assessment of the potential privacy leakage of each point in training set and may exclude sensitive points $\mathbf{v}^*$ (outliers) with high $\delta(\mathbf{v}^*, \mathcal{V}_0)$ to enhance privacy protection. This approach's efficacy is confirmed with real dataset experiments in Sec. 4. The total time complexity of the algorithm is further analyzed in Appendix C.

---

**Algorithm 1** Privacy Bound for Discrete Diffusion Models

---

1: **Input:** Dataset $\mathcal{V}_0$; Diffusion Step: $T$; Diffusion Coefficients: $\{\alpha_t\}_{t \in [T]}$ .
2: For each $\mathbf{v}, \mathbf{v}' \in \mathcal{V}_0$, calculate and store $\bar{\omega}(\mathbf{v}, \mathbf{v}')$ in $\mathcal{T}_1$ and $N_\eta(\mathbf{v})$ in $\mathcal{T}_2$ for $\eta \in [n]$.
3: Use diffusion coefficients $\{\alpha_t\}_{t \in [T]}$ to calculate $\{\mu_t^+, \mu_t^-, \bar{\mu}_t^+, \bar{\mu}_t^-\}_{t \in \{1,2,...,T\}}$.
4: Define empty array `Privacy_Bound` $= []$.
5: **for** For every $\mathbf{v} \in \mathcal{V}_0$ **do**
6:     Define empty array `Array`$_{c_t^*} = []$.
7:     **for** $t \leftarrow 1$ to $T$ **do**
8:         **for** $\eta_t \leftarrow 1$ to $n$ **do**
9:             Calculate $c_t^*$ using `Alg`$_{c_t^*}$ (Algorithm 2) and determine $\{1/s^{\psi_t}\}$ with $\mathcal{T}_1$ and $\{\mu_t^+, \mu_t^-\}$.
10:            If $\eta_t$ meets the condition in Eq.(9) using $N_{\eta_t}(\mathbf{v})$ and $N_{(1+c_t^*)\eta_t}(\mathbf{v})$ from $\mathcal{T}_2$, then **break**.
11:        **end for**
12:        `Array`$_{c_t^*}[t] \longleftarrow (1 + c_t^*)\eta_t(\mathbf{v})$
13:     **end for**
14:     Use `Array`$_{c_t^*}$ to compute and append

$$\texttt{Privacy\_Bound}(\mathbf{v}) \longleftarrow m \sum_{t=T_{\mathrm{rl}}}^{T} \left[ \min\left\{ \frac{4N_{(1+c_t^*)\eta_t}(\mathbf{v})}{s}, 1 \right\} \cdot \frac{n}{s^{\psi_t}} + \frac{n(1 - \frac{\bar{\mu}_{t-1}^-}{\bar{\mu}_{t-1}^+})}{s^2} \right] / [\epsilon(1 - e^{-\epsilon})]$$

15: **end for**
16: **Output:** `Privacy_Bound`

---

**Algorithm 2** `Alg`$_{c_t^*}$: Finding $c_t^*$

---

1: **Input:** $\eta_t, \alpha_t, N_\eta, \mathbf{v}^*$.
2: **for** $t = 1 : T$ **do**
3:     **if** $\frac{1}{\vartheta(2\eta_t)} > (2e\bar{\mu}_t^-)^{\eta_t}$ **then**
4:         ① Select the smallest $c_t^* \in \{0, \frac{1}{\eta_t}, \frac{2}{\eta_t}, ..., \frac{\eta_t - 1}{\eta_t}\}$ such that $c_t^* \geq \frac{\frac{1}{\eta_t} \log \vartheta((1+c_t^*)\eta_t) + 1 + \log 2}{\log \frac{1}{\mu_t^-}}$
5:     **else if** $\frac{1}{\vartheta(2\eta_t)} \leq (2e\bar{\mu}_t^-)^{\eta_t}$ **then**
6:         ② Select the smallest $c_t^* \in \{1, \frac{\eta_t + 1}{\eta_t}, \frac{\eta_t + 2}{\eta_t}, ..., \frac{n - \eta_t}{\eta_t}\}$ such that $c_t^* \geq \frac{\frac{1}{\eta_t} \log \vartheta((1+c_t^*)\eta_t)}{\log \frac{1}{\mu_t^-} - 1}$
7:     **end if**
8: **end for**
9: **Output:** $c_t^*$

---

## 4 Experiments

We validate our theoretical findings via computational simulations on synthetic and real-world datasets.

## 4.1 Synthetic Experiments

We first study the asymptotic behavior of privacy leakage with respect to the training dataset size $s$. Given a DDM with 100 diffusion steps and trained with a linear schedule $\alpha_t = 1 - \frac{t}{T}$, we fix $\mathbf{v}^*$ and increase the number of samples in the training set from $1e4$ to $1e7$, ensuring that the newly added samples satisfy $\bar{\omega}(\mathbf{v}, \mathbf{v}^*) = n$, which makes $\mathbf{v}^*$ with high privacy leakage risk. Results shown in Fig. 5 (LEFT, MIDDLE) confirm our theoretical prediction that, in noise-free regime ($t = 1$, Fig. 5 (LEFT)), the **main privacy term** in Theorem 1 is $\mathcal{O}_s(\frac{1}{s})$, which is almost a linear decay with a slope of $-1$ in the logarithmic scale (all lines in the figure). On the other hand, in the noisy-regime ($t = 50$, Fig. 5 (MIDDLE)), the privacy leakage term decays faster at the rate of $\mathcal{O}_s(\frac{1}{s^2})$, which is evident from the linear decay with a slope around $-2$. In the second experiment, we examine how decay rate of diffusion coefficients affects the privacy bound. Given specific $\mathbf{v}^*$ (non-majority categories along all entries), we sample the training set from the distribution with $p = 0.5$ in Sec. 3.3. We consider two noise schedules: linear schedule and sigmoid schedule. In Fig. 5 RIGHT, the red line denotes the linear schedule with decay rate $\in \{0.1, 0.3, 0.5, 0.7, 0.9\}$ and the blue line denotes the Sigmoid schedule where decay rate increases from 2.5 to 5. $\delta$ decreases along both two lines as we increase the decay rate of diffusion coefficients. This indicates that a faster decay rate in diffusion coefficients implies better privacy.

More results discussing privacy leakage and the behaviors of data-dependent quantities under various DDM configurations are given in Appendix I.

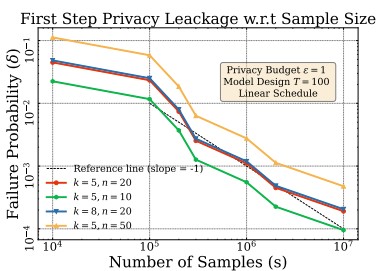 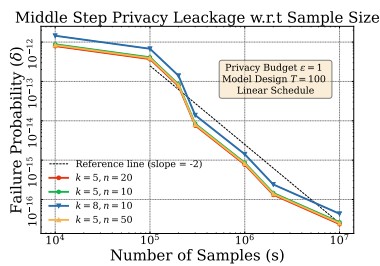 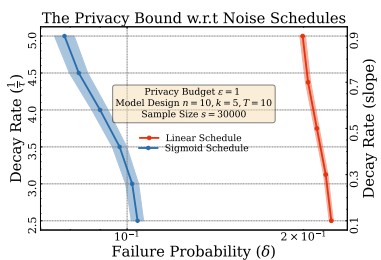

Figure 5: **LEFT:** Privacy Leakage at $t = 1$ (Noise-free Regime). **MIDDLE:** Privacy Leakage at $t = 50$ (Noisy Regime). **Right:** Privacy Leakage w.r.t Decay Rate under Linear ($\alpha_t = 1 - \text{decay rate} * \frac{t}{T}$) and Sigmoid ($\alpha_t = \frac{\text{Sigmoid}(3*\text{decay rate}) - \text{Sigmoid}(\frac{3t}{T}*\text{decay rate})}{\text{Sigmoid}(3*\text{decay rate}) - 0.5}$) Schedules. Results are based on 5 times independent tests.

## 4.2 Experiments on Real Datasets

### 4.2.1 Effectiveness of Privacy Bound Algorithm for Data Sensitivity Assessment

In this series of experiments, we aim to showcase our privacy algorithm's effectiveness in assessing the sensitivity of individual data points within real-world datasets and to delineate the relationship between sample privacy leakage and dataset similarity (as illustrated in Fig. 3). Additionally, from a data curator's perspective, we explore the potential of our algorithm assisting in outlier removal, which may enhance privacy protections while maintaining utility performance. It is important to note that the use of pDP assessment itself is kept confidential to the data curator, safeguarding against any privacy concerns. We evaluate our algorithm on three benchmark datasets: Adult (Kohavi et al., 1996), German Credit (Hofmann, 1994), and Loan (ItsSuru) with (# training samples, # feature dimensions, # categories) of (30718, 9, 5), (1000, 10, 5), and (480, 11, 4) (see Appendix H for details).

**Experimental Settings.** Our study, approaching from the perspective of a data curator who preprocesses the dataset, investigates how varying the sensitive data removal ratio according to our per-instance privacy bound assessment can potentially assist data curators in eliminating outliers to enhance privacy protection. Specifically, we calculate the privacy budget for every point in the dataset according to Eq. (7) via Algorithm 1 and remove the most sensitive points according to the assessment in the dataset amounting to a specific portion which is controlled by the removal ratio. The removal ratio ranges from 0.01 to 0.5 for the Adult and German Credit datasets, and between 0.001 and 0.05 for the Loan dataset. It is important to underscore

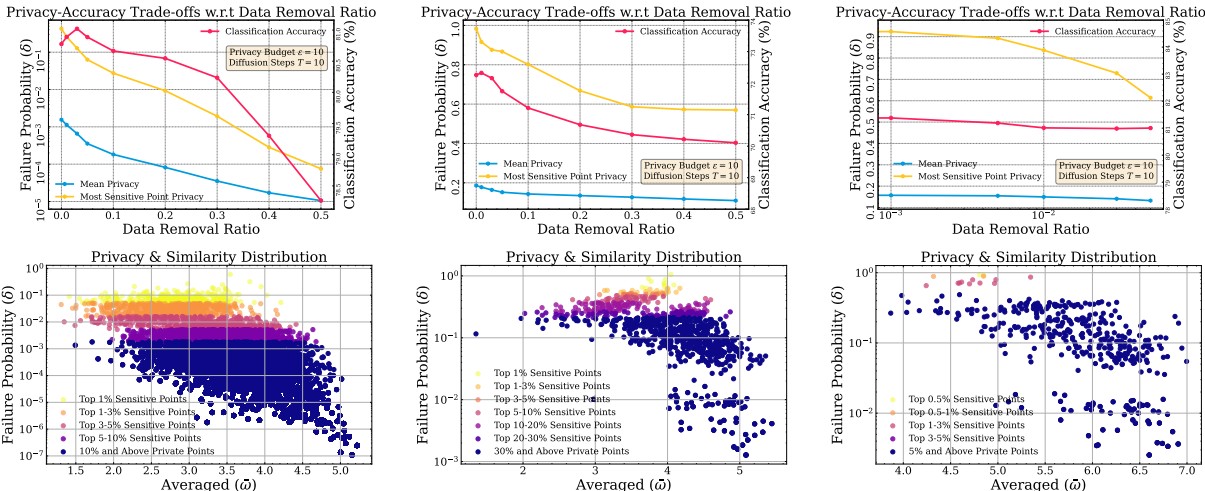

Figure 6: **First Row:** Privacy-utility trade-offs with respect to data removal ratio. **LEFT:** Adult, **MIDDLE:** German Credit, **RIGHT:** Loan. **Experimental Setup:** DDM design: T = 10, Linear Schedule. **Second Row:** Visualizing privacy budget in relation to average feature overlap. **LEFT:** Adult, **MIDDLE:** German Credit, **RIGHT:** Loan.

that, after each data removal, we conduct ***privacy recalculation*** and report the mean privacy leakage (blue line) and the most sensitive point privacy budget (yellow line) after each data removal process in Fig. 6 (First Row). We measure utility performance with respect to downstream classification task by training a binary classifier on DDM-generated samples and evaluate its performance (red line) on the original dataset. We further illustrate the sensitive points—those removed from the dataset—by graphing their potential privacy leakage alongside the average overlap with the entire dataset across all feature dimensions, denoted by $\bar{\omega}$. The visualizations are presented in Figure 6 (Second Row).

**Remove privacy sensitive points with comparable utility.** As depicted in Fig. 6 (First Row), eliminating a minor proportion of the most sensitive points from the dataset results in a decrease in privacy leakage. Meanwhile, the classification accuracy (red line) only gets slightly decreased: $81\% \rightarrow 78\%$ for Adult, $73\% \rightarrow 70\%$ for German Credit, $81.1\% \rightarrow 79.8\%$ for Loan (note that for Loan we remove at most 5% data points as its size is too small) More interestingly, by removing a certain number of those most sensitive data points, the classification model trained over the pruned generated dataset may even achieve better performance over the original dataset, say removing 3% in Adult and 1% in German Credit. We attribute such gains to the fact that the most sensitive data points are often outliers in the dataset, which may be actually not good for training an ML model. In data visualization (Fig. 6, Second Row), we note that the data points prone to greater privacy leakage tend to have less feature overlap, indicating that these data points have a lower similarity to others in the dataset. As pointed out in (Carlini et al., 2022), the mere exclusion of the most sensitive data points from the dataset does not necessarily guarantee a reduction in privacy risks as certain inliers may become outliers post-removal, a phenomenon termed "Onion Effect". However, in our experiments, we did not observe the "Onion Effect" within the Adult dataset. Empirically, we did find that as the data removal ratios increase to a certain extent, privacy leakage ceases to decrease and may fluctuate. This can be attributed to the fact that an individual data point's privacy risk is determined by its similarity to the current dataset distribution. Simply removing certain data points does not necessarily enhance this similarity, and thus privacy leakage may not decrease. Furthermore, a decrease in dataset size also intensifies privacy leakage. Therefore, in practice, data curators should perform pDP recalculations on modified datasets with varying data removal ratios to potentially achieve better privacy-utility trade-offs.

### 4.2.2 Evaluation of DDM Vulnerability to Black-box Membership Inference Attacks

In this subsection, we further investigate the privacy leakage of DDMs from membership inference attacks perspective.

**Black-box Attacks with No Auxiliary Knowledge:** In alignment with the experimental settings delineated by (Hayes et al., 2017), this study considers *black-box attacks* where the attacker has no prior knowledge or external information regarding the target model, i.e. DDM in our case, including *model parameters / hyper-parameters, model architecture, training data*, or *prediction scores*. For evaluation, it is hypothesized that the attacker possesses access to the whole dataset, denoted as $X = X_{\text{train}} \cup X_{\text{non-train}}$. Additionally, it is presumed that the adversary is aware of the size of the training set. The attack primarily utilizes the target model's generated samples to identify and exploit its vulnerabilities.

**Experimental Settings:** Our experiments utilize the Adult dataset. We partition this dataset by randomly selecting 20% of the records as the training set, denoted as $X_{\text{train}}$, while the remainder is labeled as $X_{\text{non-train}}$. We train a DDM to learn the conditional distribution of $X_{\text{train}}$ given their corresponding labels $Y_{\text{train}}$.

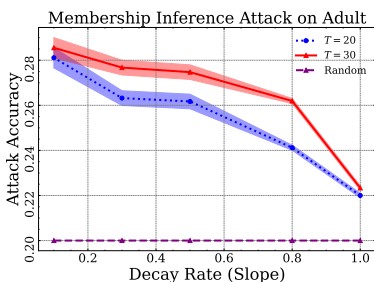

Figure 7: Black-box Attack with no auxiliary knowledge over DDMs under various designs. Results are averaged over 3 independent tests.

Next, the adversary employs a discriminative model denoted as $f(\cdot; \theta)$ : feature space $\mathcal{X} \mapsto$ label space $\mathcal{Y}$ (detailed in Appendix. H), which is trained using two-class samples $(X_{\text{gen}}, Y_{\text{gen}})$ generated by the DDM and denote trained model weights as $\theta_{\text{gen}}$. This trained discriminative model $f(\cdot; \theta_{\text{gen}})$ is then used to make predictions across the entire dataset $X_{\text{train}} \cup X_{\text{non-train}}$. We identify $|X_{\text{train}}|$ samples with the highest confidence values—those whose prediction scores are closest to the true labels—and designate these as training members $\tilde{X}_{\text{train}}$. The evaluation of the DDMs is centered around varying the decay rate of diffusion coefficients, influencing the degree of privacy leakage in the models. The outcomes of these evaluations, particularly the attack accuracy ($\frac{|\tilde{X}_{\text{train}} \cap X_{\text{train}}|}{|X_{\text{train}}|}$), are illustrated in Fig. 7. More specifically, we examine DDMs with total diffusion steps $T$ set to either 20 or 30, and a linear schedule defined as $\alpha_t = 1 - \text{decay rate} \times \frac{t}{T}$. Additionally, the decay rate is varied within the range $\{0.1, 0.3, 0.5, 0.8, 1.0\}$ to adjust the privacy guarantees of the DDMs, with a faster decay rate typically providing better privacy protection.

**Increased Privacy Leakage Enhances Vulnerability of Models to Attacks:** As illustrated in Fig. 7, the blue and red lines represent DDMs with total diffusion steps of 30 and 20, respectively. A noteworthy observation is that the DDM with $T = 30$ exhibits higher attack accuracy, suggesting a stronger capability for memorizing training data in models with a greater number of diffusion steps. Furthermore, an increase in the decay rate leads to improved privacy guarantees for the DDMs. This enhancement in privacy is evidenced by a decrease in attack accuracy for both models (as represented by the blue and red lines), which diminishes from 28% / 29% to approximately 22%. It is important to note that the performance of both lines surpasses that of random guessing (indicated by the purple line), signifying that our target models retain training data memorization across all evaluated settings.

# 5 Conclusion

In this work, we analyzed data-dependent privacy bound for the synthetic datasets generated by DDMs, which revealed a weak privacy guarantee of DDMs. Thus, to meet practical needs, other privacy-preserving techniques such as DP-SGD (Abadi et al., 2016) and PATE (Papernot et al., 2016) may have to be corporated. Our findings well align with empirical observations over synthetic and real datasets.

# 6 Limitations and Future Work

- Our research currently targets discrete-time diffusion models for discrete data. An intriguing extension of this work could explore continuous diffusion models applied to continuous data domains, such as images, which present numerous practical applications. We believe the core logic of our proof concept is adaptable to continuous diffusion models, albeit some of our current arguments, grounded in the characteristics of discrete distributions, may not directly apply or might necessitate further examination.

- Our findings are predicated on the assumptions concerning the denoising networks' expressive power and the total variation gap between forward and backward diffusion trajectories. Relaxing these assumptions can be an interesting direction for future work.

## 7    Acknowledgement

We extend our sincere gratitude to Zinan Lin for his insightful discussions and contributions to this project. We also appreciate the reviewers for their valuable feedback and actionable suggestions. R.Wei, H.Wang, H.Yin, E.Chien and P.Li are partially supported by 2023 JPMorgan Faculty Award and NSF IIS-2239565.

## 8    Disclaimer

This paper was prepared for informational purposes by the Artificial Intelligence Research group of JPMorgan Chase & Co. and its affiliates ("JP Morgan"), and is not a product of the Research Department of JP Morgan. JP Morgan makes no representation and warranty whatsoever and disclaims all liability, for the completeness, accuracy or reliability of the information contained herein. This document is not intended as investment research or investment advice, or a recommendation, offer or solicitation for the purchase or sale of any security, financial instrument, financial product or service, or to be used in any way for evaluating the merits of participating in any transaction, and shall not constitute a solicitation under any jurisdiction or to any person, if such solicitation under such jurisdiction or to such person would be unlawful.

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

# Appendix

## Contents

# A Proof of Theorem 1

## A.1 Notations

In this section, we introduce and recall important notations for the convenience of presenting proofs. Let $\mathbf{v}_{t|\lambda}, t \in [T], \lambda \in \{0,1\}$ denote the intermediate data distribution r.v.s at time $t$ trained with dataset $\lambda$ in the forward $q(\cdot)$ and backward $p(\cdot)$ processes. Recall some notations $\mu_t^+ = (1+(k-1)\alpha_t)/k, \mu_t^- = (1-\alpha_t)/k, \bar{\mu}_t^+ = (1+(k-1)\bar{\alpha}_t)/k, \bar{\mu}_t^- = (1-\bar{\alpha}_t)/k, R_t = \mu_t^+/\mu_t^-, \bar{R}_t = \bar{\mu}_t^+/\bar{\mu}_t^-$, and similarity $\mathrm{Sim}(\mathcal{V}_1, \mathbf{v}^*, t) = \sum_{\mathbf{v} \in \mathcal{V}_1} \bar{R}_t^{-\bar{\omega}(\mathbf{v}, \mathbf{v}^*)}$. Let $N_\eta(\mathbf{v}^*) = |\{\mathbf{v} \in \mathcal{V}_1 \text{ s.t. } \bar{\omega}(\mathbf{v}, \mathbf{v}^*) \leq \eta\}|$ and $\Delta N_\eta(\mathbf{v}^*) = |\{\mathbf{v} \in \mathcal{V}_1 \text{ s.t. } \bar{\omega}(\mathbf{v}, \mathbf{v}^*) = \eta\}|$. Let $\mathbb{1}_{\mathbf{v} \neq \mathbf{v}'}, \mathbb{1}_{\mathbf{v} = \mathbf{v}'}$ denote the characteristic function over $\mathbf{v}$ and $\mathbf{v}'$. Given dataset $\mathcal{V}$, define $\mathcal{V}_1^{i|l} = \{\mathbf{v} \in \mathcal{V}_1 | \mathbf{v}^i = l\}$. Further, $(\cdot)_+ = \max\{\cdot, 0\}$.

Note that in certain cases, when there is no potential for confusion, we employ a slight abuse of notation by representing intermediate data distribution r.v.s $\mathbf{v}_{t|\lambda}$ in its one-hot encoding form, and we use $\mathcal{M}_t(\mathcal{V})$ as shorthand for $\mathcal{M}_t(\mathcal{V}; 1)$.

## A.2 Proof Sketch of Theorem 1

**Characterizing Privacy Leakage with Expected Conditional KL Divergence.** First, we leverage coupled KL divergence to quantify the inherent privacy guarantees of the DDM-generated samples. Given adjacent datasets $\mathcal{V}_0, \mathcal{V}_1(\mathcal{V}_0 \backslash \{\mathbf{v}^*\} = \mathcal{V}_1)$, the mechanism $\mathcal{M}_t$ satisfies $(\epsilon, \frac{\tau}{\epsilon(1-e^{-\epsilon})})$-pDP with respect to $(\mathcal{V}_0, \mathbf{v}^*)$ if $\mathcal{D}_{\mathrm{KL}}(\mathcal{M}_t(\mathcal{V}_0)\|\mathcal{M}_t(\mathcal{V}_1)) + \mathcal{D}_{\mathrm{KL}}(\mathcal{M}_t(\mathcal{V}_1)\|\mathcal{M}_t(\mathcal{V}_0)) \leq \tau$. Note that the generation process $\mathbf{v}_{T|\lambda} \to \mathbf{v}_{T-1|\lambda} \to \cdots \mathbf{v}_{t|\lambda}$ forms a Markov chain, and in order to keep track of privacy leakage from each step, we upper bound the coupled marginal KL divergence by summation over conditional KL divergences. By leveraging sufficient training, backward path approximation (Assumption 1 and 2) with conditional independence properties, we obtain

$$\mathcal{D}_{\mathrm{KL}}(\mathcal{M}_{T_{\mathrm{rl}}}(\mathcal{V}_0)\|\mathcal{M}_{T_{\mathrm{rl}}}(\mathcal{V}_1)) + \mathcal{D}_{\mathrm{KL}}(\mathcal{M}_{T_{\mathrm{rl}}}(\mathcal{V}_1)\|\mathcal{M}_{T_{\mathrm{rl}}}(\mathcal{V}_0)) \leq \sum_{t=T_{\mathrm{rl}}}^{T} (\mathscr{B}_1 + \mathscr{B}_2) + \mathcal{O}(\sqrt{\gamma_t} + \tilde{\gamma}_t) \quad (13)$$

where

$$\mathscr{B}_1 = \mathbb{E}_{\mathbf{v}_t \sim q(\mathbf{v}_{t|0})} \sum_{i=1}^{n} [\mathcal{D}_{\mathrm{KL}}(q(\mathbf{v}_{t-1|0}^i|\mathbf{v}_{t|0})\|q(\mathbf{v}_{t-1|1}^i|\mathbf{v}_{t|1})) + \mathcal{D}_{\mathrm{KL}}(q(\mathbf{v}_{t-1|1}^i|\mathbf{v}_{t|1})\|q(\mathbf{v}_{t-1|0}^i|\mathbf{v}_{t|0}))]$$

represents the expected coupled KL divergence for each feature, conditioned on the data at time $t$, and

$$\mathscr{B}_2 = \sum_{i=1}^{n} \mathbb{E}_{\mathbf{v}_t \sim (q(\mathbf{v}_{t|1}) - q(\mathbf{v}_{t|0}))} [\mathcal{D}_{\mathrm{KL}}(q(\mathbf{v}_{t-1|1}^i|\mathbf{v}_{t|1})\|q(\mathbf{v}_{t-1|0}^i|\mathbf{v}_{t|0}))]$$

describes conditional KL averaged over measure gap and can be directly upper bounded by $\frac{n\varrho_t}{s^2}$.

**Measure Partition to Compute Expected Conditional KL Divergence in $\mathscr{B}_1$.** When quantifying $\mathscr{B}_1$, certain regions of the space $\mathcal{X}^n$ may have a large measure under $q(\mathbf{v}_{t|0})$, but a small $\mathcal{D}_{\mathrm{KL}}$ gap, while other regions may exhibit the opposite behavior (illustrated in Fig 2). Thus, our objective is to partition the probability measure into distinct sets, with the aim of achieving a balance between the expectations of $\mathcal{D}_{\mathrm{KL}}$ in each set (can be interpreted as achieving balanced privacy leakage). Such balance can be achieved by selecting two radius parameters $\eta_t, \eta_t' \in \{0, 1, ..., n\}$ to partition the space according to the distances to $\mathbf{v}^*$ and $\mathbf{v}_0 \in \mathcal{V}_1$. We partition the measure $q(\mathbf{v}_{t|0})$ into the following three parts under different conditions of $\eta_t$ and $\eta_t'$: (1) $q(\mathbf{v}_t \in \mathcal{S}_a)$ quantifies the $q(\mathbf{v}_{t|0})$ measure for points near $\mathbf{v}^*$ (within $\eta_t$-ball) or those generated from dataset points close to $\mathbf{v}^*$. (2) $q(\mathbf{v}_t \in \mathcal{S}_b)$ signifies the $q(\mathbf{v}_{t|0})$ measure for points that diffused from points in dataset $\mathcal{V}_1$ position distant from $\mathbf{v}^*$ (beyond the $\eta_t$-ball) and are closer to some elements in $\mathcal{V}_1$ than to $\mathbf{v}^*$ by a margin of at least $\eta_t'$. (3) $q(\mathbf{v}_t \in \mathcal{S}_c)$ represents the $q(\mathbf{v}_{t|0})$ measure for points that diffused from points in dataset $\mathcal{V}_1$ position distant from $\mathbf{v}^*$ (beyond the $\eta_t$-ball), yet closer to any points in $\mathcal{V}_1$ than to $\mathbf{v}^*$ with no morn than $\eta_t'$. As adjacent datasets differ by an element $\mathbf{v}^*$, the privacy leakage in $\mathcal{S}_a$ and $\mathcal{S}_c$ is

notably higher than in $\mathcal{S}_b$. However, for larger datasets, $q(\mathbf{v}_t \in \mathcal{S}_b)$ often constitutes a significant portion of the original data measure $q(\mathbf{v}_{t|0})$. This is because the likelihood of finding points in the dataset close to $\mathbf{v}^*$ within a margin of $\eta'_t$ is high, especially when $\eta'_t$ is small. We show that $\mathscr{B}_1$ has general data-dependent upper bound $n/s^{\psi_t}$. For $\mathbf{v} \in \mathcal{S}_b$, we can further improve the bound to $n(\overline{\alpha}_{t-1} - \overline{\alpha}_t)/[(k\bar{\mu}_t^+ \bar{\mu}_t^-)(\bar{R}_{t-1}^2 \bar{R}_t^{2\eta'_t})]$. Thus, we upper bound $\mathscr{B}_1$ as

$$\mathscr{B}_1 \leq q(\mathbf{v}_t \in \mathcal{S}_a) \cdot \frac{n}{s^{\psi_t}} + q(\mathbf{v}_t \in \mathcal{S}_b) \cdot \frac{n(\overline{\alpha}_{t-1} - \overline{\alpha}_t)/(k\bar{\mu}_t^+ \bar{\mu}_t^-)}{\bar{R}_{t-1}^2 \cdot \bar{R}_t^{2\eta'_t}} + q(\mathbf{v}_t \in \mathcal{S}_c) \cdot \frac{n}{s^{\psi_t}}. \tag{14}$$

Quantities $\eta_t$ and $\eta'_t$ can be adjusted to balance the three terms on the R.H.S of Eq. (14) such that $\mathscr{B}_1 \leq 2q(\mathbf{v}_t \in \mathcal{S}_a) \cdot \frac{n}{s^{\psi_t}}$. Finally, the data-dependent quantity $c_t^*$ is introduced to give a more accurate characterization of $q(\mathbf{v} \in \mathcal{S}_a)$. By selection $c_t^*$ satisfies Eq. (9), we have $q(\mathbf{v} \in \mathcal{S}_a) \leq \frac{2N_{(1+c_t^*)\eta_t}}{s}$. Hence, by gluing the pieces together, we completes the proof.

### A.3 Main Proof

In this subsection, we provide a rigorous proof of Theorem 1. The related lemma proofs are deferred to Appendix J.

**Step 1: Characterizing pDP with Marginal KL Divergence.** First, we quantify the inherent DP guarantees of the DDM-generated samples using Coupled KL divergence.

**Lemma A.1** (Characterizing pDP with Coupled KL Divergence). Given two adjacent dataset $\mathcal{V}_0, \mathcal{V}_1(\mathcal{V}_0\backslash\{\mathbf{v}^*\} = \mathcal{V}_1)$ and a specific time step $T_{\mathrm{rl}}$, if the mechanism $\mathcal{M}_{T_{\mathrm{rl}}}$ satisfies the condition:

$$\mathcal{D}_{\mathrm{KL}}(\mathcal{M}_{T_{\mathrm{rl}}}(\mathcal{V}_0)\|\mathcal{M}_{T_{\mathrm{rl}}}(\mathcal{V}_1)) + \mathcal{D}_{\mathrm{KL}}(\mathcal{M}_{T_{\mathrm{rl}}}(\mathcal{V}_1)\|\mathcal{M}_{T_{\mathrm{rl}}}(\mathcal{V}_0)) \leq \tau \tag{15}$$

then the mechanism satisfies $(\epsilon, \frac{\tau}{\epsilon(1-e^{-\epsilon})})$-pDP with respect to $(\mathcal{V}_0, \mathbf{v}^*)$.

From this lemma, we know that the pDP guarantees of mechanism $\mathcal{M}_{T_{\mathrm{rl}}}$ can be characterized by coupled KL divergence.

**Step 2: Upper Bounding Marginal KL Divergence with Expected Conditional KL Divergence.** Note that the generation process $\mathbf{v}_{T|\lambda} \to \mathbf{v}_{T-1|\lambda} \to \cdots \mathbf{v}_{t|\lambda}$ forms a Markov chain, and the coupled marginal KL divergence can be bounded by summation over conditional KL divergences.

**Lemma A.2.** Given two adjacent datasets $\mathcal{V}_0, \mathcal{V}_1(\mathcal{V}_0\backslash\{\mathbf{v}^*\} = \mathcal{V}_1)$ and a specific time step $T_{\mathrm{rl}}$, consider the mechanism $\mathcal{M}_{T_{\mathrm{rl}}}$ for generating $m$ samples, we have

$$\mathcal{D}_{\mathrm{KL}}(\mathcal{M}_{T_{\mathrm{rl}}}(\mathcal{V}_0)\|\mathcal{M}_{T_{\mathrm{rl}}}(\mathcal{V}_1)) + \mathcal{D}_{\mathrm{KL}}(\mathcal{M}_{T_{\mathrm{rl}}}(\mathcal{V}_1)\|\mathcal{M}_{T_{\mathrm{rl}}}(\mathcal{V}_0))$$
$$\leq m \sum_{t=T_{\mathrm{rl}}+1}^{T} \sum_{\lambda \in \{0,1\}} \sum_{i=1}^{n} \mathbb{E}_{\mathbf{v}_t \sim p_\phi(\mathbf{v}_{t|\lambda})}[\mathcal{D}_{\mathrm{KL}}(p_\phi(\mathbf{v}_{t-1|\lambda}^i|\mathbf{v}_{t|\lambda})\|p_\phi(\mathbf{v}_{t-1|1-\lambda}^i|\mathbf{v}_{t|1-\lambda}))] \tag{16}$$

**Step 3: Relating Backward Conditional KL Divergence with Forward Process.**

**Lemma A.3.** Assume the denoising networks trained on $\mathcal{V}_0$ and $\mathcal{V}_1$ satisfy Assumption 1 and Assumption 2, we have for any $i \in \{1, 2, ..., n\}$

$$\sum_{\lambda \in \{0,1\}} \mathbb{E}_{\mathbf{v}_t \sim p_\phi(\mathbf{v}_{t|\lambda})}[\mathcal{D}_{\mathrm{KL}}(p_\phi(\mathbf{v}_{t-1|\lambda}^i|\mathbf{v}_{t|\lambda})\|p_\phi(\mathbf{v}_{t-1|1-\lambda}^i|\mathbf{v}_{t|1-\lambda}))]$$
$$\leq \sum_{\lambda \in \{0,1\}} \mathbb{E}_{\mathbf{v}_t \sim q(\mathbf{v}_{t|\lambda})}[\mathcal{D}_{\mathrm{KL}}(q(\mathbf{v}_{t-1|\lambda}^i|\mathbf{v}_{t|\lambda})\|q(\mathbf{v}_{t-1|1-\lambda}^i|\mathbf{v}_{t|1-\lambda}))] + f_1(\gamma_t) + f_2(\tilde{\gamma}_t) \tag{17}$$

where $f_1(\gamma_t) = \frac{k \cdot (\mu_t^+ \cdot \bar{\mu}_{t-1}^+)^2}{\bar{\mu}_t^+ \cdot \mu_t^- \cdot \bar{\mu}_{t-1}^-} \cdot \sqrt{2\gamma_t}$, and $f_2(\tilde{\gamma}_t) = 2\log\left(\frac{\mu_t^+ \cdot \bar{\mu}_{t-1}^+}{\mu_t^- \cdot \bar{\mu}_{t-1}^-}\right) \cdot \tilde{\gamma}_t$.

**Step 4: Estimating Expected Measure Gap via Measuring Partition** As $\mathcal{V}_0 \backslash \{\mathbf{v}^*\} = \mathcal{V}_1$, we consider the following partition of original coupled conditional KL:

$$\sum_{\lambda \in \{0,1\}} \sum_{i=1}^n \mathbb{E}_{\mathbf{v}_t \sim q(\mathbf{v}_{t|\lambda})}[\mathcal{D}_{\mathrm{KL}}(q(\mathbf{v}_{t-1|\lambda}^i|\mathbf{v}_{t|\lambda})\|q(\mathbf{v}_{t-1|1-\lambda}^i|\mathbf{v}_{t|1-\lambda}))] \tag{18}$$

$$= \underbrace{\mathbb{E}_{\mathbf{v}_t \sim q(\mathbf{v}_{t|0})} \sum_{i=1}^n [\mathcal{D}_{\mathrm{KL}}(q(\mathbf{v}_{t-1|0}^i|\mathbf{v}_{t|0})\|q(\mathbf{v}_{t-1|1}^i|\mathbf{v}_{t|1})) + \mathcal{D}_{\mathrm{KL}}(q(\mathbf{v}_{t-1|1}^i|\mathbf{v}_{t|1})\|q(\mathbf{v}_{t-1|0}^i|\mathbf{v}_{t|0}))]}_{\mathscr{B}_1}$$

$$+ \sum_{i=1}^n \underbrace{\mathbb{E}_{\mathbf{v}_t \sim (q(\mathbf{v}_{t|1}) - q(\mathbf{v}_{t|0}))}[\mathcal{D}_{\mathrm{KL}}(q(\mathbf{v}_{t-1|1}^i|\mathbf{v}_{t|1})\|q(\mathbf{v}_{t-1|0}^i|\mathbf{v}_{t|0}))]}_{\mathscr{B}_2} \tag{19}$$

**For** $\mathscr{B}_1$, we can upper bounded coupled KL terms $\sum_{i=1}^n \mathcal{D}_{\mathrm{KL}}(q(\mathbf{v}_{t-1|0}^i|\mathbf{v}_{t|0})\|q(\mathbf{v}_{t-1|1}^i|\mathbf{v}_{t|1})) + \mathcal{D}_{\mathrm{KL}}(q(\mathbf{v}_{t-1|1}^i|\mathbf{v}_{t|1})\|q(\mathbf{v}_{t-1|0}^i|\mathbf{v}_{t|0}))$ as follows:

**Lemma A.4** (Upper Bounding Coupled Conditional KL)**.** Given adjacent datasets $\mathcal{V}_0 \backslash \{\mathbf{v}^*\} = \mathcal{V}_1$ and a specific DDM, we have

$$\sum_{i=1}^n [\mathcal{D}_{\mathrm{KL}}(q(\mathbf{v}_{t-1|0}^i|\mathbf{v}_{t|0})\|q(\mathbf{v}_{t-1|1}^i|\mathbf{v}_{t|1})) + \mathcal{D}_{\mathrm{KL}}(q(\mathbf{v}_{t-1|1}^i|\mathbf{v}_{t|1})\|q(\mathbf{v}_{t-1|0}^i|\mathbf{v}_{t|0}))] \leq \Delta_t(\mathbf{v}_t) \tag{20}$$

where $\Delta_t(\mathbf{v}_t)$ is defined as

$$\frac{\mathcal{A}_t}{1 + \zeta(\mathcal{V}_1, \mathbf{v}^*, \mathbf{v}_t, t)} \cdot \sum_{i=1}^n \log(1 + \frac{\mathcal{B}_t}{(\bar{R}_{t-1}^2 + 1)\zeta(\mathcal{V}_1^{i|\mathbf{v}^{*i}}, \mathbf{v}^*, \mathbf{v}_t, t) + \bar{R}_{t-1}/\bar{R}_t \cdot \zeta(\mathcal{V}_1, \mathbf{v}^*, \mathbf{v}_t, t) + 1})$$

such that $\zeta(\mathcal{V}, \mathbf{v}^*, \mathbf{v}_t, t) = \sum_{\mathbf{v}_0 \in \mathcal{V}} (\bar{R}_t)^{\bar{\omega}(\mathbf{v}^*, \mathbf{v}_t) - \bar{\omega}(\mathbf{v}_0, \mathbf{v}_t)}$, $\mathcal{A}_t = \mu_t^+ \cdot (\bar{\mu}_{t-1}^+/\bar{\mu}_t^+ - \bar{\mu}_{t-1}^-/\bar{\mu}_t^-)$, and $\mathcal{B}_t = \bar{R}_{t-1}^2 - 1$.

**Measure Partition.** Now, we have demonstrated that the coupled conditional KL can be constrained by $\Delta_t(\mathbf{v}_t)$. On the one hand, the magnitude of $\Delta_t(\mathbf{v}_t)$ heavily relies on the disparity between the distances of $(\mathbf{v}_t, \mathbf{v}^*)$ and $(\mathbf{v}_t, \mathcal{V}_1)$. When $\mathbf{v}_t$ exhibits greater similarity to points in $\mathcal{V}_1$ (i.e., $\mathbb{E}_{\mathbf{v}_0 \in \mathcal{V}_1} \bar{\omega}(\mathbf{v}_t, \mathbf{v}_0) \ll \bar{\omega}(\mathbf{v}_t, \mathbf{v}_0)$), the resulting privacy leakage is reduced. Otherwise, the privacy leakage will be significant. On the other hand, if $\mathbf{v}^*$ is an anomalous point compared to the points in $\mathcal{V}_1$, the probability measure $q(\mathbf{v}_{t|0})$ around $\mathbf{v}^*$ is relatively low, but the privacy leakage remains high. Therefore, in the subsequent analysis, we perform a partition of the probability measure to strike a balance between probability measure and privacy leakage.

We partition probability measure $q(\mathbf{v}_{t|0})$ as follows:

- $q(\mathbf{v}_t \in \mathcal{S}_a) := q(\mathbf{v}_t : (1)\,\mathbf{v}_t$ is diffused from $\mathbf{v}_0 \in \mathcal{V}_0$ s.t. $\bar{\omega}(\mathbf{v}_0, \mathbf{v}^*) \leq \eta_t$ or $(2)\,\bar{\omega}(\mathbf{v}_t, \mathbf{v}^*) \leq \eta_t)$,

- $q(\mathbf{v}_t \in \mathcal{S}_b) := q(\mathbf{v}_t : (1)\,\mathbf{v}_t$ is diffused from $\mathbf{v}_0 \in \mathcal{V}_0$ s.t. $\bar{\omega}(\mathbf{v}_0, \mathbf{v}^*) > \eta_t$ and $(2)\,\exists \mathbf{v}_0' \in \mathcal{V}_0$, s.t. $\bar{\omega}(\mathbf{v}_t, \mathbf{v}_0') \leq \bar{\omega}(\mathbf{v}_t, \mathbf{v}^*) - \eta_t')$,

- $q(\mathbf{v}_t \in \mathcal{S}_c) := q(\mathbf{v}_t : (1)\,\mathbf{v}_t$ is diffused from $\mathbf{v}_0 \in \mathcal{V}_0$ s.t. $\bar{\omega}(\mathbf{v}_0, \mathbf{v}^*) > \eta_t$ and $(2)\,\forall \mathbf{v}_0' \in \mathcal{V}_0$ s.t. $\bar{\omega}(\mathbf{v}_t, \mathbf{v}_0') > \bar{\omega}(\mathbf{v}_t, \mathbf{v}^*) - \eta_t')$.

In this case, $q(\mathbf{v}_t \in \mathcal{S}_a)$ and $q(\mathbf{v}_t \in \mathcal{S}_c)$ represent low-probability events that carry a higher risk of significant privacy disclosure. On the other hand, $q(\mathbf{v}_t \in \mathcal{S}_b)$ corresponds to a significant portion of the probability measure but exhibits minimal privacy leakage.

According to the above measure partition, we introduce the following lemma to further bound $\mathbb{E}_{\mathbf{v}_t \sim q(\mathbf{v}_{t|0})}[\Delta_t(\mathbf{v}_t)]$.

**Lemma A.5.** Given adjacent datasets $\mathcal{V}_0, \mathcal{V}_1$ and a specific DDM design, we have

$$\mathbb{E}_{\mathbf{v}_t \sim q(\mathbf{v}_{t|0})}[\Delta_t(\mathbf{v}_t)] \leq \min\left\{\frac{4N_{(1+c_t^*)\eta_t}(\mathbf{v}^*)}{s}, 1\right\} \cdot \frac{n}{s^{\psi_t}} \tag{21}$$

where $\frac{n}{s^{\psi_t}} = \frac{\mathcal{A}_t}{1+\mathrm{Sim}(\mathcal{V}_1, \mathbf{v}^*, t)} \cdot \sum_{i=1}^n \log(1 + \frac{\mathcal{B}_t}{\bar{R}_{t-1}^2 \cdot \mathrm{Sim}(\mathcal{V}_1^{i|\mathbf{v}^{*i}}, \mathbf{v}^*, t) + \mathrm{Sim}(\mathcal{V}_1, \mathbf{v}^*, t)+1})$, $\mathcal{A}_t = \mu_t^+ \cdot (\bar{\mu}_{t-1}^+/\bar{\mu}_t^+ - \bar{\mu}_{t-1}^-/\bar{\mu}_t^-)$ and $\mathcal{B}_t = \bar{R}_{t-1}^2 - 1$ and $\eta_t \in \{0, 1, ..., n\}$ and $c_t^* \in \{0, \frac{1}{\eta_t}, \frac{2}{\eta_t}, ..., \frac{n-\eta_t}{\eta_t}\}$ satisfy:

$$\eta_t \geq \frac{\log \frac{s - N_{\eta_t}(\mathbf{v}^*)}{N_{(1+c_t^*)\eta_t}(\mathbf{v}^*)}}{\log \frac{1}{n(1-\bar{\mu}_t^+)}} + \left(\frac{\log \frac{s - N_{\eta_t}(\mathbf{v}^*)}{N_{(1+c_t^*)\eta_t}(\mathbf{v}^*)} + \log(\mathcal{A}_t \cdot \mathcal{B}_t \cdot s^{\psi_t}/\bar{R}_{t-1}^2)}{2\log \bar{R}_t}\right)_+ - 2 \tag{22}$$

and for a given $\eta_t$, the corresponding $c_t^*$ is calculated in Algorithm 2.

As shown from the lemma that when $(1 + c_t^*)\eta_t$ is small, we will have more accurate estimate of privacy leakage. Thus, we select the smallest $\eta_t$ that satisfy the above inequality which are monotone functions on both sides and we select $c_t^*$ according to Algorithm 2. Also, we observe that $(n, 0)$ is a natural solution to the above inequalities, therefore, the feasible set of above inequalities is not empty. Further, Eq. (22) is a **weaker condition** compared with Eq. (9) in Main Theorem 1. We further relax this condition in Appendix B to more precisely calculate the privacy bound.

Now that we have the upper bound estimation for $\mathscr{B}_1$, we consider $\mathscr{B}_2$ where the measure around point $\mathbf{v}^*$ can be nearly zero when $\mathbf{v}^*$ is a outlier point compared to $\mathcal{V}_1$.

**For $\mathscr{B}_2$,** differently from previous analysis, we can directly upper bounded expected KL term $\mathbb{E}_{\mathbf{v}_t \sim (q(\mathbf{v}_{t|1}) - q(\mathbf{v}_{t|0}))}[\mathcal{D}_{\mathrm{KL}}(q(\mathbf{v}_{t-1|1}^i|\mathbf{v}_{t|1})\|q(\mathbf{v}_{t-1|0}^i|\mathbf{v}_{t|0}))]$ as follows:

**Lemma A.6.** Given adjacent datasets $\mathcal{V}_0, \mathcal{V}_1$ and a specific DDM design, we have

$$\mathbb{E}_{\mathbf{v}_t \sim (q(\mathbf{v}_{t|1}) - q(\mathbf{v}_{t|0}))}[\mathcal{D}_{\mathrm{KL}}(q(\mathbf{v}_{t-1|1}^i|\mathbf{v}_{t|1})\|q(\mathbf{v}_{t-1|0}^i|\mathbf{v}_{t|0}))] \tag{23}$$

$$\leq \frac{\mathbb{P}(\mathbf{v}_{t|0} \in \Omega)}{s(s+1)} \cdot \left[(1 - \frac{\mu_t^+ \bar{\mu}_{t-1}^+}{\bar{\mu}_t^+}) \cdot (1 - \frac{1}{\bar{R}_{t-1}}) + \frac{\mu_t^+ \bar{\mu}_{t-1}^+}{\bar{\mu}_t^+} \cdot (1 - \frac{\bar{R}_t}{\bar{R}_{t-1}})\right] \tag{24}$$

$$\leq \frac{\varrho_t}{s^2} \cdot \mathbb{P}(\mathbf{v}_{t|0} \in \Omega) \leq \frac{n(1 - \frac{1}{\bar{R}_{t-1}})}{s^2} \tag{25}$$

where $\Omega = \{\mathbf{v}_t | \zeta(\mathcal{V}_1, \mathbf{v}^*, \mathbf{v}_t, t) > s\}$ and $\varrho_t = \left[(1 - \frac{\mu_t^+ \bar{\mu}_{t-1}^+}{\bar{\mu}_t^+}) \cdot (1 - \frac{1}{\bar{R}_{t-1}}) + \frac{\mu_t^+ \bar{\mu}_{t-1}^+}{\bar{\mu}_t^+} \cdot (1 - \frac{\bar{R}_t}{\bar{R}_{t-1}})\right]$.

Summarizing the above results, we reach the final result

$$\delta(\mathcal{V}_0, \mathbf{v}^*) \leq m\left[\underbrace{\sum_{t=T_{\mathrm{rl}}}^T \min\left\{\frac{4N_{(1+c_t^*)\eta_t}(\mathbf{v}^*)}{s}, 1\right\} \cdot \frac{n}{s^{\psi_t}} + \frac{\varrho_t}{s^2}}_{\text{Main Privacy Term}} + \underbrace{\mathcal{O}\left(\sqrt{\gamma_t} + \hat{\gamma}_t\right)}_{\text{Error Term}}\right]/(\epsilon(1 - e^{-\epsilon})) \tag{26}$$

# B   Relaxed Conditions on $\eta_t$ and $c_t^*$

In this section, we further elaborate data-dependent quantities $c_t^*$ and $\eta_t$ and present a weaker conditions that are less restrictive than Eq. 9.

## B.1   Relaxed Condition on $\eta_t$

As described in proof sketch that we partition the measure $q(\mathbf{v}_{t|0})$ into three sets $\mathcal{S}_a$, $\mathcal{S}_b$ and $\mathcal{S}_c$ where we introduced $\eta_t$ and $\eta_t'$ that define the points that are close to $\mathbf{v}_t^*$ within $\eta_t$-ball and the positive margin that whether a specific point are more close to $\mathbf{v}^*$ than points in $\mathcal{V}_1$ by this margin. By balancing the

privacy leakage on the these three sets w.r.t their measure, we obtain the relaxed condition for $\eta_t$. Define $\vartheta'(\eta_t, \eta'_t) = (s - N_{\eta_t}(\mathbf{v}^*))/(N_{\eta'_t}(\mathbf{v}^*))$ and recall that $\varphi_t = \mathcal{A}_t \cdot \mathcal{B}_t \cdot s^{\psi_t}$, we have

$$\eta_t \geq \kappa^* + \left( \frac{\log \vartheta'(\eta_t, (1 + c_t^*)\eta_t)) + \log \varphi_t}{2 \log \frac{\mu_t^+}{\mu_t^-}} \right)_+ - 2 \tag{27}$$

where $\kappa^* = \operatorname{argmin}_{\kappa \in \{0,1,...,n\}} \{ \mathbb{P}(\bar{\omega}(\mathbf{v}_t, \mathbf{v}_0) \geq \kappa) \leq \frac{N_{(1+c_t^*)\eta_t}(\mathbf{v}^*)}{s - N_{\eta_t}(\mathbf{v}^*)} \} = \operatorname{argmin}_{\kappa \in \{0,1,...,n\}} \{ \sum_{j=\kappa}^n \binom{n}{j}(1 - \bar{\mu}_t^+)^j (\bar{\mu}_t^+)^{n-j} \leq \frac{N_{(1+c_t^*)\eta_t}(\mathbf{v}^*)}{s - N_{\eta_t}(\mathbf{v}^*)} \}$. For technical details, we refer readers to the proof of Lemma A.5.

## B.2  Relaxed condition for $c_t^*$

We introduce the data-dependent quantity $c_t^*$ to characterize $q(\mathbf{v}_t \in \mathcal{S}_a)$, which subsequently aids in the privacy balancing process. Based on the definition of $\mathcal{S}_a$, we further examine the measure on $\mathbf{v}_t$ stemming from points both within the $\eta_t$-ball and those external to it. Specifically,

$$
\begin{aligned}
q(\mathbf{v}_t \in \mathcal{S}_a) =& q(\mathbf{v}_t : (1)\, \mathbf{v}_t \text{ is diffused from } \mathbf{v}_0 \in \mathcal{V}_0 \text{ s.t. } \bar{\omega}(\mathbf{v}_0, \mathbf{v}^*) \leq \eta_t \text{ or } (2)\, \bar{\omega}(\mathbf{v}_t, \mathbf{v}^*) \leq \eta_t) \\
\leq & q(\mathbf{v}_t; \mathbf{v}_t \text{ is diffused from } \mathbf{v}_0 \in \mathcal{V}_0 \text{ s.t. } \bar{\omega}(\mathbf{v}_0, \mathbf{v}^*) \leq \eta_t) \\
& + q(\mathbf{v}_t; \mathbf{v}_t \text{ is diffused from } \mathbf{v}_0 \in \mathcal{V}_0 \text{ s.t. } \bar{\omega}(\mathbf{v}_0, \mathbf{v}^*) \in [\eta_t + 1, (1 + c_t^*)\eta_t], \bar{\omega}(\mathbf{v}_t, \mathbf{v}^*) \leq \eta_t) \\
& + q(\mathbf{v}_t; \mathbf{v}_t \text{ is diffused from } \mathbf{v}_0 \in \mathcal{V}_0 \text{ s.t. } \bar{\omega}(\mathbf{v}_0, \mathbf{v}^*) \geq (1 + c_t^*)\eta_t + 1, \bar{\omega}(\mathbf{v}_t, \mathbf{v}^*) \leq \eta_t)
\end{aligned}
\tag{28}
$$
$$\tag{29}$$

To this end, we improve the algorithm ($\texttt{Alg}_{c_t^*}$) with weaker condition than Eq. (9) (RIGHT) to directly calculate parameter $c_t^*$ given specific $\eta_t$ and particular model design. Recall that $\vartheta(\eta_t) = (s - N_{\eta_t}(\mathbf{v}^*))/N_{\eta_t}(\mathbf{v}^*)$.

## B.3  Comparison of Original and Enhanced Conditions for Data-Dependent Quantities

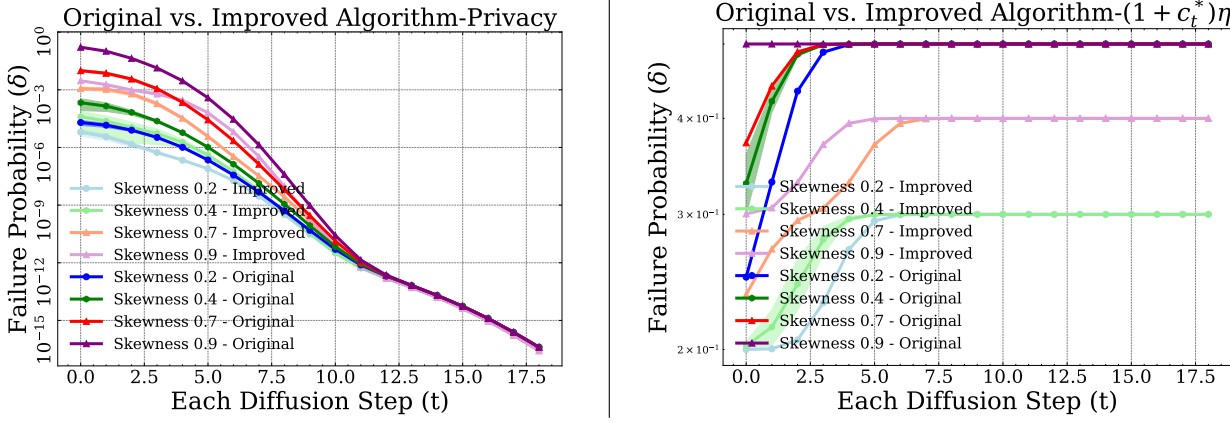

Figure 8: Comparison between two algorithms in each diffusion step (Linear Schedule): **LEFT:** Characterization of Privacy Leakage (Main Privacy Term). **Right:** Characterization of $(1 + c_t^*)\eta_t$. Experimental Setup: Given specific DDM design $k = 5, n = 5, T = 20, \epsilon = 1$ trained on dataset with $s = 30000$ following skewed distribution with various skewness parameter. We consider a fixed $\mathbf{v}^*$ where each column has a non-majority category. Data-dependent quantities are computed at each generation step based on 5 times independent tests.

In this section, we compare original conditions (Eq. (9)) and improved conditions (Eq. (27)). We detail the selection of $(1 + c_t^*)\eta_t$ alongside the privacy leakage observed at each diffusion step for various skewed distributions, as depicted in Fig. 8. Evident from the figure, the lines of a lighter shade, denoting the enhanced conditions, are notably lower than their darker-shaded counterparts. This underscores the enhanced algorithm's marked advantage over the original.

## C  Time Complexity of The Privacy Bound Algorithm for DDMs

In this section, further analyze the total time complexity of our privacy bound algorithm for DDMs. The total time complexity is $O(s \times T \times n \log n)$, where $s$ represents the number of samples for which we aim to assess privacy, $T$ denotes the total diffusion steps, and $n$ is the feature dimension. The most time-intensive part of our algorithm involves selecting the radius (lines 9 in Algorithm 1), where determining the radius coefficient for each sample at each diffusion step for a given $\eta$ requires $O(\log n)$ time through binary search.

## D  Lower Bound on pDP

In this section, we delve into the lower bound of pDP. Specifically, we examine datasets with one or two dimensions ($n = 1, 2$), and our findings illustrate the tightness of privacy bound in relation to dataset size $s$, exhibiting an order of $\mathcal{O}_s(\frac{1}{s})$. Our discussion will mainly focus on two-dimensional case as we only have exchange of dimensional features in the generation process (due to conditional independence property) for $n \geq 2$.

First, we introduce several diffusion-coefficient based quantities:

$$C_{1,t} = \frac{\mu_t^+ \cdot \bar{\mu}_{t-1}^+}{\bar{\mu}_t^+}, C_{2,t} = \frac{\mu_t^- \cdot \bar{\mu}_{t-1}^-}{\bar{\mu}_t^+}, \tilde{C}_{1,t} = \frac{\mu_t^+ \cdot \bar{\mu}_{t-1}^-}{\bar{\mu}_t^-}, \tilde{C}_{2,t} = \frac{\mu_t^- \cdot \bar{\mu}_{t-1}^+}{\bar{\mu}_t^-}. \tag{30}$$

These quantities represent the values of $q(\mathbf{v}_{t-1}^i | \mathbf{v}_t^i, \mathbf{v}_0^i)$ for any $i \in [n]$ across various combinations of $\mathbf{v}_{t-1}^i$, $\mathbf{v}_t^i$, and $\mathbf{v}_0^i$.

**Theorem D.1** (Lower Bound on Inherent pDP Guarantees for DDMs for $n = 2$)**.** Assume the denoising networks are perfectly trained ($\gamma_t$ is negligible). Given a diffusion model architecture design ($\{\alpha_t\}_{t \in [T]}$), there exist adjacent datasets $\mathcal{V}_0, \mathcal{V}_1 = \mathcal{V}_0 \backslash \{\mathbf{v}^*\}$ with feature dimension $n = 2$. For a sufficiently large $s$, the mechanism $\mathcal{M}_0(\cdot; 1)$ will not satisfies $(\epsilon, \delta)$-pDP with respect to $(\mathcal{V}_0, \mathbf{v}^*)$ for

$$\epsilon = \log\left(1 + \frac{\tilde{G}_1 + \tilde{F}_1 \tilde{G}_2 + \dots \tilde{F}_1 \tilde{F}_2 \dots \tilde{F}_{T-2} \tilde{G}_{T-1}}{2 \cdot (1 + \bar{R}_1^2 \cdot \Delta)}\right) \tag{31}$$

and for any

$$\delta < \frac{\tilde{G}_1 + \tilde{F}_1 \tilde{G}_2 + \dots \tilde{F}_1 \tilde{F}_2 \dots \tilde{F}_{T-2} \tilde{G}_{T-1}}{s}. \tag{32}$$

where $\tilde{G}_1 = \frac{\bar{R}_1^2}{s}, \tilde{G}_t = \frac{2(\tilde{C}_{2,t} - C_{2,t})}{\bar{R}_t^2}, \tilde{F}_t = C_{1,t} \cdot (C_{1,t} - \tilde{C}_{1,t})$ and $\tilde{\Delta} = \min_t \frac{C_{2,t} + \tilde{C}_{1,t} + 2\tilde{C}_{1,t} \cdot C_{2,t}}{4}$.

***Proof Sketch of Theorem D.1.*** Consider a worst case adjacent datasets with two categories $\mathcal{V}_0 = \left\{ \underbrace{\begin{bmatrix} 0 \\ 0 \end{bmatrix}, \begin{bmatrix} 0 \\ 0 \end{bmatrix}, \dots, \begin{bmatrix} 0 \\ 0 \end{bmatrix}}_{s-1}, \begin{bmatrix} 1 \\ 1 \end{bmatrix}, \begin{bmatrix} 1 \\ 1 \end{bmatrix} \right\}, \mathcal{V}_1 = \left\{ \underbrace{\begin{bmatrix} 0 \\ 0 \end{bmatrix}, \begin{bmatrix} 0 \\ 0 \end{bmatrix}, \dots, \begin{bmatrix} 0 \\ 0 \end{bmatrix}}_{s-1}, \begin{bmatrix} 1 \\ 1 \end{bmatrix} \right\}$. In this case, $\pi_0^*, \pi_1^*$ can be regarded as a probability vector over four points ($\begin{bmatrix} 0 \\ 0 \end{bmatrix}, \begin{bmatrix} 1 \\ 0 \end{bmatrix}, \begin{bmatrix} 0 \\ 1 \end{bmatrix}, \begin{bmatrix} 1 \\ 1 \end{bmatrix}$), i.e. $\pi_0^*, \pi_1^* \in \mathbb{R}^4$. $i$-th dimension denote the probability of generating $i$-th point respectively.

**Lemma D.1.** Given the same assumption and diffusion model design, Let $e_j, j \in \{1, 2, 3, 4\}$ denote the vector with a 1 in the $j$-th position and 0s elsewhere where the 1 appears in the $j$-th row, we have

$$(\pi_1^* - \pi_0^*)^T e_4 \geq G_1 + F_1 G_2 + \dots + F_1 F_2 \dots F_{T-2} G_{T-1} \tag{33}$$

$$= \Theta_s(\frac{1}{s}). \tag{34}$$

where $F_t$ and $G_t$ are positive diffusion-coefficient based constants

$$F_t = \frac{4(\bar{\mu}_t^+)^4 A + 4(s-1)(\bar{\mu}_t^+)^2(\bar{\mu}_t^-)^2 B + (s-1)^2(\bar{\mu}_t^-)^4 C}{(s+1)^2[2(\bar{\mu}_t^+)^2 + (s-1)(\bar{\mu}_t^-)^2]^2}, t \in [T], \tag{35}$$

$$G_1 = \frac{\bar{R}_1^2(s-1)}{(1+(s-1)\bar{R}_1^2)(2+(s-1)\bar{R}_1^2)}, \tag{36}$$

$$G_t = \frac{[(s-1)(\tilde{C}_{2,t} - C_{2,t})(\bar{\mu}_t^+)^2(\bar{\mu}_t^-)^2][4\tilde{C}_{2,t}(\bar{\mu}_t^-)^4 + 3(s-1)(C_{2,t} + \tilde{C}_{2,t})(\bar{\mu}_t^+)^2(\bar{\mu}_t^-)^2 + 2(s-1)^2 C_{2,t}(\bar{\mu}_t^+)^4]}{[(s-1)(\bar{\mu}_t^+)^2 + 2(\bar{\mu}_t^-)^2]^2[(s-1)(\bar{\mu}_t^+)^2 + (\bar{\mu}_t^-)^2]^2}, t \in \{2, ..., T\}. \tag{37}$$

and

$$A = (s+1)^2 C_{1,t}^2 - [(s-1)\tilde{C}_{1,t} + 2C_{1,t}][(s-1)C_{1,t} + 2\tilde{C}_{2,t}], \tag{38}$$

$$B = (s+1)^2 \tilde{C}_{1,t} \cdot C_{1,t} - [(s-1)\tilde{C}_{1,t} + 2C_{1,t}][(s-1)C_{1,t} + 2\tilde{C}_{2,t}], \tag{39}$$

$$C = (s+1)^2 \tilde{C}_{1,t}^2 - [(s-1)\tilde{C}_{1,t} + 2C_{1,t}][(s-1)C_{1,t} + 2\tilde{C}_{2,t}]. \tag{40}$$

Lemma D.1 demonstrates that when datasets $\mathcal{V}_0$ and $\mathcal{V}_1$ differ by a single sample relative to a dataset size of $s$, the output measure of the diffusion model will exhibit a difference of at least $\Omega_s(\frac{1}{s})$ in its predictions.

**Lemma D.2.** Given the same assumption and diffusion model design, we have

$$\pi_0^{*T} e_4 \leq \frac{1}{s} + \frac{R_1^2}{(R_1^2 + s)} \cdot \Delta = \Theta_s(\frac{1}{s}) \tag{41}$$

where we have

$$\Delta = \min_t \frac{1}{4}\Big[\frac{C_{2,t} \cdot s \cdot \bar{R}_t^2}{s \cdot \bar{R}_t^2 + 1} + \frac{\tilde{C}_{2,t}}{s\bar{R}_t^2 + 1} + \frac{\tilde{C}_{1,t} \cdot s}{s + \bar{R}_t^2} + \frac{C_{1,t} \cdot \bar{R}_t^2}{s + \bar{R}_t^2} + 2\big(\frac{\tilde{C}_{1,t} \cdot (s-1)}{s} + \frac{C_{1,t} \cdot}{s}\big)\big(\frac{C_{2,t} \cdot (s-1)}{s} + \frac{\tilde{C}_{2,t}}{s}\big)\Big]. \tag{42}$$

Lemma D.2 suggests that if a specific category constitutes approximately $\frac{1}{s}$ of the original dataset, then, the diffusion model will produce data from this category at a proportion of $\Theta_s(\frac{1}{s})$.

Therefore, consider the measurable set $\mathcal{B} = \{[1,1]^T\}$, from Lemma D.1 and Lemma D.2, for sufficient large $s$, for any $c \in \mathbb{N}_+$, select

$$\epsilon_0 = \log\Big(1 + \frac{\tilde{G}_1 + \tilde{F}_1 \tilde{G}_2 + \dots \tilde{F}_1 \tilde{F}_2 \dots \tilde{F}_{T-2} \tilde{G}_{T-1}}{c \cdot (1 + \bar{R}_1^2 \cdot \tilde{\Delta})}\Big) \tag{43}$$

such that

$$\delta \geq \mathcal{P}(\mathcal{M}_0(\mathcal{V}_0; 1) \in \mathcal{B}) - e^{\epsilon_0}\mathcal{P}(\mathcal{M}_0(\mathcal{V}_1; 1) \in \mathcal{B}) \tag{44}$$

$$= (\mathcal{P}(\mathcal{M}_0(\mathcal{V}_0; 1) \in \mathcal{B}) - \mathcal{P}(\mathcal{M}_0(\mathcal{V}_1; 1) \in \mathcal{B})) - (e^{\epsilon_0} - 1)\mathcal{P}(\mathcal{M}_0(\mathcal{V}_1; 1) \in \mathcal{B}) \tag{45}$$

$$\geq \frac{c-1}{c} \cdot \frac{\tilde{G}_1 + \tilde{F}_1 \tilde{G}_2 + \dots \tilde{F}_1 \tilde{F}_2 \dots \tilde{F}_{T-2} \tilde{G}_{T-1}}{s} = \mathcal{O}_s(\frac{1}{s}) \tag{46}$$

where $\tilde{G}_1 = \frac{\bar{R}_1^2}{s}, \tilde{G}_t = \frac{2(\tilde{C}_{2,t} - C_{2,t})}{\bar{R}_t^2}, \tilde{F}_t = C_{1,t} \cdot (C_{1,t} - \tilde{C}_{1,t})$ and $\tilde{\Delta} = \min_t \frac{C_{2,t} + \tilde{C}_{1,t} + 2\tilde{C}_{1,t} \cdot C_{2,t}}{4}$. Let $c = 2$, we obtain the results. $\qquad \square$

# E    DP Guarantees of DDMs from pDP

In Sec. 2, we highlight the distinction between pDP and DP. The former prioritizes protecting the privacy of a specific point with respect to a (large) dataset prior to training, proving beneficial for preprocessing datasets

with privacy assurances. The latter, however, addresses the privacy implications of publishing a model. In this section, we will discuss how to derive a general DP guarantee for a DDM from pDP by examining the worst-case adjacent datasets $(\mathcal{V}_0 \backslash \{\mathbf{v}^*\} = \mathcal{V}_1)$. As mentioned in Sec. 3, the key for bounding the coupled KL divergence can be further focusing on balancing the privacy budget over three measure partitioned sets $\mathcal{S}_a, \mathcal{S}_b$ and $\mathcal{S}_c$ i.e.

$$q(\mathbf{v}_t \in \mathcal{S}_a) \cdot \frac{n}{s^{\psi_t}} + q(\mathbf{v}_t \in \mathcal{S}_b) \cdot \frac{n \cdot \mathcal{A}_t \cdot \mathcal{B}_t}{\bar{R}_{t-1}^2 \cdot \bar{R}_t^{2\eta_t'}} + q(\mathbf{v}_t \in \mathcal{S}_c) \cdot \frac{n}{s^{\psi_t}}. \tag{47}$$

We first consider the worst case for data-dependent quantity $\psi_t$.

$$\frac{n}{s^{\psi_t}} = \frac{\mathcal{A}_t}{1 + \mathrm{Sim}(\mathcal{V}_1, \mathbf{v}^*, t)} \cdot \sum_{i=1}^{n} \log(1 + \frac{\mathcal{B}_t}{\bar{R}_{t-1}^2 \cdot \mathrm{Sim}(\mathcal{V}_1^{i|\mathbf{v}^{*i}}, \mathbf{v}^*, t) + \mathrm{Sim}(\mathcal{V}_1, \mathbf{v}^*, t) + 1}) \tag{48}$$

where $\mathcal{A}_t = \mu_t^+ \cdot (\bar{\mu}_{t-1}^+ / \bar{\mu}_t^+ - \bar{\mu}_{t-1}^- / \bar{\mu}_t^-)$, and $\mathcal{B}_t = \bar{R}_{t-1}^2 - 1$.

From the definition of similarity (Sim), we observe that the worst scenario arises when $\mathbf{v}_t^*$ is an outlier within the dataset where, for each feature dimension, only one point in $\mathcal{V}_1$ shares the same overlap on that particular dimension (i.e. $|\mathcal{V}_1^{i|\mathbf{v}^{*i}}| = 1$). From this, we can further prove that the similarity $\mathrm{Sim}(\mathcal{V}_1, \mathbf{v}^*, t) \geq (\bar{R}_t)^{-n}(s-n) + (\bar{R}_t)^{-(n-1)}, \mathrm{Sim}(\mathcal{V}_1^{i|\mathbf{v}^{*i}}, \mathbf{v}^*, t) \geq (\bar{R}_t)^{-(n-1)}$. Therefore, $\frac{1}{s^{\psi_t}}$ is upper bounded by

$$\frac{1}{s^{\psi_t}} \leq \underbrace{\frac{\mathcal{A}_t}{1 + (\bar{R}_t)^{-n}(s-n) + (\bar{R}_t)^{1-n}} \cdot \log(1 + \frac{\mathcal{B}_t}{\bar{R}_{t-1}^2 \cdot (\bar{R}_t)^{1-n} + (\bar{R}_t)^{-n}(s-n) + (\bar{R}_t)^{1-n} + 1})}_{\text{denote as } \frac{1}{s^{\Psi_t}}} \tag{49}$$

where the R.H.S $\frac{1}{s^{\Psi_t}}$ is independent from dataset properties.

**Discussion on Dataset Independent Quantity $\Psi_t$.** In the generation process, as we progressively transit from noisy regime (relatively large $t$) to noise-free regime (relatively small $t$), we have $\bar{R}_t$ monotonically increases from 1 to $\frac{1+(k-1)\alpha_1}{1-\alpha_1}$ and $\Psi_t$ evolves from $\mathcal{O}_s(\frac{1}{s^2})$ to $\mathcal{O}_s(1)$. This aligns with the behavior of data-dependent quantity $\psi_t$.

Next, we consider the privacy balancing radius $\eta_t$ and $c_t^*$. Similar to the discussion in $\psi_t$, we consider the worst case when $\mathbf{v}_t^*$ stands as an anomaly in the dataset such that for every feature dimension, there is a single point in $\mathcal{V}_1$ that coincides on that specific dimension. Under this case, the conditions for $\eta_t$ and $c_t^*$ will be

$$\eta_t \geq \kappa^* + \left( \frac{\log h(\eta_t) + \log(\mathcal{A}_t \cdot \mathcal{B}_t \cdot \Psi_t)}{2 \log \frac{\mu_t^+}{\mu_t^-}} \right)_+ - 2, c_t^* \geq \frac{\frac{1}{\eta_t} \log h((1+c_t^*)\eta_t) + \frac{3}{2}}{\log \frac{1}{\mu_t^-} - 1}. \tag{50}$$

where $\kappa^* = \mathrm{argmin}_{\kappa \in \{0,1,\ldots,n\}} \{ \mathbb{P}(\bar{\omega}(\mathbf{v}_t, \mathbf{v}_0) \geq \kappa) \leq \frac{N_{(1+c_t^*)\eta_t}(\mathbf{v}^*)}{s - N_{\eta_t}(\mathbf{v}^*)} \} = \mathrm{argmin}_{\kappa \in \{0,1,\ldots,n\}} \{ \sum_{j=\kappa}^{n} \binom{n}{j}(1 - \bar{\mu}_t^+)^j (\bar{\mu}_t^+)^{n-j} \leq \frac{N_{(1+c_t^*)\eta_t}(\mathbf{v}^*)}{s - N_{\eta_t}(\mathbf{v}^*)} \}$ and generalized $h(\eta) = s \cdot \mathbb{1}_{\{\eta < n+1\}} + \frac{s}{n} \cdot \mathbb{1}_{\{\eta = n-1\}} + (-\infty) \cdot \mathbb{1}_{\{\eta = n\}}$.

**Discussion on Dataset Independent Quantity $\eta_t, c_t^*$.** As depicted in Eq. (50), the conditions for $\eta_t$ and $c_t^*$ are **data-independent**. As $t$ diminishes from $T$ (with $\bar{\alpha}_t \to 0$ representing the noisy regime) to 1 (where $\bar{\alpha}_t \to 1$ indicates the noise-free regime), the value of $(1 + c_t^*)\eta_t$ transitions from $n$ to 0, consistent with those observed under data-dependent conditions.

Synthesizing the aforementioned results, we put forth the theorem detailing the **inherent differential privacy guarantees of DDMs**.

**Theorem E.1** (**Inherent DP Guarantee for DDMs**)**.** *Given any adjacent datasets $\mathcal{V}_0, \mathcal{V}_1$. Assume the denoising networks trained on $\mathcal{V}_0$ and $\mathcal{V}_1$ satisfy Assumption 1 and Assumption 2. Given a specific time step $T_{rl}$, the mechanism $\mathcal{M}_{T_{rl}}(\cdot; m)$ satisfies $(\boldsymbol{\epsilon}, \boldsymbol{\delta})$-**differential privacy** such that*

$$\delta(\mathcal{V}_0, \mathbf{v}^*) \leq m \left[ \underbrace{\sum_{t=T_{rl}}^{T} \min\left\{ \frac{4 N_{(1+\tilde{c}_t^*)\tilde{\eta}_t}(\mathbf{v}^*)}{s}, 1 \right\} \cdot \frac{n}{s^{\Psi_t}} + \frac{n \varrho_t}{s^2}}_{\textit{Main Privacy Term}} + \underbrace{\mathcal{O}\left( \sqrt{\gamma_t} + \tilde{\gamma}_t \right)}_{\textit{Error Term}} \right] / (\epsilon(1 - e^{-\epsilon})) \tag{51}$$

where $\Psi_t, \tilde{\eta}_t, \tilde{c}_t^*, \varrho_t$ are **diffusion coefficients-based quantities** such that $(\tilde{\eta}_t, \tilde{c}_t^*)$ satisfy Eq. (50), $\Psi_t$ satisfies Eq. (49) and $\varrho_t = (1 - \mu_t^+ \bar{\mu}_{t-1}^+ / \bar{\mu}_t^+) \cdot (1 - 1/\bar{R}_{t-1}) + \mu_t^+ \bar{\mu}_{t-1}^+ / \bar{\mu}_t^+ \cdot (1 - \bar{R}_t / \bar{R}_{t-1})$.

# F   Details on Examples

In this section, we provide detailed analysis on the examples presented in Sec. 3.

## F.1   Analysis on $\frac{1}{s^{\psi_t}}$.

**Proposition F.1.** *Given a skewed distribution with parameter $p$. Let $\mathbf{v}^*$ be a non-majority point and $\mathcal{V}_0 \backslash \{\mathbf{v}^*\}$ sampled from the distribution. Define $\tau_t = \frac{1-p}{k-1} + \frac{\bar{\mu}_t^-}{\bar{\mu}_t^+}(1 - \frac{1-p}{k-1})$. When $s = \omega_n(\frac{\log n}{\tau_t^{2n}})$, we have with high probability*

$$\frac{1}{s^{\psi_t - 2}} \to \frac{(\overline{\alpha}_{t-1} - \overline{\alpha}_t)/(k\bar{\mu}_t^+ \bar{\mu}_t^-)}{\bar{R}_{t-1}^2 \cdot \tau_t^{2n-1} \cdot \frac{1-p}{k-1} + \tau_t^{2n}} \tag{52}$$

***Proof of Proposition F.1.*** Recall the definition of $\frac{1}{s^{\psi_t}}$,

$$\frac{1}{s^{\psi_t}} = \frac{1}{n} \cdot \frac{\mathcal{A}_t}{1 + \mathrm{Sim}(\mathcal{V}_1, \mathbf{v}^*, t)} \cdot \sum_{i=1}^n \log\left(1 + \frac{\mathcal{B}_t}{\bar{R}_{t-1}^2 \cdot \mathrm{Sim}(\mathcal{V}_1^{i|\mathbf{v}^{*i}}, \mathbf{v}^*, t) + \mathrm{Sim}(\mathcal{V}_1, \mathbf{v}^*, t) + 1}\right). \tag{53}$$

where $\mathcal{A}_t = \mu_t^+ \cdot (\bar{\mu}_{t-1}^+ / \bar{\mu}_t^+ - \bar{\mu}_{t-1}^- / \bar{\mu}_t^-)$ and $\mathcal{B}_t = \bar{R}_{t-1}^2 - 1$.

Given $\mathbf{v}^* \in \mathcal{V}$ as an non-majority point, we sample $\mathbf{v} \in \mathcal{V}_1$ from skewed distribution with parameter $p$. Thus, we have $\bar{\omega}(\mathbf{v}^*, \mathbf{v})$ follows binomial $\mathcal{B}(n, 1 - \frac{1-p}{k-1})$ distribution. Therefore, $\mathbb{E}\left((\frac{\mu_t^+}{\mu_t^-})^{-\bar{\omega}(\mathbf{v}^*, \mathbf{v})}\right) = \sum_{i=0}^n (\frac{\mu_t^+}{\mu_t^-})^{-i}\binom{n}{i}(\frac{1-p}{k-1})^{n-i}(1 - (\frac{1-p}{k-1}))^i = (\frac{1-p}{k-1} + \frac{\mu_t^-}{\mu_t^+}(1 - \frac{1-p}{k-1}))^n$, and thus $\mathbb{E}(\mathrm{Sim}(\mathcal{V}_1, \tilde{\mathbf{v}}, t)) = |\mathcal{V}_1| \cdot \mathbb{E}(\sum_{\mathbf{v} \in \mathcal{V}_1}(\frac{\mu_t^+}{\mu_t^-})^{-\bar{\omega}(\mathbf{v}, \tilde{\mathbf{v}})}) = |\mathcal{V}_1| \cdot (\frac{1-p}{k-1} + \frac{\mu_t^-}{\mu_t^+}(1 - \frac{1-p}{k-1}))^n = |\mathcal{V}_1| \cdot (\frac{1-\overline{\alpha}_t}{1+(k-1)\overline{\alpha}_t}(1 - \frac{1-p}{k-1}) + \frac{1-p}{k-1})^n$. Let $E_t := (\frac{1-p}{k-1} + \frac{\mu_t^-}{\mu_t^+}(1 - \frac{1-p}{k-1}))^n = (\frac{1-\overline{\alpha}_t}{1+(k-1)\overline{\alpha}_t}(1 - \frac{1-p}{k-1}) + \frac{1-p}{k-1})^n$.

Since $\max(\frac{\mu_t^+}{\mu_t^-})^{-\bar{\omega}(\mathbf{v}^*, \mathbf{v})} - \min(\frac{\mu_t^+}{\mu_t^-})^{-\bar{\omega}(\mathbf{v}^*, \mathbf{v})} \leq 1$, from concentration inequality, we have for any small $\epsilon(\epsilon \ll E_t)$,

$$\mathbb{P}(|\frac{1}{s} \cdot \mathrm{Sim}(\mathcal{V}_1, \mathbf{v}^*, t) - \frac{1}{s} \cdot \mathbb{E}\mathrm{Sim}(\mathcal{V}_1, \mathbf{v}^*, t)| \geq \epsilon) \leq 2\exp(-2\epsilon^2 s) \tag{54}$$

Let $\epsilon = \epsilon' * E_t$. Thus, with high probability at least $1 - 2\exp(-2(\epsilon')^2 E_t^2 s)$, we have

$$\frac{1}{s} \cdot \mathrm{Sim}(\mathcal{V}_1, \mathbf{v}^*, t) \in [E_t - \epsilon, E_t + \epsilon] = [E_t(1 - \epsilon'), E_t(1 + \epsilon')] \tag{55}$$

Now, we consider $\mathrm{Sim}(\mathcal{V}_1^{i|\mathbf{v}^{*i}}, \mathbf{v}^*, t)$, let $X_1^i, X_2^i, ..., X_s^i$ be Bernoulli random variables such that

$$X_j^i = \begin{cases} 1, & \text{if } j\text{-th sample } \mathbf{v}^{(j)} \in \mathcal{V}_1 \text{ such that } (\mathbf{v}^{(j)})^i = \mathbf{v}^{*i}. \\ 0, & \text{otherwise.} \end{cases} \tag{56}$$

From Hoeffding inequality, we obtain

$$\mathbb{P}(|\frac{1}{s}\sum_{j=1}^s X_j^i - \frac{1}{s}\mathbb{E}[\sum_{j=1}^s X_j^i]| \geq \epsilon_1^i) \leq 2\exp(-2(\epsilon_1^i)^2 s) \tag{57}$$

where $\mathbb{E}[\frac{1}{s}\sum_{j=1}^s X_j^i] = \frac{1-p}{k-1}$. Let $\epsilon_1^i := \frac{1-p}{k-1} \cdot \tilde{\epsilon}_1^i$. Therefore, we have with high probability $1 - 2\exp(-2(\frac{1-p}{k-1})^2 \cdot (\tilde{\epsilon}_1^i)^2 \cdot s)$

$$\sum_{j=1}^s X_j^i \in [\frac{1-p}{k-1}(1 - \tilde{\epsilon}_1^i)s, \frac{1-p}{k-1}(1 + \tilde{\epsilon}_1^i)s] \tag{58}$$

Let $N_i = |\mathcal{V}_1^{i|\mathbf{v}^{*i}}|$. Let $Y_1^i, Y_2^i, ..., Y_{N_i}^i$ denote the features of points in $\mathcal{V}_1^{i|\mathbf{v}^{*i}}$ on dimensions other than $i$-th dimension, which can be viewed as random variables sampling from categorical distributions with $n-1$ dimensions. Recall that $\bar{\omega}_{-i}(\mathbf{v}, \mathbf{v}') = \bar{\omega}(\mathbf{v}, \mathbf{v}') - \mathbb{1}_{\mathbf{v}^i \neq \mathbf{v}^{*i}}$. Hence

$$\text{Sim}(\mathcal{V}_1^{i|\mathbf{v}^{*i}}, \mathbf{v}^*, t) = \sum_{j=1}^{N_i} (\frac{\mu_t^+}{\mu_t^-})^{-\bar{\omega}_{-i}(Y_j^i, \mathbf{v}^*)} \tag{59}$$

Similar to the derivation of $\text{Sim}(\mathcal{V}_1, \mathbf{v}^*, t)$, we have

$$\mathbb{E}\left((\frac{\mu_t^+}{\mu_t^-})^{-\bar{\omega}_{-i}(Y_j^i, \mathbf{v}^*)}\right) = \sum_{i=0}^{n-1} (\frac{\mu_t^+}{\mu_t^-})^{-i} \binom{n-1}{i} (\frac{1-p}{k-1})^{n-1-i} (1-(\frac{1-p}{k-1}))^i \tag{60}$$

$$= (\frac{1-p}{k-1} + \frac{\mu_t^-}{\mu_t^+}(1 - \frac{1-p}{k-1}))^{n-1} =: E_t^i \tag{61}$$

Applying Hoeffding inequality, we have

$$\mathbb{P}(|\frac{1}{N_i}\text{Sim}(\mathcal{V}_1^{i|\mathbf{v}^{*i}}, \mathbf{v}^*, t) - (\frac{1-p}{k-1} + \frac{\mu_t^-}{\mu_t^+}(1 - \frac{1-p}{k-1}))^{n-1}| \geq \epsilon_2^i) \overset{(i)}{\leq} 2\exp(-2(\epsilon_2^i)^2 \cdot (\frac{1-p}{k-1} - \epsilon_2^i)s) \tag{62}$$

where $(i)$ is because $\max(\frac{\mu_t^+}{\mu_t^-})^{-\bar{\omega}_{-i}(Y_j^i, \mathbf{v}^*)} - (\frac{\mu_t^+}{\mu_t^-})^{-\bar{\omega}_{-i}(Y_j^i, \mathbf{v}^*)} \leq 1$ and the concentration property of $N_i$. Let $\epsilon_2^i := (\frac{1-p}{k-1} + \frac{\mu_t^-}{\mu_t^+}(1 - \frac{1-p}{k-1}))^{n-1} \cdot \tilde{\epsilon}_2^i$. Summarizing the above, with high probability $(1 - 2\exp(-2(E_t^i)^2 \cdot (\tilde{\epsilon}_2^i)^2 \cdot s))(1 - 2\exp(-2(\frac{1-p}{k-1})^2 \cdot (\tilde{\epsilon}_1^i)^2 \cdot s))$,

$$\frac{1}{s} \cdot \text{Sim}(\mathcal{V}_1^{i|\mathbf{v}^{*i}}, \mathbf{v}^*, t) \in \left[E_t^i \cdot \frac{1-p}{k-1} \cdot (1 - \tilde{\epsilon}_2^i)(1 - \tilde{\epsilon}_1^i), E_t^i \cdot \frac{1-p}{k-1} \cdot (1 + \tilde{\epsilon}_2^i)(1 + \tilde{\epsilon}_1^i)\right] \tag{63}$$

Therefore, in order to let $\text{Sim}(\mathcal{V}_1^{i|\mathbf{v}^{*i}}, \mathbf{v}^*, t), i \in [n]$ and $\text{Sim}(\mathcal{V}_1, \mathbf{v}^*, t)$ to concentrate, we require as $\tilde{\epsilon}_1^i, \tilde{\epsilon}_2^i, \epsilon' \to 0$,

$$\rho := (1 - 2\exp(-2(\epsilon')^2 E_t^2 s)) \prod_{i=1}^{n} [(1 - 2\exp(-2(E_t^i)^2(\tilde{\epsilon}_2^i)^2 s))(1 - 2\exp(-2(\frac{1-p}{k-1})^2(\tilde{\epsilon}_1^i)^2 s))] \to 1. \tag{64}$$

Thus, $\exp(-2(E_t^i)^2(\tilde{\epsilon}_2^i)^2 s) = o_s(\frac{1}{n}), i \in [n]$ and $\exp(-2(\epsilon')^2 E_t^2 s) = o_s(1)$. Define $\tau_t := \frac{1-p}{k-1} + \frac{\mu_t^-}{\mu_t^+}(1 - \frac{1-p}{k-1})$. From above, we obtain the condition $s = \omega_n(\frac{\log n}{\tau_t^{2n}})$. Here, we list a sufficient condition for $n$ to satisfy the constraint: $n$ can be chosen as $n = \frac{1-a}{2} \cdot \frac{\log s}{\log \frac{1}{\tau_t}}$ for any positive $a < 1$.

For $\frac{1}{s^{\psi_t}} = \frac{1}{n} \sum_{i=1}^{n} \frac{\mathcal{A}_t}{1 + \text{Sim}(\mathcal{V}_1, \mathbf{v}^*, t)} \cdot \log\left(1 + \frac{\mathcal{B}_t}{\bar{R}_{t-1}^2 \cdot \text{Sim}(\mathcal{V}_1^{i|\mathbf{v}^{*i}}, \mathbf{v}^*, t) + \text{Sim}(\mathcal{V}_1, \mathbf{v}^*, t) + 1}\right)$, when $s = \omega_n(\frac{\log n}{\tau_t^{2n}})$, with high probability $\rho$, we have

$$\frac{1}{s^{\psi_t - 2}} \to \frac{\mathcal{A}_t \cdot \mathcal{B}_t}{\bar{R}_{t-1}^2 \cdot (\frac{\bar{\mu}_t^-}{\bar{\mu}_t^+}(1 - \frac{1-p}{k-1}) + \frac{1-p}{k-1})^{2n-1} \cdot \frac{1-p}{k-1} + (\frac{\bar{\mu}_t^-}{\bar{\mu}_t^+}(1 - \frac{1-p}{k-1}) + \frac{1-p}{k-1})^{2n}} \tag{65}$$

From the analysis above, we note that as the skewness of the distribution intensifies, the above term exhibits a monotonic increase with respect to $p$. This implies that the average distance between points in $\mathcal{V}_1$ and $\mathbf{v}^*$ grows, leading to a heightened sensitivity at point $\mathbf{v}^*$. Consequently, the privacy bound increases. $\square$

**Note:** From the aforementioned derivation, taking the logarithm of both sides implies that, since $s = \omega_n(\frac{\log n}{\tau_t^{2n}})$, with high probability,

$$\psi_t - 2 \to \log_s\left(\frac{\bar{R}_{t-1}^2 \cdot (\frac{\bar{\mu}_t^-}{\bar{\mu}_t^+}(1 - \frac{1-p}{k-1}) + \frac{1-p}{k-1})^{2n-1} \cdot \frac{1-p}{k-1} + (\frac{\bar{\mu}_t^-}{\bar{\mu}_t^+}(1 - \frac{1-p}{k-1}) + \frac{1-p}{k-1})^{2n}}{\mathcal{A}_t \cdot \mathcal{B}_t}\right) \to 0. \tag{66}$$

Thus, $\psi_t \to 2$ as $s \to \infty$.

### F.2 Analysis on $\eta_t, c_t^*$.

**Proposition F.2.** *Given a skewed distribution with parameter $p$. Let $\mathbf{v}^*$ be a non-majority point and $\mathcal{V}_0 \backslash \{\mathbf{v}^*\}$ sampled from the distribution. Define $\tau_t = \frac{1-p}{k-1} + \frac{\bar{\mu}_t^-}{\bar{\mu}_t^+}(1 - \frac{1-p}{k-1})$. When $s = \omega_n([\frac{1}{c}(1 - \frac{1-p}{k-1})]^{-2cn}(\frac{1-p}{k-1})^{-2(1-c)n}, \frac{\log n}{\tau_t^{2n}})$ for some $c \in (\frac{1-p}{k-2-p}, 1)$, we have with high probability,*

$$\eta_t = \underset{\eta \in \{\lceil cn \rceil, ..., n\}}{\arg\min} \left\{ \eta \Big| \eta - \frac{\log(\frac{1}{p_{in}(\eta)} - 1)}{\log \frac{1}{n(1-\bar{\mu}_t^+)}} + \max\left\{ \frac{\log(\frac{1}{p_{in}(\eta)} - 1)}{2\log \bar{R}_t} + \mathcal{C}_t, 0 \right\} - 2 \geq 0 \right\} \tag{67}$$

$$c_t^* = \underset{c_t^* \in \{0, \frac{1}{\eta_t}, ..., \frac{n-\eta_t}{\eta_t}\}}{\arg\min} \left\{ c \Big| c - \frac{\frac{1}{\eta_t}\log(\frac{1}{p_{in}((1+c)\eta_t)}) - 1) + \log 2e}{\log \frac{1}{\mu_t^-} - 1} \geq 0 \right\} \tag{68}$$

*where $p_{in}(\cdot)$ is the CDF of Binomial distribution with parameter $1 - \frac{1-p}{k-1}$ and $\mathcal{C}_t = \frac{\log(\mathcal{A}_t \cdot \mathcal{B}_t)}{2\log \bar{R}_t} + \frac{\log s}{\log \bar{R}_t}$, $\mathcal{A}_t = \mu_t^+ \cdot (\bar{\mu}_{t-1}^+/\bar{\mu}_t^+ - \bar{\mu}_{t-1}^-/\bar{\mu}_t^-)$ and $\mathcal{B}_t = \bar{R}_{t-1}^2 - 1$.*

***Proof of Proposition F.2.*** Given $\mathbf{v}^* \in \mathcal{V}$ as an non-majority point, and $\mathbf{v} \in \mathcal{V}_1$ are sampled from skewed distribution with parameter $p$. Consider specific $\eta_t, c_t^*$, the probability of failing into the $\eta_t$-ball of $\mathbf{v}^*$ is $\mathbb{P}(\mathbf{v}; \bar{\omega}(\mathbf{v}, \mathbf{v}^*) \leq \eta_t) = \sum_{i=0}^{\eta_t} \binom{n}{i}(\frac{1-p}{k-1})^{n-i}(1 - \frac{1-p}{k-1})^i \leq \min\{(\frac{1-p}{k-1})^{n-\eta_t}(en)^{\eta_t}, 1\}$.

Now consider the following inequality:

$$\eta_t \geq \frac{\log \vartheta(\eta_t)}{\log \frac{1}{n(1-\bar{\mu}_t^+)}} + \max\left\{ \frac{\log \vartheta(\eta_t) + \log \varphi_t}{2\log \frac{\bar{\mu}_t^+}{\bar{\mu}_t^-}} - 2, -2 \right\}, c_t^* \geq \frac{\frac{1}{\eta_t}\log \vartheta((1+c_t^*)\eta_t) + \log 2e}{\log \frac{1}{\mu_t^-} - 1}. \tag{69}$$

where $\vartheta(\eta_t) = (s - N_{\eta_t}(\mathbf{v}^*))/N_{\eta_t}(\mathbf{v}^*)$, $\varphi_t = \mathcal{A}_t \cdot \mathcal{B}_t \cdot s^{\psi_t}$.

To begin with, first consider $\vartheta(\eta_t) = \frac{1}{\frac{N_{\eta_t}(\mathbf{v}^*)}{s}} - 1$. Let $X_{1,\eta_t}, X_{2,\eta_t}, ..., X_{s,\eta_t}$ be the indicator random variables of whether the point in $\mathcal{V}_1$ fall in the $\eta_t$-ball of $\mathbf{v}^*$, i.e.

$$X_{i,\eta_t} = \begin{cases} 1, & \text{with } \mathbb{P}(\mathbf{v}; \bar{\omega}(\mathbf{v} \text{ follow skewed distribution}; \bar{\omega}(\mathbf{v}, \mathbf{v}^*) \leq \eta_t)) \\ 0, & \text{with } 1 - \mathbb{P}(\mathbf{v}; \bar{\omega}(\mathbf{v} \text{ follow skewed distribution}; \bar{\omega}(\mathbf{v}, \mathbf{v}^*) \leq \eta_t)) \end{cases} \tag{70}$$

Define $p_{\text{in}}(\eta_t) := \mathbb{P}(\mathbf{v}; \bar{\omega}(\mathbf{v} \text{ follow skewed distribution}; \bar{\omega}(\mathbf{v}, \mathbf{v}^*) \leq \eta_t))$. Since $X_{i,\eta_t}$ are Bernoulli random variables, from concentration inequality, with high probability $1 - 2\exp(-2(\epsilon')^2 p_{\text{in}}(\eta_t)^2 s)$,

$$\frac{1}{s} N_{\eta_t}(\mathbf{v}^*) \in [p_{\text{in}}(\eta_t)(1 - \epsilon'), p_{\text{in}}(\eta_t)(1 + \epsilon')] \tag{71}$$

Let $\epsilon = \epsilon' \cdot p_{\text{in}}(\eta_t)$. Since $\frac{\epsilon}{p_{\text{in}}(\eta_t) - \epsilon} > \frac{\epsilon}{p_{\text{in}}(\eta_t) + \epsilon}$, define $\epsilon_1 = \frac{\epsilon}{p_{\text{in}}(\eta_t) - \epsilon}$, we have with high probability $1 - 2\exp(-2(\epsilon')^2 p_{\text{in}}(\eta_t)^2 s)$

$$\vartheta(\eta_t) \in \left[ \frac{1}{p_{\text{in}}(\eta_t)} - 1 - \epsilon_1, \frac{1}{p_{\text{in}}(\eta_t)} - 1 + \epsilon_1 \right] \tag{72}$$

Similarly, define $\epsilon_2 = \frac{\epsilon_1}{\frac{1}{p_{\text{in}}(\eta_t)} - 1 - \epsilon_1}$, with same high probability,

$$\log \vartheta(\eta_t) \in \left[ \log\left(\frac{1}{p_{\text{in}}(\eta_t)} - 1\right) - \epsilon_2, \left(\frac{1}{p_{\text{in}}(\eta_t)} - 1\right) + \epsilon_2 \right] \tag{73}$$

From the analysis on $\frac{1}{s^{\psi_t}}$, define $\sigma(p, t) = \frac{\mathcal{A}_t \cdot \mathcal{B}_t}{E_t^2 + \bar{R}_{t-1}^2 \cdot E_t^{2 - \frac{1}{n}}}$. There exist $\epsilon_3$ such that $\epsilon_3 \to 0$ as $\epsilon' \to 0$. Further define $\epsilon_4 = \frac{\epsilon_3}{1 - (1 + \frac{1}{\sigma(p,t)})\epsilon_3}$, we have

$$\psi_t \log s \in \left[ 2\log s - \epsilon_4, 2\log s + \epsilon_4 \right] \tag{74}$$

Let $\epsilon'' = \frac{\epsilon_2}{\log \frac{1}{n(1-\bar{\mu}_t^+)}} + \frac{\epsilon_2}{2\log \bar{R}_t} + \epsilon_4$. Then, from above, we have with high probability $[1 - 2\exp(-2(\epsilon')^2 p_{\text{in}}(\eta_t)^2 s)]^2 \cdot \rho$,

$$f_t(\eta_t) \geq \frac{\log(\frac{1}{p_{\text{in}}(\eta_t)} - 1)}{\log \frac{1}{n(1-\bar{\mu}_t^+)}} + \max\left\{\frac{\log(\frac{1}{p_{\text{in}}(\eta_t)} - 1)}{2\log \bar{R}_t} + \mathcal{C}_t, 0\right\} - 2 - \epsilon'' \tag{75}$$

$$f_t(\eta_t) \leq \frac{\log(\frac{1}{p_{\text{in}}(\eta_t)} - 1)}{\log \frac{1}{n(1-\bar{\mu}_t^+)}} + \max\left\{\frac{\log(\frac{1}{p_{\text{in}}(\eta_t)} - 1)}{2\log \bar{R}_t} + \mathcal{C}_t, 0\right\} - 2 + \epsilon'' \tag{76}$$

where $\mathcal{C}_t := \frac{\log(\mathcal{A}_t \cdot \mathcal{B}_t)}{2\log \bar{R}_t} + \frac{\log s}{\log \bar{R}_t}$.

In the proof of Lemma A.5, we already show the existence of $\eta_t$ that satisfy the condition $\eta_t \geq f(\eta_t)$. When $p_{\text{in}}(\eta_t) = \omega_s(\frac{1}{\sqrt{s}})$ and $s = \omega_n(\frac{\log n}{\tau_t^{2n}})$, we have the concentration properties. Since $p_{\text{in}}(\eta_t)$ is strictly monotonically increasing with $\eta_t$ and $\eta_t \geq cn$, thus, we need $p_{\text{in}}(\lceil cn \rceil) = \omega_s(\frac{1}{\sqrt{s}})$. Thus, we require

$$s = \omega_n\left([\frac{1}{c}(1 - \frac{1-p}{k-1})]^{-2cn}(\frac{1-p}{k-1})^{-2(1-c)n}\right). \tag{77}$$

Therefore, when the above condition is satisfied, we only require

$$\eta_t \geq \underbrace{\frac{\log(\frac{1}{p_{\text{in}}(\eta_t)} - 1)}{\log \frac{1}{n(1-\bar{\mu}_t^+)}} + \max\left\{\frac{\log(\frac{1}{p_{\text{in}}(\eta_t)} - 1)}{2\log \bar{R}_t} + \mathcal{C}_t, 0\right\} - 2}_{(f_t^* \circ \frac{1-p_{\text{in}}}{p_{\text{in}}})(\eta_t)}, \quad \eta_t \in \{\lceil cn \rceil, ..., n\}. \tag{78}$$

where $p_{\text{in}}(\eta_t) = \sum_{j=0}^{\eta_t} \binom{n}{j}(\frac{1-p}{k-1})^{n-j}(1 - \frac{1-p}{k-1})^j$.

One direct observation from above is that

- In noise free regime ($t$ is close to 0, i.e. $\bar{\alpha}_t \to 1$), we have $\log \frac{1}{n(1-\bar{\alpha}_t^+)}, \log \bar{R}_t \to \infty$ and $\mathcal{C}_t \to 0$. Therefore, $\eta_t \to 0$.

- In noisy regime ($t$ is close to T, i.e. $\bar{\alpha}_t \to 0$), similarly, we have $\log \frac{1}{n(1-\bar{\alpha}_t^+)} \to \log \frac{1}{n}$, $\log \bar{R}_t \to 0$, and $\mathcal{C}_t \to \infty$. Therefore, $\eta_t \to n$.

Now, we consider how skewness parameter $p$ influence the selection of $\eta_t$. Given fixed diffusion coefficients $\{\alpha_t\}_{t\in[T]}$ with $\bar{\bar{\alpha}}_t$ monotonically decreases from 1 to 0 as $t$ goes from 1 to $T$. Given relatively large $s$ (neglect the influence of concentration error $\epsilon''$), and corresponding $n$ that satisfies $s = \omega_n(\max\{\log n \cdot (\frac{1}{\frac{1-p}{k-1} + \frac{\bar{\mu}_t^-}{\bar{\mu}_t^+}(1-\frac{1-p}{k-1})})^{2n}, [\frac{1}{c}(1-\frac{1-p}{k-1})]^{-2cn}(\frac{1-p}{k-1})^{-2(1-c)n}\})$. We have the following observation: When we increase the skewness of the distribution, $f_t^* \circ \frac{1-p_{\text{in}}}{p_{\text{in}}}(\eta_t)$ will increase monotonically with $p$, such that $\eta_t$ will increase to $n$ faster. In other words, given two skewness parameter $p, p'$ with $p > p'$, we have $\eta_t^p \geq \eta_t^{p'}$ where $\eta_t^p, \eta_t^{p'}$ denote the minimal $\eta$ that satisfy the constraint under two skewness parameters. Similar derivation can be applied to $g_t^* \circ \frac{1-p_{\text{in}}}{p_{\text{in}}}$ and we obtain the results for $c_t^*$. $\qquad \square$

**Proposition F.3** (Sufficient Condition)**.** *Given a skewed distribution with parameter $p$. Let $\mathbf{v}^*$ be a non-majority point and $\mathcal{V}_0 \backslash \{\mathbf{v}^*\}$ sampled from the distribution. Define $\tau_t = \frac{1-p}{k-1} + \frac{\bar{\mu}_t^-}{\bar{\mu}_t^+}(1 - \frac{1-p}{k-1})$. When $s = \omega_n([\frac{1}{c}(1-\frac{1-p}{k-1})]^{-2cn}(\frac{1-p}{k-1})^{-2(1-c)n}, \frac{\log n}{\tau_t^{2n}})$ for some $c \in (\frac{1-p}{k-2-p}, 1)$, the sufficient conditions of $\eta_t, c_t^*$ that satisfy Eq. 9 are*

$$\eta_t \geq n - \left(\frac{n - \log(s\sqrt{\frac{\bar{\alpha}_{t-1} - \bar{\alpha}_t}{k\bar{\mu}_t^+ \bar{\mu}_t^-}})/\log \bar{R}_t}{2\log \frac{k-1}{1-p}/\log(\max\{\frac{1}{n\bar{\mu}_t^-}, 1\}) + 1}\right)_+, \tag{79}$$

$$c_t^* \geq \frac{n - \eta_t}{\eta_t} - \frac{\log\left(\frac{1}{2e} \cdot (\frac{1}{e\bar{\mu}_t^-})^{\frac{n-\eta_t}{\eta_t}}\right)}{\log \frac{k-1}{1-p} + \log \frac{1}{e\bar{\mu}_t^-}}. \tag{80}$$

***Proof of Proposition F.3.*** From Proposition F.2, when $s = \omega_n([\frac{1}{c}(1 - \frac{1-p}{k-1})]^{-2cn}(\frac{1-p}{k-1})^{-2(1-c)n})$, we have $\eta_t, c_t^*$ from Eq. (68). Further since

$$p_{\text{in}}(\eta_t) \geq \sum_{i=0}^{\eta_t} (\frac{n}{i})^i (\frac{1-p}{k-1})^{n-i} (1 - \frac{1-p}{k-1})^i \geq (\frac{n}{\eta_t})^{\eta_t} (\frac{1-p}{k-1})^{n-\eta_t} (1 - \frac{1-p}{k-1})^{\eta_t} \tag{81}$$

From $c \geq \frac{1-p}{k-2-p}$, we have

$$\log(\frac{1}{p_{\text{in}}(\eta_t)} - 1) \leq n \log(\frac{k-1}{1-p}) + \eta_t (\log \frac{n}{\eta_t} - \log \frac{k-1}{1-p} + \log \frac{1}{1 - \frac{1-p}{k-1}}) \tag{82}$$

$$\leq (n - \eta_t) \log(\frac{k-1}{1-p}) \tag{83}$$

Therefore, the sufficient conditions of $\eta_t$ and $c_t^*$ will be

$$\eta_t \geq \frac{(n - \eta_t) \log(\frac{k-1}{1-p})}{\log \frac{1}{n(1-\bar{\mu}_t^+)}} + \max\left\{ \frac{(n - \eta_t) \log(\frac{k-1}{1-p})}{2 \log \bar{R}_t} + \mathcal{C}_t, 0 \right\} - 2, \tag{84}$$

$$c_t^* \geq \frac{\frac{n}{\eta_t} \log(\frac{k-1}{1-p}) - (1 + c_t^*) \log(\frac{k-1}{1-p}) + \log 2e}{\log \frac{1}{\bar{\mu}_t^-} - 1}. \tag{85}$$

Further simplify the above terms, we get

$$\eta_t \geq n - \left( \frac{n - \log(s\sqrt{\frac{\overline{\alpha}_{t-1} - \overline{\alpha}_t}{k\bar{\mu}_t^+ \bar{\mu}_t^-}})/\log \bar{R}_t}{2 \log \frac{k-1}{1-p} / \log(\max\{\frac{1}{n\bar{\mu}_t^-}, 1\}) + 1} \right)_+, \tag{86}$$

$$c_t^* \geq \frac{n - \eta_t}{\eta_t} - \frac{\log\left(\frac{1}{2e} \cdot (\frac{1}{e\bar{\mu}_t^-})^{\frac{n-\eta_t}{\eta_t}}\right)}{\log \frac{k-1}{1-p} + \log \frac{1}{e\bar{\mu}_t^-}}. \tag{87}$$

$\square$

## G Experimental Settings

### G.1 Datasets

In this section, we briefly introduce the real dataset included in the paper.

**Adult (Kohavi et al., 1996).** The Adult dataset, also known as the Census Income dataset, contains information collected from the 1994 US Census Bureau database. It serves the purpose of predicting an individual's income category (above or below $50,000 per year) based on demographic attributes such as age, education, occupation, and more. With around 32,000 records and 14 features, it is widely used for classification tasks.

**German Credit (Hofmann, 1994).** The German Credit dataset, curated by Prof. Hofmann, encompasses data from 1000 individuals seeking bank credit. Each entry is characterized by 20 categorical / numerical attributes, with labels as either good or bad credit risk based on these features.

**Loan (ItsSuru).** The Loan Status dataset consists of 500 unique entries, each with 11 features. These records capture customer interactions with a bullet loan product. Labels within the dataset indicate whether a loan was approved or not, making it suitable for classification tasks.

### G.2 Environment

Experiments were performed on a server with four Intel 24-Core Gold 6248R CPUs, 1TB DRAM, and eight NVIDIA QUADRO RTX 6000 (24GB) GPUs.

## H  Additional Setup Configurations for Experiments on Real Datasets

In this section, we provide additional details regarding the experiments on real dataset in Section 4.2.

### H.1  Dataset Preprocessing

**Adult.**  The original Adult dataset consists of 14 continuous or discrete features. In our setting, we have selected the main 9 features, namely: **age, workclass, education, marital-status, occupation, relationship, race, gender, hours-per-week.** We have performed interval partitioning on the continuous features and merged some multi-category features (e.g., education) into a single category for discrete features. As a result, the number of categories for each feature does not exceed 5.

**German Credit.** The German Credit dataset is comprised of 20 features, both categorical and numerical. For our study, we focused on 10 primary features: **status of existing checking account, duration in month, credit history, purpose, credit amount, savings account / bonds, employment, personal status and sex, age in years, job.** We categorized numerical features, ensuring no category contained more than 5 subdivisions.

**Loan.**  The Loan dataset encompasses 11 features, both numerical and categorical. In our study, we incorporated all these categories and further segmented the numerical features into categories, each containing a maximum of four subdivisions. The specific features are: **Gender, Married, Dependents, Education, Self Employed, Applicant Income, Coapplicant Income, Loan Amount, Loan Amount Term, Credit History, Property Area.**

### H.2  Experiments on Testing Effectiveness of Privacy Bound Algorithm in Sec. 4.2.1

#### H.2.1  Denoising Network Architecture and Training Procedure

We designed a four-layer MLP equipped with 256 hidden neurons. Throughout the architecture, batch normalization was integrated, and the leakyReLU served as the activation function. For consistency during training, we anchored the random seed at 123. The denoising network underwent training via the Adam optimizer, set with a learning rate of 1e-3 and a weight decay of 5e-4. Our approach involved uniformly sampling the diffusion step and drawing batches of 30 samples each. The training spanned 100 epochs, focusing on minimizing the binary cross-entropy loss.

#### H.2.2  Evaluation via DownStream Tasks

To evaluate the efficacy of DDMs, we divided the original dataset into three segments: training, validation, and testing, adhering to an 8:1:1 ratio. Leveraging the trained denoising network, we generated a synthetic dataset, ensuring the number of instances for each class mirrored that of the original dataset. We then engaged in a downstream binary classification to measure the utility of the DDMs. In this context, we trained an independent MLP classifier using the synthetic dataset and subsequently evaluate its classification performance on the test set (original dataset).

### H.3  Experiments on Membership Inference Attack in Sec. 4.2.2

#### H.3.1  Discriminative Model Architecture and Training Procedure

The discriminative model is designed to overfit the data released by the target model, which is indicative of the training data. We implemented a MLP for classification, featuring a three-layer architecture with each layer containing 256 neurons. The model leverages the ReLU activation function across its structure. The

network's training process is guided by the Adam optimizer and the learning rate is adaptively adjusted, starting at 0.01, to optimize performance during training. The model iterates up to 1000 times or until the tolerance level of 1e-6 is reached.

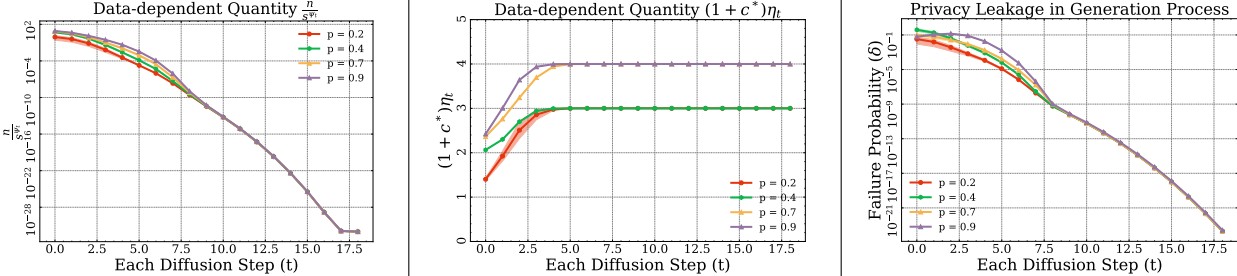

Figure 9: All based on single step-$t$ term in Eq.main (Sigmoid Schedule): **LEFT:** Characterization of $\frac{1}{s^{\psi_t}}$. **MIDDLE:** Characterization of $(1 + c_t^*)\eta_t$. **RIGHT:** Characterization of Privacy Leakage (Main Privacy Term). Experimental Setup: Given specific DDM design $k = 5, n = 5, T = 20, \epsilon = 10$ trained on dataset with $s = 1000$ following skewed distribution with parameter $p$. We consider a fixed $\mathbf{v}^*$ where each column has a non-majority category.

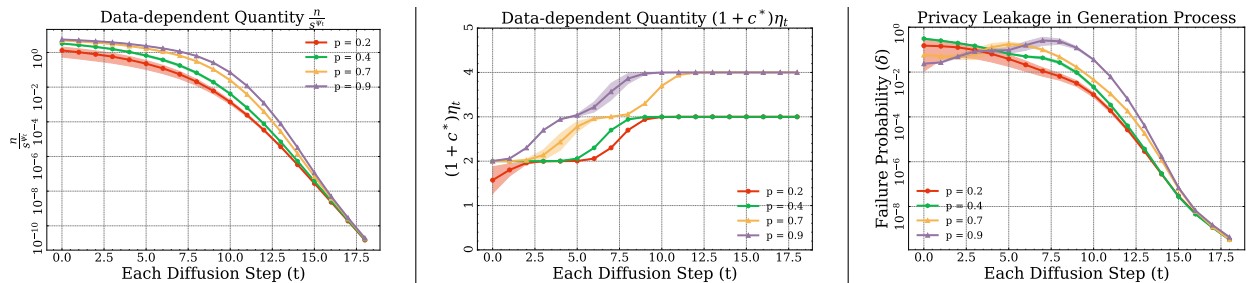

Figure 10: All based on single step-$t$ term in Eq.main (Cosine Schedule): **LEFT:** Characterization of $\frac{1}{s^{\psi_t}}$. **MIDDLE:** Characterization of $(1 + c_t^*)\eta_t$. **RIGHT:** Characterization of Privacy Leakage (Main Privacy Term). Experimental Setup: Given specific DDM design $k = 5, n = 5, T = 20, \epsilon = 10$ trained on dataset with $s = 1000$ following skewed distribution with parameter $p$. We consider a fixed $\mathbf{v}^*$ where each column has a non-majority category.

# I  Additional Experimental Results

In this section, we present additional experiments to backup our main results in Sec. 4.

## I.1  Privacy Leakage and Behavior of Data-dependent Quantities under Various Noise Schedules

In our experiments, we assess the privacy leakage and the behavior of data-dependent quantities $\psi_t$, $\eta_t$, and $c_t^*$ across different DDM model configurations. In particular, we explore two additional noise schedules: sigmoid and cosine (defined as $\overline{\alpha}_t = \frac{f(t)}{f(0)}$, where $f(t) = \cos\left(\frac{t/T+s}{1+s} \cdot \frac{\pi}{2}\right)^2$). The outcomes, depicted in Fig. 9 and Fig.10, resonate with our theoretical findings. It is observed that under all noise schedules, the privacy leakage intensifies for distributions that are more skewed.

## I.2  Most Private and Most Sensitive Points under Skewed Distribution

Next, we delineate the evolving trends of the most private and most sensitive samples as the skewness parameter shifts, providing insights into its impact on privacy leakage. We specifically focus on three DDM configurations: $n = 5, k = 5, T = 20$, $n = 10, k = 5, T = 20$, and $n = 20, k = 5, T = 100$. For each setup, we adjust the skewness parameters $p$ and curate the dataset $\mathcal{V}_0$ in accordance with the respective distributions. The most private and sensitive data points, based on the privacy budget, are illustrated in Fig. 11.

As evident in the figure, escalating the distribution's skewness boosts the privacy budget for the most sensitive points (represented by dark-colored lines). Conversely, the privacy budget for the most private points (depicted by light-colored lines) diminishes. This observation is consistent with our theoretical analyses: for highly skewed distributions, points distant from the main cluster exhibit reduced similarity, leading to augmented privacy leakage. Meanwhile, predominant points benefit from enhanced similarity, resulting in diminished privacy concerns.

Figure 11: Privacy budget trends across DDM configurations with varying skewness.

## J    Main Lemmas and Corresponding Proofs

### J.1    Proof of Lemma A.1

*Proof.* To prove the pDP guarantee of DDMs, we seek to a stronger notion termed probabilistically differential privacy. First, we present the definition of $(\epsilon, \delta)$-probabilistically differential privacy (PDP).

**Definition J.1** $((\epsilon, \delta)$-Probabilistically Differential Privacy (Meiser, 2018))**.** A randomized mechanism $\mathcal{M} : \mathcal{D} \to \mathcal{R}$ satisfies $(\epsilon, \delta)$-probabilistically differential private if for any adjacent datasets $\mathcal{V}_0, \mathcal{V}_1 \in \mathcal{D}$ and there exists sets $\mathcal{O}_0 \subseteq range(\mathcal{M})$ where $\mathcal{P}(\mathcal{M}(\mathcal{V}_0) \subseteq \mathcal{O}_0) \leq \delta$, such that $\forall \mathcal{O} \subseteq range(\mathcal{M})$, we have

$$\mathcal{P}(\mathcal{M}(\mathcal{V}_0) \in \mathcal{O} \backslash \mathcal{O}_0) \leq e^{\epsilon} \mathcal{P}(\mathcal{M}(\mathcal{V}_1) \in \mathcal{O} \backslash \mathcal{O}_0) \tag{88}$$

The following lemma connects the relationship between PDP and DP.

**Lemma J.1** $((\epsilon, \delta)$-PDP to $(\epsilon, \delta)$-pDP (Meiser, 2018))**.** If a randomized mechanism $\mathcal{M}$ satisfies $(\epsilon, \delta)$-PDP, it also satisfies $(\epsilon, \delta)$-pDP with respect to $(\mathcal{V}_0, \mathbf{v}^*)$.

The following lemma further characterizes PDP with KL divergence.

**Lemma J.2** (Characterization of $(\epsilon, \delta)$-PDP with Kl Divergence (Lin et al., 2021))**.** Given two adjacent tabular datasets $\mathcal{V}_0, \mathcal{V}_1$, consider a mechanism $\mathcal{M}$, let $\pi_0^*$ and $\pi_1^*$ denote the probability measure of $\mathcal{M}(\mathcal{V}_0), \mathcal{M}(\mathcal{V}_1)$ respectively. If there exist a constant $\tau$ such that

$$\mathcal{D}_{\mathrm{KL}}(\pi_0^* \| \pi_1^*) + \mathcal{D}_{\mathrm{KL}}(\pi_1^* \| \pi_0^*) \leq \tau \tag{89}$$

the mechanism $\mathcal{M}$ satisfies $(\epsilon, \frac{\tau}{\epsilon(1-e^{-\epsilon})})$-probabilistic differential privacy for all $\epsilon > 0$.

$\square$

### J.2    Proof of Lemma A.2

*Proof.* First consider generating a single sample. In the generation process, $\mathbf{v}_{T|\lambda} \to \mathbf{v}_{T-1|\lambda} \to \cdots \to \mathbf{v}_{t|\lambda} \to \cdots \to \mathbf{v}_{T_{\mathrm{rl}}|\lambda}$ forms a Markov chain. First we consider $\mathcal{D}_{\mathrm{KL}}(p_\phi(\mathbf{v}_{T_{\mathrm{rl}}|0}) \| p_\phi(\mathbf{v}_{T_{\mathrm{rl}}|1}))$:

$$\mathcal{D}_{\mathrm{KL}}(p_\phi(\mathbf{v}_{T_{\mathrm{rl}}|0}) \| p_\phi(\mathbf{v}_{T_{\mathrm{rl}}|1})) \tag{90}$$

$$\overset{(i)}{\leq} \mathcal{D}_{\mathrm{KL}}(p_\phi(\mathbf{v}_{T_{\mathrm{rl}}|0}, \mathbf{v}_{T_{\mathrm{rl}}+1|0}, ..., \mathbf{v}_{T|0}) \| p_\phi(\mathbf{v}_{T_{\mathrm{rl}}|1}, \mathbf{v}_{T_{\mathrm{rl}}+1|1}, ..., \mathbf{v}_{T|1})) \tag{91}$$

$$\overset{(ii)}{=} \mathbb{E}_{p_\phi(\mathbf{v}_{T_{\mathrm{rl}}|0}, \mathbf{v}_{T_{\mathrm{rl}}+1|0}, ..., \mathbf{v}_{T|0})} \left[ \frac{p_\phi(\mathbf{v}_{T|0})}{p_\phi(\mathbf{v}_{T|1})} + \sum_{t=T_{\mathrm{rl}}+1}^{T} \log \frac{p_\phi(\mathbf{v}_{t-1|0} | \mathbf{v}_{t|0})}{p_\phi(\mathbf{v}_{t-1|1} | \mathbf{v}_{t|1})} \right] \tag{92}$$

$$= \sum_{\mathbf{v}_{T_{\mathrm{rl}}}, ..., \mathbf{v}_T} p_\phi(\mathbf{v}_{T|0}) p_\phi(\mathbf{v}_{T-1|0} | \mathbf{v}_{T|0}) \cdots p_\phi(\mathbf{v}_{T_{\mathrm{rl}}|0} | \mathbf{v}_{T_{\mathrm{rl}}+1|0}) \left[ \frac{p_\phi(\mathbf{v}_{T|0})}{p_\phi(\mathbf{v}_{T|1})} + \sum_{t=T_{\mathrm{rl}}+1}^{T} \log \frac{p_\phi(\mathbf{v}_{t-1|0} | \mathbf{v}_{t|0})}{p_\phi(\mathbf{v}_{t-1|1} | \mathbf{v}_{t|1})} \right] \tag{93}$$

$$\overset{(iii)}{=} \mathcal{D}_{\mathrm{KL}}(p_\phi(\mathbf{v}_{T|0}) \| p_\phi(\mathbf{v}_{T|1})) + \sum_{t=T_{\mathrm{rl}}+1}^{T} \sum_{(\mathbf{v}_{t-1}, \mathbf{v}_t)} p_\phi(\mathbf{v}_{t|0}) \prod_{i=1}^{n} p_\phi(\mathbf{v}_{t-1|0}^i | \mathbf{v}_{t|0}) \left( \sum_i \log \frac{p_\phi(\mathbf{v}_{t-1|0}^i | \mathbf{v}_{t|0})}{p_\phi(\mathbf{v}_{t-1|1}^i | \mathbf{v}_{t|1})} \right) \tag{94}$$

$$= \mathcal{D}_{\mathrm{KL}}(p_\phi(\mathbf{v}_{T|0}) \| p_\phi(\mathbf{v}_{T|1})) + \sum_{t=T_{\mathrm{rl}}+1}^{T} \sum_{i} \mathbb{E}_{\mathbf{v} \sim p_\phi(\mathbf{v}_{t|0})} [\mathcal{D}_{\mathrm{KL}}(p_\phi(\mathbf{v}_{t-1|0}^i | \mathbf{v}_{t|0} = \mathbf{v}) \| p_\phi(\mathbf{v}_{t-1|1}^i | \mathbf{v}_{t|1} = \mathbf{v}))] \quad (95)$$

where $(i)$ follows from Lemma K.1, $(ii)$ leverages Markov property and $(iii)$ is due to conditional independence. When generating $m$ independent samples, consider the coupled sum and we prove that

$$\mathcal{D}_{\mathrm{KL}}(\mathcal{M}_{T_{\mathrm{rl}}}(\mathcal{V}_0) \| \mathcal{M}_{T_{\mathrm{rl}}}(\mathcal{V}_1)) + \mathcal{D}_{\mathrm{KL}}(\mathcal{M}_{T_{\mathrm{rl}}}(\mathcal{V}_1) \| \mathcal{M}_{T_{\mathrm{rl}}}(\mathcal{V}_0)) \quad (96)$$

$$\leq m \sum_{t=T_{\mathrm{rl}}+1}^{T} \sum_{\lambda \in \{0,1\}} \sum_{i=1}^{n} \mathbb{E}_{\mathbf{v}_{t|\lambda} \sim p_\phi(\mathbf{v}_{t|\lambda})} [\mathcal{D}_{\mathrm{KL}}(p_\phi(\mathbf{v}_{t-1|\lambda}^i | \mathbf{v}_{t|\lambda}) \| p_\phi(\mathbf{v}_{t-1|1-\lambda}^i | \mathbf{v}_{t|1-\lambda}))] \quad (97)$$

$\square$

### J.3 Proof of Lemma A.3

*Proof.* According to the generation process, from Lemma K.2, we know that for any $\lambda \in \{0,1\}$

$$p_\phi(\mathbf{v}_{t-1|\lambda}^i | \mathbf{v}_{t|\lambda}) = \sum_{\mathbf{v}_{0|\lambda}} p_\phi(\mathbf{v}_{i-1|\lambda} | \mathbf{v}_{0|\lambda}, \mathbf{v}_{t|\lambda}) p_\phi(\mathbf{v}_{0|\lambda} | \mathbf{v}_{t|\lambda}) \in \left[ \frac{\mu_t^- \cdot \bar{\mu}_{t-1}^-}{\bar{\mu}_t^+}, \frac{\mu_t^+ \cdot \bar{\mu}_{t-1}^+}{\bar{\mu}_t^+} \right]. \quad (98)$$

Define $f_1(\gamma_t) = \frac{k \cdot (\mu_t^+ \cdot \bar{\mu}_{t-1}^+)^2}{\bar{\mu}_t^+ \cdot \mu_t^- \cdot \bar{\mu}_{t-1}^-} \cdot \sqrt{2\gamma_t}$, $f_2(\tilde{\gamma}_t) = 2 \log \left( \frac{\mu_t^+ \cdot \bar{\mu}_{t-1}^+}{\mu_t^- \cdot \bar{\mu}_{t-1}^-} \right) \cdot \tilde{\gamma}_t$. We have

$$\sum_{\lambda \in \{0,1\}} \mathbb{E}_{\mathbf{v}_t \sim p_\phi(\mathbf{v}_{t|\lambda})} [\mathcal{D}_{\mathrm{KL}}(p_\phi(\mathbf{v}_{t-1|\lambda}^i | \mathbf{v}_{t|\lambda}) \| p_\phi(\mathbf{v}_{t-1|1-\lambda}^i | \mathbf{v}_{t|1-\lambda}))]$$

$$\overset{(i)}{\leq} \sum_{\lambda \in \{0,1\}} \mathbb{E}_{\mathbf{v}_t \sim q(\mathbf{v}_{t|\lambda})} [\mathcal{D}_{\mathrm{KL}}(p_\phi(\mathbf{v}_{t-1|\lambda}^i | \mathbf{v}_{t|\lambda}) \| p_\phi(\mathbf{v}_{t-1|1-\lambda}^i | \mathbf{v}_{t|1-\lambda}))] + f_2(\tilde{\gamma}_t) \quad (99)$$

$$\overset{(ii)}{\leq} \sum_{\lambda \in \{0,1\}} \mathbb{E}_{\mathbf{v}_t \sim q(\mathbf{v}_{t|\lambda})} [\mathcal{D}_{\mathrm{KL}}(q(\mathbf{v}_{t-1|\lambda}^i | \mathbf{v}_{t|\lambda}) \| q(\mathbf{v}_{t-1|1-\lambda}^i | \mathbf{v}_{t|1-\lambda}))] + f_1(\gamma_t) + f_2(\tilde{\gamma}_t) \quad (100)$$

where $(i)$ is from Assumption 2 while $(ii)$ is owing to Assumption 1 and Lemma K.3. $\square$

### J.4 Proof of Lemma A.4

*Proof.* To begin with, we first calculate $q(\mathbf{v}_{t-1|\lambda}^i = \mathbf{v}_{t-1}^i, \mathbf{v}_{t|\lambda} = \mathbf{v}_t)$ and $q(\mathbf{v}_{t|\lambda} = \mathbf{v}_t), \lambda \in \{0,1\}$.

$$q(\mathbf{v}_{t-1|\lambda}^i = \mathbf{v}_{t-1}^i, \mathbf{v}_{t|\lambda} = \mathbf{v}_t) \quad (101)$$

$$= \frac{1}{|\mathcal{V}_\lambda|} \sum_{\mathbf{v}_0 \in \mathcal{V}_0} q(\mathbf{v}_{t-1|\lambda}^i = \mathbf{v}_{t-1}^i, \mathbf{v}_{t|\lambda} = \mathbf{v}_t | \mathbf{v}_{0|\lambda} = \mathbf{v}_0) \quad (102)$$

$$= \frac{1}{|\mathcal{V}_\lambda|} \sum_{\mathbf{v}_0 \in \mathcal{V}_0} q(\mathbf{v}_{t|\lambda} = \mathbf{v}_t^i | \mathbf{v}_{t-1|\lambda}^i = \mathbf{v}_{t-1}^i) \frac{q(\mathbf{v}_{t-1|\lambda} = \mathbf{v}_{t-1}^i | \mathbf{v}_{0|\lambda}^i = \mathbf{v}_0^i)}{q(\mathbf{v}_{t|\lambda} = \mathbf{v}_t^i | \mathbf{v}_{0|\lambda}^i = \mathbf{v}_0^i)} \prod_{j} q(\mathbf{v}_{t|\lambda} = \mathbf{v}_t^j | \mathbf{v}_{0|\lambda}^j = \mathbf{v}_{t-1}^j) \quad (103)$$

$$= \frac{1}{|\mathcal{V}_\lambda|} \sum_{\mathbf{v}_0 \in \mathcal{V}_0} \frac{(\mu_t^+)^{\mathbb{1}_{\mathbf{v}_t^i = \mathbf{v}_{t-1}^i}} (\mu_t^-)^{\mathbb{1}_{\mathbf{v}_t^i \neq \mathbf{v}_{t-1}^i}} (\bar{\mu}_{t-1}^+)^{\mathbb{1}_{\mathbf{v}_{t-1}^i = \mathbf{v}_0^i}} (\bar{\mu}_{t-1}^-)^{\mathbb{1}_{\mathbf{v}_{t-1}^i \neq \mathbf{v}_0^i}}}{(\bar{\mu}_t^+)^{\mathbb{1}_{\mathbf{v}_t^i = \mathbf{v}_0^i}} (\bar{\mu}_t^-)^{\mathbb{1}_{\mathbf{v}_t^i \neq \mathbf{v}_0^i}}} \prod_{j} q(\mathbf{v}_{t|\lambda} = \mathbf{v}_t^j | \mathbf{v}_{0|\lambda}^j = \mathbf{v}_{t-1}^j) \quad (104)$$

$$= \frac{1}{|\mathcal{V}_\lambda|} \sum_{\mathbf{v}_0 \in \mathcal{V}_\lambda} \underbrace{\frac{[(\mu_t^+)^{1-\mathbb{1}_{\mathbf{v}_t^i \neq \mathbf{v}_{t-1}^i}} (\mu_t^-)^{\mathbb{1}_{\mathbf{v}_t^i \neq \mathbf{v}_{t-1}^i}}] \cdot [(\bar{\mu}_{t-1}^+)^{1-\mathbb{1}_{\mathbf{v}_{t-1}^i \neq \mathbf{v}_0^i}} (\bar{\mu}_{t-1}^-)^{\mathbb{1}_{\mathbf{v}_{t-1}^i \neq \mathbf{v}_0^i}}]}{(\bar{\mu}_t^+)^{1-\mathbb{1}_{\mathbf{v}_t^i \neq \mathbf{v}_0^i}} (\bar{\mu}_t^-)^{\mathbb{1}_{\mathbf{v}_t^i \neq \mathbf{v}_0^i}}}}_{\text{denote as } \tau(\mathbf{v}_0^i, \mathbf{v}_{t-1}^i, \mathbf{v}_t^i)} \cdot \frac{(\bar{\mu}_t^+)^{n-\bar{\omega}(\mathbf{v}_0, \mathbf{v}_t)}}{(\bar{\mu}_t^-)^{-\bar{\omega}(\mathbf{v}_0, \mathbf{v}_t)}} \quad (105)$$

Similar derivation for $q(\mathbf{v}_{t|\lambda} = \mathbf{v}_t)$, we obtain

$$q(\mathbf{v}_{t|\lambda} = \mathbf{v}_t) = \frac{1}{|\mathcal{V}_\lambda|} \sum_{\mathbf{v}_0 \in \mathcal{V}_\lambda} (\bar{\mu}_t^+)^{n-\bar{\omega}(\mathbf{v}_0, \mathbf{v}_t)} (\bar{\mu}_t^-)^{\bar{\omega}(\mathbf{v}_0, \mathbf{v}_t)} \quad (106)$$

Now, we consider the coupled KL divergence:

$$\mathcal{D}_{\text{KL}}(q(\mathbf{v}^i_{t-1|0}|\mathbf{v}_{t|0})\|q(\mathbf{v}^i_{t-1|1}|\mathbf{v}_{t|1})) + \mathcal{D}_{\text{KL}}(q(\mathbf{v}^i_{t-1|1}|\mathbf{v}_{t|1})\|q(\mathbf{v}^i_{t-1|0}|\mathbf{v}_{t|0})) \tag{107}$$

$$= \sum_{\mathbf{v}^i_{t-1}} (q(\mathbf{v}^i_{t-1|0} = \mathbf{v}^i_{t-1}|\mathbf{v}_{t|0}) - q(\mathbf{v}^i_{t-1|1} = \mathbf{v}^i_{t-1}|\mathbf{v}_{t|1})) \log \frac{q(\mathbf{v}^i_{t-1|0} = \mathbf{v}^i_{t-1}|\mathbf{v}_{t|0})}{q(\mathbf{v}^i_{t-1|1} = \mathbf{v}^i_{t-1}|\mathbf{v}_{t|1})} \tag{108}$$

$$= \sum_{\mathbf{v}^i_{t-1}} (q(\mathbf{v}^i_{t-1|0} = \mathbf{v}^i_{t-1}|\mathbf{v}_{t|0}) - q(\mathbf{v}^i_{t-1|1} = \mathbf{v}^i_{t-1}|\mathbf{v}_{t|1})) \left[ \log \frac{q(\mathbf{v}^i_{t-1|0} = \mathbf{v}^i_{t-1}, \mathbf{v}_{t|0})}{q(\mathbf{v}^i_{t-1|1} = \mathbf{v}^i_{t-1}, \mathbf{v}_{t|1})} + \log \frac{q(\mathbf{v}_{t|1})}{q(\mathbf{v}_{t|0})} \right] \tag{109}$$

$$\overset{(i)}{=} \sum_{\mathbf{v}^i_{t-1}} (q(\mathbf{v}^i_{t-1|0} = \mathbf{v}^i_{t-1}|\mathbf{v}_{t|0}) - q(\mathbf{v}^i_{t-1|1} = \mathbf{v}^i_{t-1}|\mathbf{v}_{t|1})) \log \frac{q(\mathbf{v}^i_{t-1|0} = \mathbf{v}^i_{t-1}, \mathbf{v}_{t|0})}{q(\mathbf{v}^i_{t-1|1} = \mathbf{v}^i_{t-1}, \mathbf{v}_{t|1})} \tag{110}$$

$$\overset{(ii)}{\leq} \frac{1}{2} \|q(\mathbf{v}^i_{t-1|0} = \mathbf{v}^i_{t-1}|\mathbf{v}_{t|0}) - q(\mathbf{v}^i_{t-1|1} = \mathbf{v}^i_{t-1}|\mathbf{v}_{t|1})\|_{l_1} \left[ \max_{\mathbf{v}^i_{t-1}} \log \frac{(s+1) \cdot q(\mathbf{v}^i_{t-1|0} = \mathbf{v}^i_{t-1}, \mathbf{v}_{t|0})}{s \cdot q(\mathbf{v}^i_{t-1|1} = \mathbf{v}^i_{t-1}, \mathbf{v}_{t|1})} \right.$$
$$\left. - \min_{\mathbf{v}^i_{t-1}} \log \frac{(s+1) \cdot q(\mathbf{v}^i_{t-1|0} = \mathbf{v}^i_{t-1}, \mathbf{v}_{t|0})}{s \cdot q(\mathbf{v}^i_{t-1|1} = \mathbf{v}^i_{t-1}, \mathbf{v}_{t|1})} \right] \tag{111}$$

where $(i)$ is due to the fact that $\log \frac{q(\mathbf{v}_{t|1})}{q(\mathbf{v}_{t|0})}$ is independent of $\mathbf{v}^i_{t-1|\lambda}, \lambda \in \{0, 1\}$ and $(ii)$ is from Lemma K.4.

Before we dive into specific terms, we define two terms for simplicity: $\kappa(\mathbf{v}^{*i}, \mathbf{v}^i_0, \mathbf{v}^i_{t-1}, \mathbf{v}^i_t) = \frac{\tau(\mathbf{v}^i_0, \mathbf{v}^i_{t-1}, \mathbf{v}^i_t)}{\tau(\mathbf{v}^{*i}, \mathbf{v}^i_{t-1}, \mathbf{v}^i_t)}$ and $\bar{\kappa}(\mathbf{v}^{*i}, \mathbf{v}^i_0, \mathbf{v}^i_{t-1}, \mathbf{v}^i_t) = \frac{\tau(\mathbf{v}^i_0, \mathbf{v}^i_{t-1}, \mathbf{v}^i_t)}{\tau(\mathbf{v}^{*i}, \mathbf{v}^i_{t-1}, \mathbf{v}^i_t)} \cdot \left(\frac{\bar{\mu}^+_t}{\bar{\mu}^-_t}\right)^{1_{v^* \neq \mathbf{v}^i_t} - 1_{\mathbf{v}^i_0 \neq \mathbf{v}^i_t}}$. Discussion on specific values of $\kappa(\mathbf{v}^i_0, \mathbf{v}^i_{t-1}, \mathbf{v}^i_t), \kappa(\mathbf{v}^{*i}, \mathbf{v}^i_0, \mathbf{v}^i_{t-1}, \mathbf{v}^i_t)$ and $\bar{\kappa}(\mathbf{v}^{*i}, \mathbf{v}^i_0, \mathbf{v}^i_{t-1}, \mathbf{v}^i_t)$ are presented in Appendix L.

Now, we consider the term $\log \frac{(s+1) \cdot q(\mathbf{v}^i_{t-1|0} = \mathbf{v}^i_{t-1}, \mathbf{v}_{t|0})}{s \cdot q(\mathbf{v}^i_{t-1|1} = \mathbf{v}^i_{t-1}, \mathbf{v}_{t|1})}$. For any $\mathbf{v}_t \in \mathcal{X}^n$,

$$\log \frac{(s+1) \cdot q(\mathbf{v}^i_{t-1|0} = \mathbf{v}^i_{t-1}, \mathbf{v}_{t|0} = \mathbf{v}_t)}{s \cdot q(\mathbf{v}^i_{t-1|1} = \mathbf{v}^i_{t-1}, \mathbf{v}_{t|1} = \mathbf{v}_t)} \tag{112}$$

$$= \log \left( 1 + \frac{1}{\sum_{\mathbf{v}_0 \in \mathcal{V}_1} \frac{\tau(\mathbf{v}^i_0, \mathbf{v}^i_{t-1}, \mathbf{v}^i_t)}{\tau(\mathbf{v}^{*i}, \mathbf{v}^i_{t-1}, \mathbf{v}^i_t)} (\bar{\mu}^+_t)^{\bar{\omega}(\mathbf{v}^*, \mathbf{v}_t) - \bar{\omega}(\mathbf{v}_0, \mathbf{v}_t)} (\bar{\mu}^-_t)^{\bar{\omega}(\mathbf{v}_0, \mathbf{v}_t) - \bar{\omega}(\mathbf{v}^*, \mathbf{v}_t)}} \right) \tag{113}$$

$$= \log \left( 1 + \frac{1}{\sum_{\mathbf{v}_0 \in \mathcal{V}_1} \frac{\tau(\mathbf{v}^i_0, \mathbf{v}^i_{t-1}, \mathbf{v}^i_t)}{\tau(\mathbf{v}^{*i}, \mathbf{v}^i_{t-1}, \mathbf{v}^i_t)} (\bar{R}_t)^{\bar{\omega}(\mathbf{v}^*, \mathbf{v}_t) - \bar{\omega}(\mathbf{v}_0, \mathbf{v}_t)}} \right) \tag{114}$$

Denote $Z^i_t(\mathbf{v}_0; \mathbf{v}^*, \mathbf{v}_t) = (\bar{R}_t)^{-(\bar{\omega}_{-i}(\mathbf{v}_0, \mathbf{v}_t) - \bar{\omega}_{-i}(\mathbf{v}^*, \mathbf{v}_t))}$ where $\bar{\omega}_{-i}(\mathbf{v}, \tilde{\mathbf{v}}) = \sum_{j=1, j \neq i}^n 1_{\mathbf{v}^j \neq \tilde{\mathbf{v}}^j} = \bar{\omega}(\mathbf{v}, \tilde{\mathbf{v}}) - 1_{\mathbf{v}^i \neq \mathbf{v}^i}$. $Z^i_t(\mathbf{v}_0; \mathbf{v}^*, \mathbf{v}_t) = (\bar{R}_t)^{-(\bar{\omega}(\mathbf{v}_0, \mathbf{v}_t) - \bar{\omega}(\mathbf{v}^*, \mathbf{v}_t))} \cdot \bar{R}_t^{-1_{\mathbf{v}^{*i} \neq \mathbf{v}^i_t} + 1_{(\mathbf{v}_0)^i \neq \mathbf{v}^i_t}}$. For simplicity, we shorthand $Z^i_t(\mathbf{v}_0; \mathbf{v}^*, \mathbf{v}_t)$ as $Z^i_t(\mathbf{v}_0)$. We have

$$\log \frac{(s+1) \cdot q(\mathbf{v}^i_{t-1|0} = \mathbf{v}^i_{t-1}, \mathbf{v}_{t|0} = \mathbf{v}_t)}{s \cdot q(\mathbf{v}^i_{t-1|1} = \mathbf{v}^i_{t-1}, \mathbf{v}_{t|1} = \mathbf{v}_t)} \tag{115}$$

$$= \log \left( 1 + \frac{(s+1) \cdot q(\mathbf{v}^i_{t-1|0} = \mathbf{v}^i_{t-1}, \mathbf{v}_{t|0} = \mathbf{v}_t) - s \cdot q(\mathbf{v}^i_{t-1|1} = \mathbf{v}^i_{t-1}, \mathbf{v}_{t|1} = \mathbf{v}_t)}{s \cdot q(\mathbf{v}^i_{t-1|1} = \mathbf{v}^i_{t-1}, \mathbf{v}_{t|1} = \mathbf{v}_t)} \right) \tag{116}$$

$$= \log \left( 1 + \frac{1}{\sum_{\mathbf{v}_0 \in \mathcal{V}_1} \bar{\kappa}(\mathbf{v}^{*i}, \mathbf{v}^i_0, \mathbf{v}^i_{t-1}, \mathbf{v}^i_t) Z^i_t(\mathbf{v}_0)} \right) \tag{117}$$

$$= \log \left( 1 + \frac{1}{\sum_{\mathbf{v}_0 \in \mathcal{V}_1, \mathbf{v}^i_0 = \mathbf{v}^i_{t-1}} \bar{\kappa}(\mathbf{v}^{*i}, \mathbf{v}^i_0, \mathbf{v}^i_{t-1}, \mathbf{v}^i_t) Z^i_t(\mathbf{v}_0) + \sum_{\mathbf{v}_0 \in \mathcal{V}_1, \mathbf{v}^i_0 \neq \mathbf{v}^i_{t-1}} \bar{\kappa}(\mathbf{v}^{*i}, \mathbf{v}^i_0, \mathbf{v}^i_{t-1}, \mathbf{v}^i_t) Z^i_t(\mathbf{v}_0)} \right) \tag{118}$$

Observed that if $\mathbf{v}_{t-1}^i = \mathbf{v}^{*i}$, $\bar{\kappa}(\mathbf{v}^{*i}, \mathbf{v}_0^i, \mathbf{v}_{t-1}^i, \mathbf{v}_t^i) = 1$ or $\bar{R}_{t-1}^{-1}$. If $\mathbf{v}_{t-1}^i \neq \mathbf{v}^*$, $\bar{\kappa}(\mathbf{v}^{*i}, \mathbf{v}_0^i, \mathbf{v}_{t-1}^i, \mathbf{v}_t^i) = 1$ or $\bar{R}_{t-1}$. According to Appendix L, we have the following qualities for the maximum and minimum of $\log \frac{(s+1) \cdot q(\mathbf{v}_{t-1|0}^i = \mathbf{v}_{t-1}^i, \mathbf{v}_{t|0} = \mathbf{v}_t)}{s \cdot q(\mathbf{v}_{t-1|1}^i = \mathbf{v}_{t-1}^i, \mathbf{v}_{t|1} = \mathbf{v}_t)}$.

$$\max_{\mathbf{v}_{t-1}^i \in [k]} \log \frac{(s+1) \cdot q(\mathbf{v}_{t-1|0}^i = \mathbf{v}_{t-1}^i, \mathbf{v}_{t|0} = \mathbf{v}_t)}{s \cdot q(\mathbf{v}_{t-1|1}^i = \mathbf{v}_{t-1}^i, \mathbf{v}_{t|1} = \mathbf{v}_t)} = \log \frac{(s+1) \cdot q(\mathbf{v}_{t-1|0}^i = \mathbf{v}^{*i}, \mathbf{v}_{t|0} = \mathbf{v}_t)}{s \cdot q(\mathbf{v}_{t-1|1}^i = \mathbf{v}^{*i}, \mathbf{v}_{t|1} = \mathbf{v}_t)} \tag{119}$$

$$= \log \left( 1 + \frac{1}{\sum_{\mathbf{v}_0 \in \mathcal{V}_1, \mathbf{v}_0^i = \mathbf{v}^{*i}} Z_t^i(\mathbf{v}_0) + \bar{R}_{t-1}^{-1} \sum_{\mathbf{v}_0 \in \mathcal{V}_1, \mathbf{v}_0^i \neq \mathbf{v}^{*i}} Z_t^i(\mathbf{v}_0)} \right) \tag{120}$$

$$\min_{\mathbf{v}_{t-1}^i \in [k]} \log \frac{(s+1) \cdot q(\mathbf{v}_{t-1|0}^i = \mathbf{v}_{t-1}^i, \mathbf{v}_{t|0} = \mathbf{v}_t)}{s \cdot q(\mathbf{v}_{t-1|1}^i = \mathbf{v}_{t-1}^i, \mathbf{v}_{t|1} = \mathbf{v}_t)} = \log \frac{(s+1) \cdot q(\mathbf{v}_{t-1|0}^i = \mathbf{v}_{t-1}^{i*}, \mathbf{v}_{t|0} = \mathbf{v}_t)}{s \cdot q(\mathbf{v}_{t-1|1}^i = \mathbf{v}_{t-1}^{i*}, \mathbf{v}_{t|1} = \mathbf{v}_t)} \tag{121}$$

$$= \log \left( 1 + \frac{1}{\bar{R}_{t-1} \sum_{\mathbf{v}_0 \in \mathcal{V}_1, \mathbf{v}_0^i = \mathbf{v}_{t-1}^{i*}} Z_t^i(\mathbf{v}_0) + \sum_{\mathbf{v}_0 \in \mathcal{V}_1, \mathbf{v}_0^i \neq \mathbf{v}_{t-1}^{i*}} Z_t^i(\mathbf{v}_0)} \right) \tag{122}$$

where $\mathbf{v}_{t-1}^{i*} = \operatorname{argmax}_{\mathbf{v}_{t-1}^i \in [k] \setminus \{\mathbf{v}^{*i}\}} \sum_{\mathbf{v}_0 \in \mathcal{V}_1, \mathbf{v}_0^i = \mathbf{v}_{t-1}^{i*}} Z_t^i(\mathbf{v}_0)$.

Now, we consider $\max_{\mathbf{v}_{t-1}^i} \log \frac{(s+1) \cdot q(\mathbf{v}_{t-1|0}^i = \mathbf{v}_{t-1}^i, \mathbf{v}_{t|0})}{s \cdot q(\mathbf{v}_{t-1|1}^i = \mathbf{v}_{t-1}^i, \mathbf{v}_{t|1})} - \min_{\mathbf{v}_{t-1}^i} \log \frac{(s+1) \cdot q(\mathbf{v}_{t-1|0}^i = \mathbf{v}_{t-1}^i, \mathbf{v}_{t|0})}{s \cdot q(\mathbf{v}_{t-1|1}^i = \mathbf{v}_{t-1}^i, \mathbf{v}_{t|1})}$:

$$\max_{\mathbf{v}_{t-1}^i} \log \frac{(s+1) \cdot q(\mathbf{v}_{t-1|0}^i = \mathbf{v}_{t-1}^i, \mathbf{v}_{t|0})}{s \cdot q(\mathbf{v}_{t-1|1}^i = \mathbf{v}_{t-1}^i, \mathbf{v}_{t|1})} - \min_{\mathbf{v}_{t-1}^i} \log \frac{(s+1) \cdot q(\mathbf{v}_{t-1|0}^i = \mathbf{v}_{t-1}^i, \mathbf{v}_{t|0})}{s \cdot q(\mathbf{v}_{t-1|1}^i = \mathbf{v}_{t-1}^i, \mathbf{v}_{t|1})} \tag{123}$$

$$= \log \left( 1 + \frac{1}{\sum_{\mathbf{v}_0 \in \mathcal{V}_1, \mathbf{v}_0^i = \mathbf{v}^{*i}} Z_t^i(\mathbf{v}_0) + \bar{R}_{t-1}^{-1} \sum_{\mathbf{v}_0 \in \mathcal{V}_1, \mathbf{v}_0^i \neq \mathbf{v}^{*i}} Z_t^i(\mathbf{v}_0)} \right)$$
$$- \log \left( 1 + \frac{1}{\bar{R}_{t-1} \sum_{\mathbf{v}_0 \in \mathcal{V}_1, \mathbf{v}_0^i = \mathbf{v}_{t-1}^{i*}} Z_t^i(\mathbf{v}_0) + \sum_{\mathbf{v}_0 \in \mathcal{V}_1, \mathbf{v}_0^i \neq \mathbf{v}_{t-1}^{i*}} Z_t^i(\mathbf{v}_0)} \right) \tag{124}$$

$$= \log \left( \frac{1 + \frac{1}{\sum_{\mathbf{v}_0 \in \mathcal{V}_1, \mathbf{v}_0^i = \mathbf{v}^{*i}} Z_t^i(\mathbf{v}_0) + \bar{R}_{t-1}^{-1} \sum_{\mathbf{v}_0 \in \mathcal{V}_1, \mathbf{v}_0^i \neq \mathbf{v}^{*i}} Z_t^i(\mathbf{v}_0)}}{1 + \frac{1}{\bar{R}_{t-1} \sum_{\mathbf{v}_0 \in \mathcal{V}_1, \mathbf{v}_0^i = \mathbf{v}_{t-1}^{i*}} Z_t^i(\mathbf{v}_0) + \sum_{\mathbf{v}_0 \in \mathcal{V}_1, \mathbf{v}_0^i \neq \mathbf{v}_{t-1}^{i*}} Z_t^i(\mathbf{v}_0)}} \right) \tag{125}$$

$$\overset{(i)}{=} \log \left( 1 + \frac{\bar{R}_{t-1} \sum_{\mathbf{v}_0^i = \mathbf{v}_{t-1}^{i*}} Z_t^i(\mathbf{v}_0) + \sum_{\mathbf{v}_0^i \neq \mathbf{v}_{t-1}^{i*}} Z_t^i(\mathbf{v}_0) - \sum_{\mathbf{v}_0^i = \mathbf{v}^{*i}} Z_t^i(\mathbf{v}_0) - \bar{R}_{t-1}^{-1} \sum_{\mathbf{v}_0^i \neq \mathbf{v}^{*i}} Z_t^i(\mathbf{v}_0)}{(\sum_{\mathbf{v}_0^i = \mathbf{v}^{*i}} Z_t^i(\mathbf{v}_0) + \bar{R}_{t-1}^{-1} \sum_{\mathbf{v}_0^i \neq \mathbf{v}^{*i}} Z_t^i(\mathbf{v}_0))(1 + \bar{R}_{t-1} \sum_{\mathbf{v}_0^i = \mathbf{v}_{t-1}^{i*}} Z_t^i(\mathbf{v}_0) + \sum_{\mathbf{v}_0^i \neq \mathbf{v}_{t-1}^{i*}} Z_t^i(\mathbf{v}_0))} \right) \tag{126}$$

where in $(i)$, we have omitted the summation constraint $\mathbf{v}_0 \in \mathcal{V}_1$ for brevity.

Further, we split $\sum_{\mathbf{v}_0^i \neq \mathbf{v}_{t-1}^{i*}} Z_t^i(\mathbf{v}_0), \sum_{\mathbf{v}_0^i \neq \mathbf{v}^{*i}} Z_t^i(\mathbf{v}_0)$ two terms as follows:

$$\sum_{\mathbf{v}_0^i \neq \mathbf{v}_{t-1}^{i*}} Z_t^i(\mathbf{v}_0) = \sum_{\mathbf{v}_0^i = \mathbf{v}^{*i}} Z_t^i(\mathbf{v}_0) + \sum_{\mathbf{v}_0^i \neq \mathbf{v}_{t-1}^{i*}, \mathbf{v}^{*i}} Z_t^i(\mathbf{v}_0) \tag{127}$$

$$\sum_{\mathbf{v}_0^i \neq \mathbf{v}^{*i}} Z_t^i(\mathbf{v}_0) = \sum_{\mathbf{v}_0^i = \mathbf{v}_{t-1}^{i*}} Z_t^i(\mathbf{v}_0) + \sum_{\mathbf{v}_0^i \neq \mathbf{v}_{t-1}^{i*}, \mathbf{v}^{*i}} Z_t^i(\mathbf{v}_0) \tag{128}$$

Plugging back the split terms into Eq. (126), we get

$$\max_{\mathbf{v}_{t-1}^i} \log \frac{(s+1) \cdot q(\mathbf{v}_{t-1|0}^i = \mathbf{v}_{t-1}^i, \mathbf{v}_{t|0})}{s \cdot q(\mathbf{v}_{t-1|1}^i = \mathbf{v}_{t-1}^i, \mathbf{v}_{t|1})} - \min_{\mathbf{v}_{t-1}^i} \log \frac{(s+1) \cdot q(\mathbf{v}_{t-1|0}^i = \mathbf{v}_{t-1}^i, \mathbf{v}_{t|0})}{s \cdot q(\mathbf{v}_{t-1|1}^i = \mathbf{v}_{t-1}^i, \mathbf{v}_{t|1})} \tag{129}$$

$$= \log \left( 1 + \frac{(a-1)[(a+1)y+z]}{(ax+y+z)(ay+x+z+1)} \right) \tag{130}$$

$$\overset{(i)}{\leq} \log\left(1 + \frac{(a-1)(a+1)}{(a^2+1)x + ay + az + 1}\right) \tag{131}$$

$$= \log(1 + \frac{(\bar{R}_{t-1}-1)(\bar{R}_{t-1}+1)}{(\bar{R}_{t-1}^2+1)\sum_{\mathbf{v}_0^i=\mathbf{v}^{*i}} Z_t^i(\mathbf{v}_0) + \bar{R}_{t-1}\sum_{\mathbf{v}_0^i=\mathbf{v}_{t-1}^{i*}} Z_t^i(\mathbf{v}_0) + \bar{R}_{t-1}\sum_{\mathbf{v}_0^i\neq\mathbf{v}_{t-1}^{i*},\mathbf{v}^{*i}} Z_t^i(\mathbf{v}_0) + 1}) \tag{132}$$

$$\overset{(ii)}{\leq} \log(1 + \frac{(\bar{R}_{t-1}-1)(\bar{R}_{t-1}+1)}{(\bar{R}_{t-1}^2+1)\cdot\zeta(\mathcal{V}_1^{i|\mathbf{v}^{*i}},\mathbf{v}^*,\mathbf{v}_t,t) + \bar{R}_{t-1}/\bar{R}_t\cdot\zeta(\mathcal{V}_1\backslash\mathcal{V}_1^{i|\mathbf{v}^{*i}},\mathbf{v}^*,\mathbf{v}_t,t) + 1}) \tag{133}$$

$$\leq \log(1 + \frac{(\bar{R}_{t-1}-1)(\bar{R}_{t-1}+1)}{(\bar{R}_{t-1}^2+1)\cdot\zeta(\mathcal{V}_1^{i|\mathbf{v}^{*i}},\mathbf{v}^*,\mathbf{v}_t,t) + \bar{R}_{t-1}/\bar{R}_t\cdot\zeta(\mathcal{V}_1,\mathbf{v}^*,\mathbf{v}_t,t) + 1}) \tag{134}$$

where we define $a = \bar{R}_{t-1}, x = \sum_{\mathbf{v}_0\in\mathcal{V}_1,\mathbf{v}_0^i=\mathbf{v}^{*i}} Z_t^i(\mathbf{v}_0), y = \sum_{\mathbf{v}_0\in\mathcal{V}_1,\mathbf{v}_0^i=\mathbf{v}_{t-1}^{i*}} Z_t^i(\mathbf{v}_0), z = \sum_{\mathbf{v}_0\in\mathcal{V}_1,\mathbf{v}_0^i\neq\mathbf{v}^{*i},\mathbf{v}_{t-1}^{i*}} Z_t^i(\mathbf{v}_0)$, and $(i)$ is from Lemma K.5, $(ii)$ is from the definition of $Z_t^i(\mathbf{v}_0)$.

Further, we consider $\|q(\mathbf{v}_{t-1|0}^i|\mathbf{v}_{t|0}=\mathbf{v}_t) - q(\mathbf{v}_{t-1|1}^i|\mathbf{v}_{t|1}=\mathbf{v}_t)\|_{l_1}$

$$\|q(\mathbf{v}_{t-1|0}^i|\mathbf{v}_{t|0}=\mathbf{v}_t) - q(\mathbf{v}_{t-1|1}^i|\mathbf{v}_{t|1}=\mathbf{v}_t)\|_{l_1} \tag{135}$$

$$= \sum_{\mathbf{v}_{t-1}^i=1}^k \left| \frac{(\tau(\mathbf{v}^{*i},\mathbf{v}_{t-1}^i,\mathbf{v}_t^i)(\bar{R}_t)^{-\bar{\omega}(\mathbf{v}^*,\mathbf{v}_t)})(\sum_{\mathbf{v}_0\in\mathcal{V}_1}(\bar{R}_t)^{-\bar{\omega}(\mathbf{v}_0,\mathbf{v}_t)})}{(\sum_{\mathbf{v}_0\in\mathcal{V}_1}(\bar{R}_t)^{-\bar{\omega}(\mathbf{v}_0,\mathbf{v}_t)}))(\sum_{\mathbf{v}_0\in\mathcal{V}_0}(\bar{R}_t)^{-\bar{\omega}(\mathbf{v}_0,\mathbf{v}_t)}))} \right.$$

$$\left. - \frac{((\bar{R}_t)^{-\bar{\omega}(\mathbf{v}^*,\mathbf{v}_t)})(\sum_{\mathbf{v}_0\in\mathcal{V}_1}\tau(\mathbf{v}_0^i,\mathbf{v}_{t-1}^i,\mathbf{v}_t^i)(\bar{R}_t)^{-\bar{\omega}(\mathbf{v}_0,\mathbf{v}_t)})}{(\sum_{\mathbf{v}_0\in\mathcal{V}_1}(\bar{R}_t)^{-\bar{\omega}(\mathbf{v}_0,\mathbf{v}_t)})(\sum_{\mathbf{v}_0\in\mathcal{V}_0}(\bar{R}_t)^{-\bar{\omega}(\mathbf{v}_0,\mathbf{v}_t)})} \right| \tag{136}$$

$$= \sum_{\mathbf{v}_{t-1}^i=1}^k \left| \frac{\sum_{\mathbf{v}_0\in\mathcal{V}_1}\tau(\mathbf{v}^{*i},\mathbf{v}_{t-1}^i,\mathbf{v}_t^i)\cdot(1-\frac{\tau(\mathbf{v}_0^i,\mathbf{v}_{t-1}^i,\mathbf{v}_t^i)}{\tau(\mathbf{v}^{*i},\mathbf{v}_{t-1}^i,\mathbf{v}_t^i)})(\bar{R}_t)^{\bar{\omega}(\mathbf{v}^*,\mathbf{v}_t)-\bar{\omega}(\mathbf{v}_0,\mathbf{v}_t)}}{(\sum_{\mathbf{v}_0\in\mathcal{V}_1}(\bar{R}_t)^{\bar{\omega}(\mathbf{v}^*,\mathbf{v}_t)-\bar{\omega}(\mathbf{v}_0,\mathbf{v}_t)})(1+\sum_{\mathbf{v}_0\in\mathcal{V}_1}(\bar{R}_t)^{\bar{\omega}(\mathbf{v}^*,\mathbf{v}_t)-\bar{\omega}(\mathbf{v}_0,\mathbf{v}_t)})} \right| \tag{137}$$

$$\overset{(i)}{\leq} \mu_t^+ \cdot (\frac{\bar{\mu}_{t-1}^+}{\bar{\mu}_t^+} - \frac{\bar{\mu}_{t-1}^-}{\bar{\mu}_t^-}) \cdot \frac{1}{1+\sum_{\mathbf{v}_0\in\mathcal{V}_1}(\bar{R}_t)^{\bar{\omega}(\mathbf{v}^*,\mathbf{v}_t)-\bar{\omega}(\mathbf{v}_0,\mathbf{v}_t)}} \tag{138}$$

where $(i)$ is from Lemma K.6.

Therefore, summarizing the above, we obtain

$$\mathcal{D}_{\mathrm{KL}}(q(\mathbf{v}_{t-1|0}^i|\mathbf{v}_{t|0})\|q(\mathbf{v}_{t-1|1}^i|\mathbf{v}_{t|1})) + \mathcal{D}_{\mathrm{KL}}(q(\mathbf{v}_{t-1|1}^i|\mathbf{v}_{t|1})\|q(\mathbf{v}_{t-1|0}^i|\mathbf{v}_{t|0})) \tag{139}$$

$$\leq \frac{1}{2}\|q(\mathbf{v}_{t-1|0}^i=\mathbf{v}_{t-1}^i|\mathbf{v}_{t|0}) - q(\mathbf{v}_{t-1|1}^i=\mathbf{v}_{t-1}^i|\mathbf{v}_{t|1})\|_{l_1} \left[ \max_{\mathbf{v}_{t-1}^i} \log\frac{q(\mathbf{v}_{t-1|0}^i=\mathbf{v}_{t-1}^i,\mathbf{v}_{t|0})}{q(\mathbf{v}_{t-1|1}^i=\mathbf{v}_{t-1}^i,\mathbf{v}_{t|1})} \right.$$

$$\left. - \min_{\mathbf{v}_{t-1}^i} \log\frac{q(\mathbf{v}_{t-1|0}^i=\mathbf{v}_{t-1}^i,\mathbf{v}_{t|0})}{q(\mathbf{v}_{t-1|1}^i=\mathbf{v}_{t-1}^i,\mathbf{v}_{t|1})} \right] \tag{140}$$

$$\leq \frac{\mu_t^+ \cdot (\frac{\bar{\mu}_{t-1}^+}{\bar{\mu}_t^+} - \frac{\bar{\mu}_{t-1}^-}{\bar{\mu}_t^-})}{1+\zeta(\mathcal{V}_1,\mathbf{v}^*,\mathbf{v}_t,t)} \cdot \log\left(1 + \frac{(\bar{R}_{t-1}-1)(\bar{R}_{t-1}+1)}{(\bar{R}_{t-1}^2+1)\cdot\zeta(\mathcal{V}_1^{i|\mathbf{v}^{*i}},\mathbf{v}^*,\mathbf{v}_t,t) + \bar{R}_{t-1}/\bar{R}_t\cdot\zeta(\mathcal{V}_1,\mathbf{v}^*,\mathbf{v}_t,t) + 1}\right) \tag{141}$$

Define $\Delta_t^i(\cdot)$ as:

$$\Delta_t^i(\mathbf{v}_t) = \frac{\mu_t^+ \cdot (\frac{\bar{\mu}_{t-1}^+}{\bar{\mu}_t^+} - \frac{\bar{\mu}_{t-1}^-}{\bar{\mu}_t^-})}{1+\zeta(\mathcal{V}_1,\mathbf{v}^*,\mathbf{v}_t,t)} \cdot \log\left(1 + \frac{(\bar{R}_{t-1}-1)(\bar{R}_{t-1}+1)}{(\bar{R}_{t-1}^2+1)\cdot\zeta(\mathcal{V}_1^{i|\mathbf{v}^{*i}},\mathbf{v}^*,\mathbf{v}_t,t) + \bar{R}_{t-1}/\bar{R}_t\cdot\zeta(\mathcal{V}_1,\mathbf{v}^*,\mathbf{v}_t,t) + 1}\right) \tag{142}$$

We consider a upper bound on $\sum_{i=1}^n \Delta_t^i$. Let $C_t = \frac{\mu_t^+ \cdot (\frac{\bar{\mu}_{t-1}^+}{\bar{\mu}_t^+} - \frac{\bar{\mu}_{t-1}^-}{\bar{\mu}_t^-})}{1+\zeta(\mathcal{V}_1,\mathbf{v}^*,\mathbf{v}_t,t)}$, we have

$$\sum_{i=1}^n \Delta_t^i = C_t \sum_{i=1}^n \log\left(1 + \frac{(\bar{R}_{t-1}-1)(\bar{R}_{t-1}+1)}{(\bar{R}_{t-1}^2+1)\cdot\zeta(\mathcal{V}_1^{i|\mathbf{v}^{*i}},\mathbf{v}^*,\mathbf{v}_t,t) + \bar{R}_{t-1}/\bar{R}_t\cdot\zeta(\mathcal{V}_1,\mathbf{v}^*,\mathbf{v}_t,t) + 1}\right) \tag{143}$$

$$=: \Delta_t \tag{144}$$

From above, we have,

$$\sum_{i=1}^{n} \mathcal{D}_{\mathrm{KL}}(q(\mathbf{v}_{t-1|0}^i|\mathbf{v}_{t|0})\|q(\mathbf{v}_{t-1|1}^i|\mathbf{v}_{t|1})) + \mathcal{D}_{\mathrm{KL}}(q(\mathbf{v}_{t-1|1}^i|\mathbf{v}_{t|1})\|q(\mathbf{v}_{t-1|0}^i|\mathbf{v}_{t|0})) \le \Delta_t(\mathbf{v}_t) \tag{145}$$

$\square$

### J.5 Proof of Lemma A.5

According to measure partition, we have split measure $q(\mathbf{v}_{t|0})$ into $q(\mathbf{v}_t \in \mathcal{S}_a)$, $q(\mathbf{v}_t \in \mathcal{S}_b)$ and $q(\mathbf{v}_t \in \mathcal{S}_c)$. Therefore,

$$\mathbb{E}_{\mathbf{v}_t \sim q(\mathbf{v}_{t|0})}[\Delta_t(\mathbf{v}_t)] = q(\mathbf{v}_t \in \mathcal{S}_a) \cdot \Delta_t(\mathbf{v}_t) + q(\mathbf{v}_t \in \mathcal{S}_b) \cdot \Delta_t(\mathbf{v}_t) + q(\mathbf{v}_t \in \mathcal{S}_c) \cdot \Delta_t(\mathbf{v}_t) \tag{146}$$

Now, we consider the behavior of $\Delta_t(\cdot)$ with respect to $\mathbf{v}_t$ for each measure. For measure $q(\mathbf{v}_t \in \mathcal{S}_a)$ and $q(\mathbf{v}_t \in \mathcal{S}_c)$, since $\zeta(\mathcal{V}_1, \mathbf{v}^*, \mathbf{v}_t, t) \ge \mathrm{Sim}(\mathcal{V}_1, \mathbf{v}^*, t)$, we have

$$\Delta_t(\mathbf{v}_t) \le \frac{\mu_t^+ (\bar{\mu}_{t-1}^+/\bar{\mu}_t^+ - \bar{\mu}_{t-1}^-/\bar{\mu}_t^-)}{1 + \mathrm{Sim}(\mathcal{V}_1, \mathbf{v}^*, t)} \cdot \sum_{i=1}^{n} \log(1 + \frac{\bar{R}_{t-1}^2 - 1}{\bar{R}_{t-1}^2 \cdot \mathrm{Sim}(\mathcal{V}_1^{i|\mathbf{v}^{*i}}, \mathbf{v}^*, t) + \mathrm{Sim}(\mathcal{V}_1, \mathbf{v}^*, t) + 1}) \tag{147}$$

$$\stackrel{\mathrm{def}}{=} \frac{n}{s^{\psi_t}} \tag{148}$$

where $\mathcal{A}_t = \mu_t^+ \cdot (\bar{\mu}_{t-1}^+/\bar{\mu}_t^+ - \bar{\mu}_{t-1}^-/\bar{\mu}_t^-)$ and $\mathcal{B}_t = \bar{R}_{t-1}^2 - 1$. Here, we use notation $\frac{1}{s^{\psi_t}}$ instead of a single symbol is to clearly show how the order of privacy leakage varies with respect to $t$ in generation process. For measure $q(\mathbf{v}_t \in \mathcal{S}_b)$, we have much smaller privacy leakage than $q(\mathbf{v}_t \in \mathcal{S}_a)$ and $q(\mathbf{v}_t \in \mathcal{S}_c)$. When the $\mathrm{supp}(\mathcal{V}_0) = \mathrm{supp}(\mathcal{V}_1)$, $\mathcal{V}_1^{i|\mathbf{v}^{*i}} \ne \emptyset$, we have

$$\Delta_t(\mathbf{v}_t) \le \frac{\mathcal{A}_t}{1 + \bar{R}_t^{\eta_t'}} \cdot \sum_{i=1}^{n} \log\left(1 + \frac{\mathcal{B}_t}{1 + \bar{R}_{t-1}^2 \cdot \bar{R}_t^{\eta_t'}}\right) \tag{149}$$

$$\le \frac{n \cdot \mathcal{A}_t \cdot \mathcal{B}_t}{(1 + \bar{R}_t^{\eta_t'})(1 + \bar{R}_{t-1}^2 \cdot \bar{R}_t^{\eta_t'})} \le \frac{n \cdot \mathcal{A}_t \cdot \mathcal{B}_t}{\bar{R}_{t-1}^2 \cdot \bar{R}_t^{2\eta_t'}} \tag{150}$$

Therefore,

$$\mathbb{E}_{\mathbf{v}_t}[\Delta_t(\mathbf{v}_t)] \le q(\mathbf{v}_t \in \mathcal{S}_a) \cdot \frac{n}{s^{\psi_t}} + q(\mathbf{v}_t \in \mathcal{S}_b) \cdot \frac{n \cdot \mathcal{A}_t \cdot \mathcal{B}_t}{\bar{R}_{t-1}^2 \cdot \bar{R}_t^{2\eta_t'}} + q(\mathbf{v}_t \in \mathcal{S}_c) \cdot \frac{n}{s^{\psi_t}} \tag{151}$$

Now, based on the above partition, we further estimate the measure of $q(\mathbf{v}_t \in \mathcal{S}_a)$, $q(\mathbf{v}_t \in \mathcal{S}_b)$ and $q(\mathbf{v}_t \in \mathcal{S}_c)$. Recall that $N_{\eta_t}(\mathbf{v}^*) = |\{\mathbf{v} \in \mathcal{V}_1 \text{ s.t. } \bar{\omega}(\mathbf{v}, \mathbf{v}^*) \le \eta_t\}|$. We have the following:

$$q(\mathbf{v}_t \in \mathcal{S}_b) \le q(\mathbf{v}_t; (1)\mathbf{v}_t \text{ is diffused from } \mathbf{v}_0 \in \mathcal{V}_0 \text{ s.t. } \bar{\omega}(\mathbf{v}', \mathbf{v}^*) > \eta_t) \tag{152}$$

$$\le \frac{s - N_{\eta_t}(\mathbf{v}^*)}{s} \tag{153}$$

$$q(\mathbf{v}_t \in \mathcal{S}_c) \le q(\mathbf{v}_t; (1)\mathbf{v}_t \text{ is diffused from } \mathbf{v}_0 \in \mathcal{V}_0 \text{ s.t } \bar{\omega}(\mathbf{v}_0, \mathbf{v}^*) \ge \eta_t + 1 \text{ and}$$
$$(2)\forall \mathbf{v}_0' \in \mathcal{V}_0, \bar{\omega}(\mathbf{v}_t, \mathbf{v}_0') \ge \eta_t - \eta_t' + 2) \tag{154}$$

$$\le q(\mathbf{v}_t; \mathbf{v}_t \text{ is diffused from } \mathbf{v}_0 \in \mathcal{V}_0 \text{ s.t. } \bar{\omega}(\mathbf{v}_0, \mathbf{v}^*) \ge \eta_t + 1, \bar{\omega}(\mathbf{v}_t, \mathbf{v}_0) \ge \eta_t - \eta_t' + 2) \tag{155}$$

$$\le \frac{s - N_{\eta_t}(\mathbf{v}^*)}{s} \cdot \min\{(n(1 - \bar{\mu}_t^+))^{\eta_t - \eta_t' + 2}, 1\} \tag{156}$$

As for $q(\mathbf{v}_t \in \mathcal{S}_a)$, we need further characterization. Let $c_t^*$ be a tunable positive constant and we have the following:

$$q(\mathbf{v}_t \in \mathcal{S}_a) \le q(\mathbf{v}_t; \mathbf{v}_t \text{ is diffused from } \mathbf{v}_0 \in \mathcal{V}_0 \text{ s.t. } \bar{\omega}(\mathbf{v}_0, \mathbf{v}^*) \le \eta_t)$$

$$+q(\mathbf{v}_t; \mathbf{v}_t \text{ is diffused from } \mathbf{v}_0 \in \mathcal{V}_0 \text{ s.t. } \bar{\omega}(\mathbf{v}_0, \mathbf{v}^*) \in [\eta_t + 1, (1 + c_t^*)\eta_t], \bar{\omega}(\mathbf{v}_t, \mathbf{v}^*) \le \eta_t)$$
$$+q(\mathbf{v}_t; \mathbf{v}_t \text{ is diffused from } \mathbf{v}_0 \in \mathcal{V}_0 \text{ s.t. } \bar{\omega}(\mathbf{v}_0, \mathbf{v}^*) \ge (1 + c_t^*)\eta_t + 1, \bar{\omega}(\mathbf{v}_t, \mathbf{v}^*) \le \eta_t) \tag{157}$$

$$\le \underbrace{\frac{N_{\eta_t}(\mathbf{v}^*)}{s} + \sum_{\eta=\eta_t+1}^{(1+c_t^*)\eta_t} \frac{\Delta N_\eta(\mathbf{v}^*)}{s} \binom{n}{\eta-\eta_t}(\bar{\mu}_t^-)^{\eta-\eta_t}}_{\text{merge together}} + \frac{s - N_{(1+c_t^*)\eta_t}(\mathbf{v}^*)}{s} \max_{k;k\ge\eta_t} \binom{k}{k-\eta_t}(\bar{\mu}_t^-)^{k-\eta_t} \tag{158}$$

$$\le \frac{N_{(1+c_t^*)\eta_t}(\mathbf{v}^*)}{s} + \frac{s - N_{(1+c_t^*)\eta_t}(\mathbf{v}^*)}{s} \max_{h;h\ge(1+c_t^*)\eta_t} \binom{h}{h-\eta_t}(\bar{\mu}_t^-)^{h-\eta_t} \tag{159}$$

$$\overset{(i)}{\le} \frac{N_{(1+c_t^*)\eta_t}(\mathbf{v}^*)}{s} + \frac{s - N_{(1+c_t^*)\eta_t}(\mathbf{v}^*)}{s} \min \max_{h;h\ge(1+c_t^*)\eta_t} \left\{ \left[\frac{heq_t}{h-\eta_t}\right]^{h-\eta_t}, \left[\frac{he}{\eta_t}\right]^{\eta_t} q_t^{h-\eta_t} \right\} \tag{160}$$

$$\overset{(ii)}{\le} \frac{N_{(1+c_t^*)\eta_t}(\mathbf{v}^*)}{s}$$
$$+ \frac{s - N_{(1+c_t^*)\eta_t}(\mathbf{v}^*)}{s} \left[ ((1+c_t^*)e)^{\eta_t}(\bar{\mu}_t^-)^{c_t^*\eta_t} \mathbb{1}_{c_t^*<1} + \left(\frac{(1+c_t^*)e}{c_t^*}\right)^{c_t^*\eta_t}(\bar{\mu}_t^-)^{c_t^*\eta_t} \mathbb{1}_{c_t^*\ge1} \right] \tag{161}$$

$(i)$ is from $\binom{n}{k} \le (\frac{en}{k})^k$ and $(ii)$ is from Corollary K.1 where we need $e\bar{\mu}_t^- < \frac{4}{5}$ (i.e. $k \ge 3$).

Now, we want further balance the two terms $\frac{N_{(1+c_t^*)\eta_t}(\mathbf{v}^*)}{s}$ and $\frac{s-N_{(1+c_t^*)\eta_t}(\mathbf{v}^*)}{s}\left[((1+c_t^*)e)^{\eta_t}(\bar{\mu}_t^-)^{c_t^*\eta_t}\mathbb{1}_{c_t^*<1} + \left(\frac{(1+c_t^*)e}{c_t^*}\right)^{c_t^*\eta_t}(\bar{\mu}_t^-)^{c_t^*\eta_t}\mathbb{1}_{c_t^*\ge1}\right]$, i.e. balancing $\frac{N_{(1+c_t^*)\eta_t}(\mathbf{v}^*)}{s-N_{(1+c_t^*)\eta_t}(\mathbf{v}^*)}$ and $\left[((1+c_t^*)e)_t^{\eta}(\bar{\mu}_t^-)^{c_t^*\eta_t}\mathbb{1}_{c_t^*<1} + \left(\frac{(1+c_t^*)e}{c_t^*}\right)^{c_t^*\eta_t}(\bar{\mu}_t^-)^{c_t^*\eta_t}\mathbb{1}_{c_t^*\ge1}\right]$. First, compare $\frac{N_{2\eta_t}(\mathbf{v}^*)}{s-N_{2\eta_t}(\mathbf{v}^*)}$ and $(2e\bar{\mu}_t^-)^{\eta_t}$.

(1) If $\frac{N_{2\eta_t}(\mathbf{v}^*)}{s-N_{2\eta_t}(\mathbf{v}^*)} > (2e\bar{\mu}_t^-)^{\eta_t}$, we can find a smaller $c_t^*$ to make the bound Eq. (161) tight. Instead of working directly on $((1+c_t^*)e)^{\eta_t}(\bar{\mu}_t^-)^{c_t^*\eta_t} \le \frac{N_{(1+c_t^*)\eta_t}(\mathbf{v}^*)}{s-N_{(1+c_t^*)\eta_t}(\mathbf{v}^*)}$, we consider a sufficient condition: found smallest $c_t^*$ such that

$$(2e)^{\eta_t}(\bar{\mu}_t^-)^{c_t^*\eta_t} \le \frac{N_{(1+c_t^*)\eta_t}(\mathbf{v}^*)}{s-N_{(1+c_t^*)\eta_t}(\mathbf{v}^*)} \tag{162}$$

i.e.

$$c_t^* \ge \frac{\frac{1}{\eta_t}\log\left(\frac{s-N_{(1+c_t^*)\eta_t}(\mathbf{v}^*)}{N_{(1+c_t^*)\eta_t}(\mathbf{v}^*)}\right) + 1 + \log 2}{\log\frac{1}{\bar{\mu}_t^-}} \tag{163}$$

Select the smallest $c_t^*$ that satisfies Eq. (163). Since $(1+c_t^*)\eta_t \in \{0, 1, 2, ..., n\}$. Let $\eta_t^* = \lceil(1+c^*)\eta_t\rceil$.

(2) If $\frac{N_{2\eta_t}(\mathbf{v}^*)}{s-N_{2\eta_t}(\mathbf{v}^*)} \le (2e\mu_t^-)^{\eta_t}$, we want to find the smallest $c$ such that $\left(\frac{(1+c_t^*)e}{c_t^*}\right)^{c_t^*\eta_t}(\mu_t^-)^{c_t^*\eta_t} \le \frac{N_{(1+c_t^*)\eta_t}(\mathbf{v}^*)}{s-N_{(1+c_t^*)\eta_t}(\mathbf{v}^*)}$. Similar, since $\left(\frac{1+c_t^*}{c_t^*}\right)^{c_t^*} \le e$, we also consider a sufficient condition: found smallest $c_t^*$ such that

$$e^{\eta_t}(e\mu_t^-)^{c_t^*\eta_t} \le \frac{N_{(1+c_t^*)\eta_t}(\mathbf{v}^*)}{s-N_{(1+c_t^*)\eta_t}(\mathbf{v}^*)} \tag{164}$$

i.e.

$$c_t^* \ge \frac{\frac{1}{\eta_t}\log\left(\frac{s-N_{(1+c_t^*)\eta_t}(\mathbf{v}^*)}{N_{(1+c_t^*)\eta_t}(\mathbf{v}^*)}\right)}{\log\frac{1}{\bar{\mu}_t^-} - 1} \tag{165}$$

Select the smallest $c_t^*$ that satisfies Eq. (165).

The above procedure for finding $(1 + c_t^*)\eta_t$ can be described in Algorithm **??**. After finding the most suitable $(1 + c_t^*)\eta_t$, $q(\mathbf{v}_t \in \mathcal{S}_a)$ can be upper bounded by

$$q(\mathbf{v}_t \in \mathcal{S}_a) \leq \frac{2N_{(1+c_t^*)\eta_t}(\mathbf{v}^*)}{s} \tag{166}$$

Now, we examine the selection of $\eta_t'$. Our aim is to let $q(\mathbf{v}_t \in \mathcal{S}_b) \cdot \frac{n \cdot \mathcal{A}_t \cdot \mathcal{B}_t}{\bar{R}_{t-1}^2 \cdot \bar{R}_t^{2\eta_t'}}$ be dominated by $\frac{N_{(1+c_t^*)\eta_t}(\mathbf{v}^*)}{s} \cdot \frac{n}{s^{\psi_t}}$. Consider the following sufficient condition:

$$\frac{N_{(1+c_t^*)\eta_t}(\mathbf{v}^*)}{s} \cdot \frac{n}{s^{\psi_t}} \geq \frac{s - N_{\eta_t}(\mathbf{v}^*)}{s} \cdot \frac{n \cdot \mathcal{A}_t \cdot \mathcal{B}_t}{\bar{R}_{t-1}^2 \cdot \bar{R}_t^{2\eta_t'}} \tag{167}$$

i.e.

$$\eta_t' \geq \max \left\{ \frac{\log \frac{s - N_{\eta_t}(\mathbf{v}^*)}{N_{(1+c_t^*)\eta_t}(\mathbf{v}^*)} + \log(\mathcal{A}_t \cdot \mathcal{C}_t \cdot s^{\psi_t})}{2 \log \bar{R}_t}, 0 \right\} \tag{168}$$

where $\mathcal{C}_t = \mathcal{B}_t / \bar{R}_{t-1}^2 = 1 - 1/\bar{R}_{t-1}^2$.

Note that the r.h.s is monotonically decreasing w.r.t $\eta_t$. However, since $\eta_t' \in \{0, 1, 2, ..., n\}$, we have to make sure the r.h.s is dominated by $n$. In fact, this constraint is naturally satisfied as $\log(\frac{s - N_{\eta_t}(\mathbf{v}^*)}{N_{(1+c_t^*)\eta_t}(\mathbf{v}^*)+1})$ goes to $-\infty$ when $\eta_t \to n$.

Finally, we control the third term, $\frac{s - N_{\eta_t}(\mathbf{v}^*)}{s} \min\{(n(1 - \bar{\mu}_t^+))^{\eta_t - \eta_t' + 2}, 1\} \cdot \frac{n}{s^{\psi_t}} 1$ to be dominated be the first term $\frac{N_{(1+c_t^*)\eta_t}(\mathbf{v}^*)}{s} \cdot \frac{n}{s^{\psi_t}}$. That is

$$\frac{N_{(1+c_t^*)\eta_t}(\mathbf{v}^*)}{s} \geq \frac{s - N_{\eta_t}(\mathbf{v}^*)}{s} \min\{(n(1 - \bar{\mu}_t^+))^{\eta_t - \eta_t' + 2}, 1\} \tag{169}$$

i.e.

$$\frac{N_{(1+c_t^*)\eta_t}(\mathbf{v}^*)}{s} \geq \frac{s - N_{\eta_t}(\mathbf{v}^*)}{s} (\min\{n(1 - \bar{\mu}_t^+), 1\})^{\eta_t - \eta_t' + 2} \tag{170}$$

$$\eta_t - \eta_t' + 2 \geq \frac{\log \frac{s - N_{\eta_t}(\mathbf{v}^*)}{N_{(1+c_t^*)\eta_t}(\mathbf{v}^*)}}{\log \frac{1}{\min\{n(1 - \bar{\mu}_t^+), 1\}}} \tag{171}$$

Similarly, $\log(\frac{s - N_{\eta_t}(\mathbf{v}^*)}{N_{\eta_t^*}(\mathbf{v}^*)+1})$ can goes to $-\infty$ when $\eta_t \to n$ and the above inequality naturally holds. Thus, we combine the above results, we have when $\eta_t \in [n]$ satisfies:

$$\eta_t \geq \frac{\log \frac{s - N_{\eta_t}(\mathbf{v}^*)}{N_{(1+c_t^*)\eta_t}(\mathbf{v}^*)}}{\log \frac{1}{n(1 - \bar{\mu}_t^+)}} + \max \left\{ \frac{\log \frac{s - N_{\eta_t}(\mathbf{v}^*)}{N_{(1+c_t^*)\eta_t}(\mathbf{v}^*)} + \log(\mathcal{A}_t \cdot \mathcal{C}_t \cdot s^{\psi_t})}{2 \log \bar{R}_t}, 0 \right\} - 2 \tag{172}$$

Summarizing the above results, we obtain that

$$\mathbb{E}_{\mathbf{v}_t \sim q(\mathbf{v}_{t|0})}[\Delta_t(\mathbf{v}_t)] \leq q(\mathbf{v}_t \in \mathcal{S}_a) \cdot \frac{n}{s^{\psi_t}} + q(\mathbf{v}_t \in \mathcal{S}_b) \cdot \frac{n \cdot \mathcal{A}_t \cdot \mathcal{B}_t}{\bar{R}_{t-1}^2 \cdot \bar{R}_t^{2\eta_t'}} + q(\mathbf{v}_t \in \mathcal{S}_c) \cdot \frac{n}{s^{\psi_t}} \tag{173}$$

---

[1]Instead of using Eq. 156, a better bound to estimate the third term is $\frac{s - N_{\eta_t}(\mathbf{v}^*)}{s} \cdot \mathbb{P}(\bar{\omega}(\mathbf{v}_t, \mathbf{v}_0) \geq \eta_t - \eta_t' + 2)$ and trade-off with first term, we get the smallest $\eta_t - \eta_t' + 2$ such that $\mathbb{P}(\bar{\omega}(\mathbf{v}_t, \mathbf{v}_0) \geq \eta_t - \eta_t' + 2) \leq \frac{N_{(1+c_t^*)\eta_t}(\mathbf{v}^*)}{s - N_{\eta_t}(\mathbf{v}^*)}$ where $\mathbb{P}(\bar{\omega}(\mathbf{v}_t, \mathbf{v}_0) \geq \eta_t - \eta_t' + 2) = \sum_{j = \eta_t - \eta_t' + 2}^{n} \binom{n}{j}(1 - \bar{\mu}_t^+)^j (\bar{\mu}_t^+)^{n-j}$.

$$\leq \min\left\{\frac{4N_{(1+c_t^*)\eta_t}(\mathbf{v}^*)}{s}, 1\right\} \cdot \frac{n}{s^{\psi_t}} \tag{174}$$

Summarizing the above results, we calculate the privacy bound of discrete diffusion model by following Algorithm **??**.

### J.6 Proof of Lemma A.6

*Proof.* To begin with,

$$\mathbb{E}_{\mathbf{v}_t \sim (q(\mathbf{v}_{t|1}) - q(\mathbf{v}_{t|0}))}[\mathcal{D}_{\mathrm{KL}}(q(\mathbf{v}_{t-1|1}^i | \mathbf{v}_{t|1} = \mathbf{v}_t) \| q(\mathbf{v}_{t-1|0}^i | \mathbf{v}_{t|0} = \mathbf{v}_t))] \tag{175}$$

$$=\mathbb{E}_{\mathbf{v}_t \sim q(\mathbf{v}_{t|0})}\left[\left(1 - \frac{q(\mathbf{v}_{t|0} = \mathbf{v}_t)}{q(\mathbf{v}_{t|1} = \mathbf{v}_t)}\right)\frac{q(\mathbf{v}_{t|1} = \mathbf{v}_t)}{q(\mathbf{v}_{t|0} = \mathbf{v}_t)}\mathcal{D}_{\mathrm{KL}}(q(\mathbf{v}_{t-1|1}^i | \mathbf{v}_{t|1} = \mathbf{v}_t) \| q(\mathbf{v}_{t-1|0}^i | \mathbf{v}_{t|0} = \mathbf{v}_t))\right] \tag{176}$$

Now consider $1 - \frac{q(\mathbf{v}_{t|0} = \mathbf{v}_t)}{q(\mathbf{v}_{t|1} = \mathbf{v}_t)}$:

$$1 - \frac{q(\mathbf{v}_{t|0} = \mathbf{v}_t)}{q(\mathbf{v}_{t|1} = \mathbf{v}_t)} = \frac{1}{s+1} \cdot \left(1 - \frac{s}{\zeta(\mathcal{V}_1, \mathbf{v}^*, \mathbf{v}_t, t)}\right) \leq \frac{1}{s+1} \cdot \mathbb{1}_{\mathbf{v}_t \in \Omega} \tag{177}$$

where $\Omega = \{\mathbf{v}_t | \zeta(\mathcal{V}_1, \mathbf{v}^*, \mathbf{v}_t, t) > s\}$. Therefore,

$$\mathbb{E}_{\mathbf{v}_t \sim (q(\mathbf{v}_{t|1}) - q(\mathbf{v}_{t|0}))}[\mathcal{D}_{\mathrm{KL}}(q(\mathbf{v}_{t-1|1}^i | \mathbf{v}_{t|1} = \mathbf{v}_t) \| q(\mathbf{v}_{t-1|0}^i | \mathbf{v}_{t|0} = \mathbf{v}_t))] \tag{178}$$

$$\leq \mathbb{E}_{\mathbf{v}_t \sim q(\mathbf{v}_{t|0}), \mathbf{v}_t \in \Omega}\left[\frac{1}{1+s}\sum_{\mathbf{v}_{t-1}^i = 1}^{k}\frac{q(\mathbf{v}_{t-1|1}^i = \mathbf{v}_{t-1}^i, \mathbf{v}_{t|1} = \mathbf{v}_t)}{q(\mathbf{v}_{t|0} = \mathbf{v}_t)}\log\frac{q(\mathbf{v}_{t-1|1}^i = \mathbf{v}_{t-1}^i | \mathbf{v}_{t|1} = \mathbf{v}_t)}{q(\mathbf{v}_{t-1|0}^i = \mathbf{v}_{t-1}^i | \mathbf{v}_{t|0} = \mathbf{v}_t)}\right] \tag{179}$$

$$= \mathbb{E}_{\mathbf{v}_t \sim q(\mathbf{v}_{t|0}), \mathbf{v}_t \in \Omega}\left[\frac{1}{s} \cdot \frac{\sum_{\mathbf{v}_0 \in \mathcal{V}_1} \tau(\mathbf{v}_0^i, \mathbf{v}_{t-1}^i, \mathbf{v}_t^i) \cdot (\bar{R}_t)^{\bar{\omega}(\mathbf{v}^*, \mathbf{v}_t) - \bar{\omega}(\mathbf{v}_0, \mathbf{v}_t)}}{1 + \zeta(\mathcal{V}_1, \mathbf{v}^*, \mathbf{v}_t, t)}\log\frac{q(\mathbf{v}_{t-1|1}^i = \mathbf{v}_{t-1}^i | \mathbf{v}_{t|1} = \mathbf{v}_t)}{q(\mathbf{v}_{t-1|0}^i = \mathbf{v}_{t-1}^i | \mathbf{v}_{t|0} = \mathbf{v}_t)}\right] \tag{180}$$

Now consider $\log\frac{q(\mathbf{v}_{t-1|1}^i = \mathbf{v}_{t-1}^i | \mathbf{v}_{t|1} = \mathbf{v}_t)}{q(\mathbf{v}_{t-1|0}^i = \mathbf{v}_{t-1}^i | \mathbf{v}_{t|0} = \mathbf{v}_t)}$:

$$\log\frac{q(\mathbf{v}_{t-1|1}^i = \mathbf{v}_{t-1}^i | \mathbf{v}_{t|1} = \mathbf{v}_t)}{q(\mathbf{v}_{t-1|0}^i = \mathbf{v}_{t-1}^i | \mathbf{v}_{t|0} = \mathbf{v}_t)} = \log\frac{q(\mathbf{v}_{t-1|1}^i = \mathbf{v}_{t-1}^i, \mathbf{v}_{t|1} = \mathbf{v}_t)q(\mathbf{v}_{t|0} = \mathbf{v}_t)}{q(\mathbf{v}_{t-1|0}^i = \mathbf{v}_{t-1}^i, \mathbf{v}_{t|0} = \mathbf{v}_t)q(\mathbf{v}_{t|1} = \mathbf{v}_t)} \tag{181}$$

$$= \log(1 - \frac{1}{1 + \sum_{\mathbf{v}_0 \in \mathcal{V}_1}\frac{\tau(\mathbf{v}_0^i, \mathbf{v}_{t-1}^i, \mathbf{v}_t^i)}{\tau(\mathbf{v}^{*i}, \mathbf{v}_{t-1}^i, \mathbf{v}_t^i)}(\bar{R}_t)^{\bar{\omega}(\mathbf{v}^*, \mathbf{v}_t) - \bar{\omega}(\mathbf{v}_0, \mathbf{v}_t)}})(1 + \frac{1}{\zeta(\mathcal{V}_1, \mathbf{v}^*, \mathbf{v}_t, t)}) \tag{182}$$

$$= \log\left[(1 + \frac{\sum_{\mathbf{v}_0 \in \mathcal{V}_1}(\frac{\tau(\mathbf{v}_0^i, \mathbf{v}_{t-1}^i, \mathbf{v}_t^i)}{\tau(\mathbf{v}^{*i}, \mathbf{v}_{t-1}^i, \mathbf{v}_t^i)} - 1)(\bar{R}_t)^{\bar{\omega}(\mathbf{v}^*, \mathbf{v}_t) - \bar{\omega}(\mathbf{v}_0, \mathbf{v}_t)}}{(1 + \sum_{\mathbf{v}_0 \in \mathcal{V}_1}\frac{\tau(\mathbf{v}_0^i, \mathbf{v}_{t-1}^i, \mathbf{v}_t^i)}{\tau(\mathbf{v}^{*i}, \mathbf{v}_{t-1}^i, \mathbf{v}_t^i)}(\bar{R}_t)^{\bar{\omega}(\mathbf{v}^*, \mathbf{v}_t) - \bar{\omega}(\mathbf{v}_0, \mathbf{v}_t)}) \cdot \zeta(\mathcal{V}_1, \mathbf{v}^*, \mathbf{v}_t, t)})\right] \tag{183}$$

$$\leq \frac{\max_{\mathbf{v}_0^i}\frac{\tau(\mathbf{v}_0^i, \mathbf{v}_{t-1}^i, \mathbf{v}_t^i)}{\tau(\mathbf{v}^{*i}, \mathbf{v}_{t-1}^i, \mathbf{v}_t^i)} - 1}{\max_{\mathbf{v}_0^i}\frac{\tau(\mathbf{v}_0^i, \mathbf{v}_{t-1}^i, \mathbf{v}_t^i)}{\tau(\mathbf{v}^{*i}, \mathbf{v}_{t-1}^i, \mathbf{v}_t^i)} \cdot \zeta(\mathcal{V}_1, \mathbf{v}^*, \mathbf{v}_t, t) + 1} \tag{184}$$

1. When $\mathbf{v}_t^i = \mathbf{v}^{*i}$, we further compare $\mathbf{v}_{t-1}^i, \mathbf{v}_t^i$: (1) If $\mathbf{v}_{t-1}^i = \mathbf{v}_t^i$, we have $\frac{\tau(\mathbf{v}_0^i, \mathbf{v}_{t-1}^i, \mathbf{v}_t^i)}{\tau(\mathbf{v}^{*i}, \mathbf{v}_{t-1}^i, \mathbf{v}_t^i)} \leq 1$. (2) If $\mathbf{v}_{t-1}^i \neq \mathbf{v}_t^i$, we get $\frac{\tau(\mathbf{v}_0^i, \mathbf{v}_{t-1}^i, \mathbf{v}_t^i)}{\tau(\mathbf{v}^{*i}, \mathbf{v}_{t-1}^i, \mathbf{v}_t^i)} \leq \bar{R}_t\bar{R}_{t-1}$. Hence,

$$\mathbb{E}_{\mathbf{v}_t \sim (q(\mathbf{v}_{t|1}) - q(\mathbf{v}_{t|0}))}[\mathcal{D}_{\mathrm{KL}}(q(\mathbf{v}_{t-1|1}^i | \mathbf{v}_{t|1} = \mathbf{v}_t) \| q(\mathbf{v}_{t-1|0}^i | \mathbf{v}_{t|0} = \mathbf{v}_t))] \tag{185}$$

$$\leq \mathbb{E}_{\mathbf{v}_t \sim q(\mathbf{v}_{t|0}), \mathbf{v}_t \in \Omega}\left[\frac{1}{s}\frac{\sum_{\mathbf{v}_0 \in \mathcal{V}_1} \tau(\mathbf{v}_0^i, \mathbf{v}_{t-1}^i, \mathbf{v}_t^i) \cdot (\bar{R}_t)^{\bar{\omega}(\mathbf{v}^*, \mathbf{v}_t) - \bar{\omega}(\mathbf{v}_0, \mathbf{v}_t)}}{1 + \zeta(\mathcal{V}_1, \mathbf{v}^*, \mathbf{v}_t, t)}\log\frac{q(\mathbf{v}_{t-1|1}^i = \mathbf{v}_{t-1}^i | \mathbf{v}_{t|1} = \mathbf{v}_t)}{q(\mathbf{v}_{t-1|0}^i = \mathbf{v}_{t-1}^i | \mathbf{v}_{t|0} = \mathbf{v}_t)}\right] \tag{186}$$

$$\leq \mathbb{E}_{\mathbf{v}_t \sim q(\mathbf{v}_{t|0}), \mathbf{v}_t \in \Omega} \left[ \frac{1}{s} \frac{\sum_{\mathbf{v}_0 \in \mathcal{V}_1} \sum_{\mathbf{v}_{t-1}^i \neq \mathbf{v}_t^i} \tau(\mathbf{v}_0^i, \mathbf{v}_{t-1}^i, \mathbf{v}_t^i) \cdot (\bar{R}_t)^{\bar{\omega}(\mathbf{v}^*, \mathbf{v}_t) - \bar{\omega}(\mathbf{v}_0, \mathbf{v}_t)}}{1 + \zeta(\mathcal{V}_1, \mathbf{v}^*, \mathbf{v}_t, t)} \cdot \frac{\bar{R}_t \bar{R}_{t-1} - 1}{\bar{R}_t \bar{R}_{t-1} \zeta(\mathcal{V}_1, \mathbf{v}^*, \mathbf{v}_t, t)} \right] \quad (187)$$

$$= \mathbb{E}_{\mathbf{v}_t \sim q(\mathbf{v}_{t|0}), \mathbf{v}_t \in \Omega} \left[ \frac{1}{s} \frac{\sum_{\mathbf{v}_0 \in \mathcal{V}_1} (1 - \tau(\mathbf{v}_0^i, \mathbf{v}_t^i, \mathbf{v}_t^i)) \cdot (\bar{R}_t)^{\bar{\omega}(\mathbf{v}^*, \mathbf{v}_t) - \bar{\omega}(\mathbf{v}_0, \mathbf{v}_t)}}{1 + \zeta(\mathcal{V}_1, \mathbf{v}^*, \mathbf{v}_t, t)} \cdot \frac{\bar{R}_t \bar{R}_{t-1} - 1}{\bar{R}_t \bar{R}_{t-1} \zeta(\mathcal{V}_1, \mathbf{v}^*, \mathbf{v}_t, t)} \right] \quad (188)$$

$$\leq \mathbb{E}_{\mathbf{v}_t \sim q(\mathbf{v}_{t|0}), \mathbf{v}_t \in \Omega} \left[ \frac{(1 - \frac{\mu_t^+ \bar{\mu}_{t-1}^+}{\bar{\mu}_t^+}) \cdot (\bar{R}_t \bar{R}_{t-1} - 1)}{\bar{R}_t \bar{R}_{t-1} \cdot s(1 + s)} \right] \quad (189)$$

$$= \frac{1}{s(s+1)} \cdot (1 - \frac{\mu_t^+ \bar{\mu}_{t-1}^+}{\bar{\mu}_t^+}) \cdot (1 - \frac{1}{\bar{R}_t \bar{R}_{t-1}}) \cdot \mathbb{P}(\mathbf{v}_{t|0} \in \Omega) \quad (190)$$

2. When $\mathbf{v}_t^i \neq \mathbf{v}^{*i}$, similarly, (1) If $\mathbf{v}_{t-1}^i = \mathbf{v}_t^i$, we have $\frac{\tau(\mathbf{v}_0^i, \mathbf{v}_{t-1}^i, \mathbf{v}_t^i)}{\tau(\mathbf{v}^{*i}, \mathbf{v}_{t-1}^i, \mathbf{v}_t^i)} \leq \bar{R}_{t-1}/\bar{R}_t$. (2) If $\mathbf{v}_{t-1}^i \neq \mathbf{v}_t^i$, we have $\frac{\tau(\mathbf{v}_0^i, \mathbf{v}_{t-1}^i, \mathbf{v}_t^i)}{\tau(\mathbf{v}^{*i}, \mathbf{v}_{t-1}^i, \mathbf{v}_t^i)} \leq \bar{R}_{t-1}$. Therefore

$$\mathbb{E}_{\mathbf{v}_t \sim (q(\mathbf{v}_{t|1}) - q(\mathbf{v}_{t|0}))} [\mathcal{D}_{\mathrm{KL}}(q(\mathbf{v}_{t-1|1}^i | \mathbf{v}_{t|1} = \mathbf{v}_t) \| q(\mathbf{v}_{t-1|0}^i | \mathbf{v}_{t|0} = \mathbf{v}_t))] \quad (191)$$

$$\leq \mathbb{E}_{\mathbf{v}_t \sim q(\mathbf{v}_{t|0}), \mathbf{v}_t \in \Omega} \left[ \frac{1}{s} \frac{\sum_{\mathbf{v}_0 \in \mathcal{V}_1} \tau(\mathbf{v}_0^i, \mathbf{v}_{t-1}^i, \mathbf{v}_t^i) \cdot (\bar{R}_t)^{\bar{\omega}(\mathbf{v}^*, \mathbf{v}_t) - \bar{\omega}(\mathbf{v}_0, \mathbf{v}_t)}}{1 + \zeta(\mathcal{V}_1, \mathbf{v}^*, \mathbf{v}_t, t)} \log \frac{q(\mathbf{v}_{t-1|1}^i = \mathbf{v}_{t-1}^i | \mathbf{v}_{t|1} = \mathbf{v}_t)}{q(\mathbf{v}_{t-1|0}^i = \mathbf{v}_{t-1}^i | \mathbf{v}_{t|0} = \mathbf{v}_t)} \right] \quad (192)$$

$$\leq \mathbb{E}_{\mathbf{v}_t \sim q(\mathbf{v}_{t|0}), \mathbf{v}_t \in \Omega} \left[ \frac{1}{s} \frac{\sum_{\mathbf{v}_0 \in \mathcal{V}_1} \sum_{\mathbf{v}_{t-1}^i = \mathbf{v}_t^i} \tau(\mathbf{v}_0^i, \mathbf{v}_{t-1}^i, \mathbf{v}_t^i) \cdot (\bar{R}_t)^{\bar{\omega}(\mathbf{v}^*, \mathbf{v}_t) - \bar{\omega}(\mathbf{v}_0, \mathbf{v}_t)}}{1 + \zeta(\mathcal{V}_1, \mathbf{v}^*, \mathbf{v}_t, t)} \frac{\bar{R}_{t-1}/\bar{R}_t - 1}{\bar{R}_{t-1}/\bar{R}_t \zeta(\mathcal{V}_1, \mathbf{v}^*, \mathbf{v}_t, t)} \right. \quad (193)$$

$$\left. + \frac{1}{s} \cdot \frac{\sum_{\mathbf{v}_0 \in \mathcal{V}_1} \sum_{\mathbf{v}_{t-1}^i \neq \mathbf{v}_t^i} \tau(\mathbf{v}_0^i, \mathbf{v}_{t-1}^i, \mathbf{v}_t^i) \cdot (\bar{R}_t)^{\bar{\omega}(\mathbf{v}^*, \mathbf{v}_t) - \bar{\omega}(\mathbf{v}_0, \mathbf{v}_t)}}{1 + \zeta(\mathcal{V}_1, \mathbf{v}^*, \mathbf{v}_t, t)} \frac{\bar{R}_{t-1} - 1}{\bar{R}_{t-1} \zeta(\mathcal{V}_1, \mathbf{v}^*, \mathbf{v}_t, t)} \right] \quad (194)$$

$$= \mathbb{E}_{\mathbf{v}_t \sim q(\mathbf{v}_{t|0}), \mathbf{v}_t \in \Omega} \left[ \frac{1}{s} \frac{\sum_{\mathbf{v}_0 \in \mathcal{V}_1} (\bar{R}_{t-1} - 1 - (\bar{R}_t - 1) \tau(\mathbf{v}_0^i, \mathbf{v}_t^i, \mathbf{v}_t^i)) \cdot (\bar{R}_t)^{\bar{\omega}(\mathbf{v}^*, \mathbf{v}_t) - \bar{\omega}(\mathbf{v}_0, \mathbf{v}_t)}}{(1 + \zeta(\mathcal{V}_1, \mathbf{v}^*, \mathbf{v}_t, t)) \bar{R}_{t-1} \zeta(\mathcal{V}_1, \mathbf{v}^*, \mathbf{v}_t, t)} \right] \quad (195)$$

$$\leq \mathbb{E}_{\mathbf{v}_t \sim q(\mathbf{v}_{t|0}), \mathbf{v}_t \in \Omega} \left[ \frac{((\bar{R}_{t-1} - 1) - \frac{\mu_t^+ \bar{\mu}_{t-1}^+}{\bar{\mu}_t^+} (\bar{R}_t - 1))}{\bar{R}_{t-1} s(1 + s)} \right] \quad (196)$$

$$= \frac{1}{s(s+1)} \cdot \left[ (1 - \frac{\mu_t^+ \bar{\mu}_{t-1}^+}{\bar{\mu}_t^+}) \cdot (1 - \frac{1}{\bar{R}_{t-1}}) + \frac{\mu_t^+ \bar{\mu}_{t-1}^+}{\bar{\mu}_t^+} \cdot (1 - \frac{\bar{R}_t}{\bar{R}_{t-1}}) \right] \cdot \mathbb{P}(\mathbf{v}_{t|0} \in \Omega) \quad (197)$$

Compare two bounds Eq. (190) and Eq. (197), since we have

$$(1 - \frac{\mu_t^+ \bar{\mu}_{t-1}^+}{\bar{\mu}_t^+})(1 - \frac{1}{\bar{R}_{t-1}}) + \frac{\mu_t^+ \bar{\mu}_{t-1}^+}{\bar{\mu}_t^+}(1 - \frac{\bar{R}_t}{\bar{R}_{t-1}}) > (1 - \frac{\mu_t^+ \bar{\mu}_{t-1}^+}{\bar{\mu}_t^+})(1 - \frac{1}{\bar{R}_t \bar{R}_{t-1}}) \quad (198)$$

We derive the following bound:

$$\mathbb{E}_{\mathbf{v}_t \sim (q(\mathbf{v}_{t|1}) - q(\mathbf{v}_{t|0}))} [\mathcal{D}_{\mathrm{KL}}(q(\mathbf{v}_{t-1|1}^i | \mathbf{v}_{t|1} = \mathbf{v}_t) \| q(\mathbf{v}_{t-1|0}^i | \mathbf{v}_{t|0} = \mathbf{v}_t))] \quad (199)$$

$$\leq \frac{1}{s(s+1)} \cdot \left[ (1 - \frac{\mu_t^+ \bar{\mu}_{t-1}^+}{\bar{\mu}_t^+}) \cdot (1 - \frac{1}{\bar{R}_{t-1}}) + \frac{\mu_t^+ \bar{\mu}_{t-1}^+}{\bar{\mu}_t^+} \cdot (1 - \frac{\bar{R}_t}{\bar{R}_{t-1}}) \right] \cdot \mathbb{P}(\mathbf{v}_{t|0} \in \Omega) \quad (200)$$

$\square$

### J.7 Proof of Lemma D.1

*Proof.* Given adjacent datasets $\mathcal{V}_0, \mathcal{V}_1$ and diffusion coefficients $\{\alpha_t\}_{t \in [T]}$, we consider the probability transitions in the generation process.

$$p_\phi(\mathbf{v}_{t-1|j} | \mathbf{v}_{t|j}) = p_\phi(\mathbf{v}_{t-1|j}^1 | \mathbf{v}_{t|j}) \cdot p_\phi(\mathbf{v}_{t-1|j}^2 | \mathbf{v}_{t|j}) \quad (201)$$

$$= \prod_{i=1}^{2} [\sum_{\mathbf{v}_{0|j}^i=0}^{1} q(\mathbf{v}_{t-1|j}^i, \mathbf{v}_{0|j}^i | \mathbf{v}_{t|j}^i) p_\phi(\mathbf{v}_{0|j}^i | \mathbf{v}_{t|j})] \tag{202}$$

where $j \in \{0, 1\}$ denote the dataset index.

For $q(\mathbf{v}_{t-1}^i, \mathbf{v}_0^i | \mathbf{v}_t^i)$, from Lemma K.2, we obtain

$$q(\mathbf{v}_{t-1}^i | \mathbf{v}_t^i, \mathbf{v}_0^i) = \begin{cases} \text{When } \mathbf{v}_0^i = \mathbf{v}_t^i \begin{cases} \text{If } \mathbf{v}_0^i = \mathbf{v}_t^i = \mathbf{v}_{t-1}^i, \dfrac{\mu_t^+ \cdot \bar{\mu}_{t-1}^+}{\bar{\mu}_t^+} =: C_{1,t} \\[2mm] \text{If } \mathbf{v}_0^i = \mathbf{v}_t^i \neq \mathbf{v}_{t-1}^i, \dfrac{\mu_t^- \cdot \bar{\mu}_{t-1}^-}{\bar{\mu}_t^+} =: C_{2,t} \end{cases} \\[6mm] \text{When } \mathbf{v}_0^i \neq \mathbf{v}_t^i \begin{cases} \text{If } \mathbf{v}_0^i \neq \mathbf{v}_t^i = \mathbf{v}_{t-1}^i, \dfrac{\mu_t^+ \cdot \bar{\mu}_{t-1}^-}{\bar{\mu}_t^-} =: \tilde{C}_{1,t} \\[2mm] \text{If } \mathbf{v}_t^i \neq \mathbf{v}_0^i = \mathbf{v}_{t-1}^i, \dfrac{\mu_t^- \cdot \bar{\mu}_{t-1}^+}{\bar{\mu}_t^-} =: \tilde{C}_{2,t} \end{cases} \end{cases} \tag{203}$$

where we have $C_{1,t} + C_{2,t} = \tilde{C}_{1,t} + \tilde{C}_{2,t}$ and $C_{1,t} > \tilde{C}_{1,t}, C_{2,t} < \tilde{C}_{2,t}$.

Now, consider $q(\mathbf{v}_{t|1})$ and $q(\mathbf{v}_{t|0})$. From the forward diffusion process, we have

$$q(\mathbf{v}_{t|1} = \begin{bmatrix} 1 \\ 1 \end{bmatrix}) = \frac{(\bar{\mu}_t^+)^2 + (s-1) \cdot (\bar{\mu}_t^-)^2}{s}, q(\mathbf{v}_{t|1} = \begin{bmatrix} 0 \\ 0 \end{bmatrix}) = \frac{(\bar{\mu}_t^-)^2 + (s-1) \cdot (\bar{\mu}_t^+)^2}{s}, \tag{204}$$

$$q(\mathbf{v}_{t|1} = \begin{bmatrix} 1 \\ 0 \end{bmatrix}) = q(\mathbf{v}_{t|1} = \begin{bmatrix} 0 \\ 1 \end{bmatrix}) = \bar{\mu}_t^+ \cdot \bar{\mu}_t^-. \tag{205}$$

and

$$q(\mathbf{v}_{t|0} = \begin{bmatrix} 1 \\ 1 \end{bmatrix}) = \frac{2 \cdot (\bar{\mu}_t^+)^2 + (s-1) \cdot (\bar{\mu}_t^-)^2}{s+1}, q(\mathbf{v}_{t|0} = \begin{bmatrix} 0 \\ 0 \end{bmatrix}) = \frac{2 \cdot (\bar{\mu}_t^-)^2 + (s-1) \cdot (\bar{\mu}_t^+)^2}{s+1}, \tag{206}$$

$$q(\mathbf{v}_{t|0} = \begin{bmatrix} 1 \\ 0 \end{bmatrix}) = q(\mathbf{v}_{t|1} = \begin{bmatrix} 0 \\ 1 \end{bmatrix}) = \bar{\mu}_t^+ \cdot \bar{\mu}_t^-. \tag{207}$$

Further, we consider the prediction probability $p_\phi(\mathbf{v}_{0|j}^i | \mathbf{v}_{t|j})$, which is determined by the training of denoising networks. In the training procedure, we are optimizing the following objective function

$$\text{minimize } \mathcal{D}_{\text{KL}}(q(\mathbf{v}_{0|j}) \| \prod_{i=1}^{2} \sum_{\mathbf{v}_{t|1}} p_\phi(\mathbf{v}_{0|1}^i | \mathbf{v}_{t|1}) q(\mathbf{v}_{t|1})) \tag{208}$$

From Lemma K.8, we obtain the optimal solution is when $p_\phi(\mathbf{v}_{0|1}^i | \mathbf{v}_{t|1}) = q(\mathbf{v}_{0|1}^i | \mathbf{v}_{t|1})$.

Therefore, $p_\phi(\mathbf{v}_{t-1|j} | \mathbf{v}_{t|j}) = \prod_{i=1}^{2} [\sum_{\mathbf{v}_{0|j}^i=0}^{1} q(\mathbf{v}_{t-1|j}^i, \mathbf{v}_{0|j}^i | \mathbf{v}_{t|j}^i) q(\mathbf{v}_{0|j}^i | \mathbf{v}_{t|j})]$. For $q(\mathbf{v}_{0|j}^i | \mathbf{v}_{t|j})]$, we have the following calculation: for dataset $\mathcal{V}_1$,

$$q(\mathbf{v}_{0|1}^i = 0 | \mathbf{v}_{t|1} = \begin{bmatrix} 0 \\ 0 \end{bmatrix}) = \frac{(s-1) \cdot (\bar{\mu}_t^+)^2}{(s-1) \cdot (\bar{\mu}_t^+)^2 + (\bar{\mu}_t^-)^2}, q(\mathbf{v}_{0|1}^i = 1 | \mathbf{v}_{t|1} = \begin{bmatrix} 0 \\ 0 \end{bmatrix}) = \frac{(\bar{\mu}_t^-)^2}{(s-1) \cdot (\bar{\mu}_t^+)^2 + (\bar{\mu}_t^-)^2},$$

$$q(\mathbf{v}_{0|1}^i = 0 | \mathbf{v}_{t|1} = \begin{bmatrix} 1 \\ 1 \end{bmatrix}) = \frac{(s-1) \cdot (\bar{\mu}_t^-)^2}{(s-1) \cdot (\bar{\mu}_t^-)^2 + (\bar{\mu}_t^+)^2}, q(\mathbf{v}_{0|1}^i = 1 | \mathbf{v}_{t|1} = \begin{bmatrix} 1 \\ 1 \end{bmatrix}) = \frac{(\bar{\mu}_t^+)^2}{(s-1) \cdot (\bar{\mu}_t^-)^2 + (\bar{\mu}_t^+)^2},$$

$$q(\mathbf{v}_{0|1}^i = 0 | \mathbf{v}_{t|1} = \begin{bmatrix} 1 \\ 0 \end{bmatrix}) = \frac{s-1}{s}, q(\mathbf{v}_{0|1}^i = 1 | \mathbf{v}_{t|1} = \begin{bmatrix} 0 \\ 1 \end{bmatrix}) = \frac{1}{s}.$$

and for dataset $\mathcal{V}_0$

$$q(\mathbf{v}_{0|0}^i = 0 | \mathbf{v}_{t|0} = \begin{bmatrix} 0 \\ 0 \end{bmatrix}) = \frac{(s-1) \cdot (\bar{\mu}_t^+)^2}{(s-1) \cdot (\bar{\mu}_t^+)^2 + 2 \cdot (\bar{\mu}_t^-)^2}, q(\mathbf{v}_{0|0}^i = 1 | \mathbf{v}_{t|0} = \begin{bmatrix} 0 \\ 0 \end{bmatrix}) = \frac{2 \cdot (\bar{\mu}_t^-)^2}{(s-1) \cdot (\bar{\mu}_t^+)^2 + 2 \cdot (\bar{\mu}_t^-)^2},$$

$$q(\mathbf{v}_{0|0}^i = 0|\mathbf{v}_{t|0} = \begin{bmatrix}1\\1\end{bmatrix}) = \frac{(s-1)\cdot(\bar{\mu}_t^-)^2}{(s-1)\cdot(\bar{\mu}_t^-)^2 + 2\cdot(\bar{\mu}_t^+)^2}, q(\mathbf{v}_{0|0}^i = 1|\mathbf{v}_{t|0} = \begin{bmatrix}1\\1\end{bmatrix}) = \frac{2\cdot(\bar{\mu}_t^+)^2}{(s-1)\cdot(\bar{\mu}_t^-)^2 + 2\cdot(\bar{\mu}_t^+)^2},$$

$$q(\mathbf{v}_{0|0}^i = 0|\mathbf{v}_{t|0} = \begin{bmatrix}1\\0\end{bmatrix}) = \frac{s-1}{s+1}, q(\mathbf{v}_{0|0}^i = 1|\mathbf{v}_{t|0} = \begin{bmatrix}0\\1\end{bmatrix}) = \frac{2}{s+1}.$$

For the convenience of further derivation, we introduce the notation $q_j(x_1|x_2, x_3) := q(\mathbf{v}_{0|j}^i = x_1|\mathbf{v}_{t|j} = \begin{bmatrix}x_1\\x_2\end{bmatrix})]$ for simplicity. Therefore,

$$p_\phi(\mathbf{v}_{t-1|j} = \begin{bmatrix}0\\0\end{bmatrix}|\mathbf{v}_{t|j} = \begin{bmatrix}0\\0\end{bmatrix}) = (C_{1,t}\cdot q_j(0|0,0) + \tilde{C}_{1,t}\cdot q_j(1|0,0))^2$$

$$p_\phi(\mathbf{v}_{t-1|j} = \begin{bmatrix}1\\1\end{bmatrix}|\mathbf{v}_{t|j} = \begin{bmatrix}0\\0\end{bmatrix}) = (C_{2,t}\cdot q_j(0|0,0) + \tilde{C}_{2,t}\cdot q_j(1|0,0))^2$$

$$p_\phi(\mathbf{v}_{t-1|j} = \begin{bmatrix}1\\0\end{bmatrix}|\mathbf{v}_{t|j} = \begin{bmatrix}0\\0\end{bmatrix}) = (C_{2,t}\cdot q_j(0|0,0) + \tilde{C}_{2,t}\cdot q_j(1|0,0))(C_{1,t}\cdot q_j(0|0,0) + \tilde{C}_{1,t}\cdot q_j(1|0,0))$$

$$p_\phi(\mathbf{v}_{t-1|j} = \begin{bmatrix}0\\1\end{bmatrix}|\mathbf{v}_{t|j} = \begin{bmatrix}0\\0\end{bmatrix}) = (C_{2,t}\cdot q_j(0|0,0) + \tilde{C}_{2,t}\cdot q_j(1|0,0))(C_{1,t}\cdot q_j(0|0,0) + \tilde{C}_{1,t}\cdot q_j(1|0,0))$$

$$p_\phi(\mathbf{v}_{t-1|j} = \begin{bmatrix}0\\0\end{bmatrix}|\mathbf{v}_{t|j} = \begin{bmatrix}1\\1\end{bmatrix}) = (\tilde{C}_{2,t}\cdot q_j(0|1,1) + C_{2,t}\cdot q_j(1|1,1))^2$$

$$p_\phi(\mathbf{v}_{t-1|j} = \begin{bmatrix}1\\1\end{bmatrix}|\mathbf{v}_{t|j} = \begin{bmatrix}1\\1\end{bmatrix}) = (\tilde{C}_{1,t}\cdot q_j(0|1,1) + C_{1,t}\cdot q_j(1|1,1))^2$$

$$p_\phi(\mathbf{v}_{t-1|j} = \begin{bmatrix}0\\1\end{bmatrix}|\mathbf{v}_{t|j} = \begin{bmatrix}1\\1\end{bmatrix}) = (\tilde{C}_{1,t}\cdot q_j(0|1,1) + C_{1,t}\cdot q_j(1|1,1))(\tilde{C}_{2,t}\cdot q_j(0|1,1) + C_{2,t}\cdot q_j(1|1,1))$$

$$p_\phi(\mathbf{v}_{t-1|j} = \begin{bmatrix}1\\0\end{bmatrix}|\mathbf{v}_{t|j} = \begin{bmatrix}1\\1\end{bmatrix}) = (\tilde{C}_{1,t}\cdot q_j(0|1,1) + C_{1,t}\cdot q_j(1|1,1))(\tilde{C}_{2,t}\cdot q_j(0|1,1) + C_{2,t}\cdot q_j(1|1,1))$$

$$p_\phi(\mathbf{v}_{t-1|j} = \begin{bmatrix}0\\0\end{bmatrix}|\mathbf{v}_{t|j} = \begin{bmatrix}1\\0\end{bmatrix}) = (\tilde{C}_{1,t}\cdot q_j(1|1,0) + C_{1,t}\cdot q_j(0|1,0))(\tilde{C}_{2,t}\cdot q_j(0|1,0) + C_{2,t}\cdot q_j(1|1,0))$$

$$p_\phi(\mathbf{v}_{t-1|j} = \begin{bmatrix}1\\1\end{bmatrix}|\mathbf{v}_{t|j} = \begin{bmatrix}1\\0\end{bmatrix}) = (\tilde{C}_{1,t}\cdot q_j(0|1,0) + C_{1,t}\cdot q_j(1|1,0))(\tilde{C}_{2,t}\cdot q_j(1|1,0) + C_{2,t}\cdot q_j(0|1,0))$$

$$p_\phi(\mathbf{v}_{t-1|j} = \begin{bmatrix}1\\0\end{bmatrix}|\mathbf{v}_{t|j} = \begin{bmatrix}1\\0\end{bmatrix}) = (\tilde{C}_{1,t}\cdot q_j(0|1,0) + C_{1,t}\cdot q_j(1|1,0))(\tilde{C}_{1,t}\cdot q_j(1|1,0) + C_{1,t}\cdot q_j(0|1,0))$$

$$p_\phi(\mathbf{v}_{t-1|j} = \begin{bmatrix}0\\1\end{bmatrix}|\mathbf{v}_{t|j} = \begin{bmatrix}1\\0\end{bmatrix}) = (\tilde{C}_{2,t}\cdot q_j(0|1,0) + C_{2,t}\cdot q_j(1|1,0))(\tilde{C}_{2,t}\cdot q_j(1|1,0) + C_{2,t}\cdot q_j(0|1,0))$$

$$p_\phi(\mathbf{v}_{t-1|j} = \begin{bmatrix}0\\0\end{bmatrix}|\mathbf{v}_{t|j} = \begin{bmatrix}0\\1\end{bmatrix}) = (\tilde{C}_{1,t}\cdot q_j(1|1,0) + C_{1,t}\cdot q_j(0|1,0))(\tilde{C}_{2,t}\cdot q_j(0|1,0) + C_{2,t}\cdot q_j(1|1,0))$$

$$p_\phi(\mathbf{v}_{t-1|j} = \begin{bmatrix}1\\1\end{bmatrix}|\mathbf{v}_{t|j} = \begin{bmatrix}0\\1\end{bmatrix}) = (\tilde{C}_{1,t}\cdot q_j(0|1,0) + C_{1,t}\cdot q_j(1|1,0))(\tilde{C}_{2,t}\cdot q_j(1|1,0) + C_{2,t}\cdot q_j(0|1,0))$$

$$p_\phi(\mathbf{v}_{t-1|j} = \begin{bmatrix}1\\0\end{bmatrix}|\mathbf{v}_{t|j} = \begin{bmatrix}0\\1\end{bmatrix}) = (\tilde{C}_{2,t}\cdot q_j(0|1,0) + C_{2,t}\cdot q_j(1|1,0))(\tilde{C}_{2,t}\cdot q_j(1|1,0) + C_{2,t}\cdot q_j(0|1,0))$$

$$p_\phi(\mathbf{v}_{t-1|j} = \begin{bmatrix}0\\1\end{bmatrix}|\mathbf{v}_{t|j} = \begin{bmatrix}0\\1\end{bmatrix}) = (\tilde{C}_{1,t}\cdot q_j(0|1,0) + C_{1,t}\cdot q_j(1|1,0))(\tilde{C}_{1,t}\cdot q_j(1|1,0) + C_{1,t}\cdot q_j(0|1,0))$$

With above the circumstances discussed above, we define $H_{t-1} = p_\phi(\mathbf{v}_{t-1|0} = \begin{bmatrix}1\\1\end{bmatrix}) - p_\phi(\mathbf{v}_{t-1|1} = \begin{bmatrix}1\\1\end{bmatrix})$, we have

$$H_{t-1} = p_\phi(\mathbf{v}_{t-1|0} = \begin{bmatrix}1\\1\end{bmatrix}|\mathbf{v}_{t|0} = \begin{bmatrix}1\\1\end{bmatrix})p_\phi(\mathbf{v}_{t|0} = \begin{bmatrix}1\\1\end{bmatrix}) + p_\phi(\mathbf{v}_{t-1|0} = \begin{bmatrix}1\\1\end{bmatrix}|\mathbf{v}_{t|0} = \begin{bmatrix}0\\0\end{bmatrix})p_\phi(\mathbf{v}_{t|0} = \begin{bmatrix}0\\0\end{bmatrix})$$

$$+ p_\phi(\mathbf{v}_{t-1|0} = \begin{bmatrix}1\\1\end{bmatrix}|\mathbf{v}_{t|0} = \begin{bmatrix}1\\0\end{bmatrix})p_\phi(\mathbf{v}_{t|0} = \begin{bmatrix}1\\0\end{bmatrix}) + p_\phi(\mathbf{v}_{t-1|0} = \begin{bmatrix}1\\1\end{bmatrix}|\mathbf{v}_{t|0} = \begin{bmatrix}0\\1\end{bmatrix})p_\phi(\mathbf{v}_{t|0} = \begin{bmatrix}0\\1\end{bmatrix})$$

$$- p_\phi(\mathbf{v}_{t-1|1} = \begin{bmatrix} 1 \\ 1 \end{bmatrix} | \mathbf{v}_{t|1} = \begin{bmatrix} 1 \\ 1 \end{bmatrix}) p_\phi(\mathbf{v}_{t|1} = \begin{bmatrix} 1 \\ 1 \end{bmatrix}) + p_\phi(\mathbf{v}_{t-1|1} = \begin{bmatrix} 1 \\ 1 \end{bmatrix} | \mathbf{v}_{t|1} = \begin{bmatrix} 0 \\ 0 \end{bmatrix}) p_\phi(\mathbf{v}_{t|1} = \begin{bmatrix} 0 \\ 0 \end{bmatrix})$$

$$- p_\phi(\mathbf{v}_{t-1|1} = \begin{bmatrix} 1 \\ 1 \end{bmatrix} | \mathbf{v}_{t|1} = \begin{bmatrix} 1 \\ 0 \end{bmatrix}) p_\phi(\mathbf{v}_{t|1} = \begin{bmatrix} 1 \\ 0 \end{bmatrix}) + p_\phi(\mathbf{v}_{t-1|1} = \begin{bmatrix} 1 \\ 1 \end{bmatrix} | \mathbf{v}_{t|1} = \begin{bmatrix} 0 \\ 1 \end{bmatrix}) p_\phi(\mathbf{v}_{t|1} = \begin{bmatrix} 0 \\ 1 \end{bmatrix}) \quad (209)$$

$$= p_\phi(\mathbf{v}_{t-1|0} = \begin{bmatrix} 1 \\ 1 \end{bmatrix} | \mathbf{v}_{t|0} = \begin{bmatrix} 1 \\ 1 \end{bmatrix})(p_\phi(\mathbf{v}_{t|0} = \begin{bmatrix} 1 \\ 1 \end{bmatrix}) - p_\phi(\mathbf{v}_{t|1} = \begin{bmatrix} 1 \\ 1 \end{bmatrix}))$$

$$+ p_\phi(\mathbf{v}_{t|1} = \begin{bmatrix} 1 \\ 1 \end{bmatrix})(p_\phi(\mathbf{v}_{t-1|0} = \begin{bmatrix} 1 \\ 1 \end{bmatrix} | \mathbf{v}_{t|0} = \begin{bmatrix} 1 \\ 1 \end{bmatrix}) - p_\phi(\mathbf{v}_{t-1|1} = \begin{bmatrix} 1 \\ 1 \end{bmatrix} | \mathbf{v}_{t|1} = \begin{bmatrix} 1 \\ 1 \end{bmatrix}))$$

$$+ p_\phi(\mathbf{v}_{t-1|0} = \begin{bmatrix} 1 \\ 1 \end{bmatrix} | \mathbf{v}_{t|0} = \begin{bmatrix} 0 \\ 0 \end{bmatrix})(p_\phi(\mathbf{v}_{t|0} = \begin{bmatrix} 0 \\ 0 \end{bmatrix}) - p_\phi(\mathbf{v}_{t|1} = \begin{bmatrix} 0 \\ 0 \end{bmatrix}))$$

$$+ p_\phi(\mathbf{v}_{t|1} = \begin{bmatrix} 0 \\ 0 \end{bmatrix})(p_\phi(\mathbf{v}_{t-1|0} = \begin{bmatrix} 1 \\ 1 \end{bmatrix} | \mathbf{v}_{t|0} = \begin{bmatrix} 0 \\ 0 \end{bmatrix}) - p_\phi(\mathbf{v}_{t-1|1} = \begin{bmatrix} 1 \\ 1 \end{bmatrix} | \mathbf{v}_{t|1} = \begin{bmatrix} 0 \\ 0 \end{bmatrix}))$$

$$+ p_\phi(\mathbf{v}_{t-1|0} = \begin{bmatrix} 1 \\ 1 \end{bmatrix} | \mathbf{v}_{t|0} = \begin{bmatrix} 1 \\ 0 \end{bmatrix})(p_\phi(\mathbf{v}_{t|0} = \begin{bmatrix} 1 \\ 0 \end{bmatrix}) - p_\phi(\mathbf{v}_{t|1} = \begin{bmatrix} 1 \\ 0 \end{bmatrix}))$$

$$+ p_\phi(\mathbf{v}_{t|1} = \begin{bmatrix} 1 \\ 0 \end{bmatrix})(p_\phi(\mathbf{v}_{t-1|0} = \begin{bmatrix} 1 \\ 1 \end{bmatrix} | \mathbf{v}_{t|0} = \begin{bmatrix} 1 \\ 0 \end{bmatrix}) - p_\phi(\mathbf{v}_{t-1|1} = \begin{bmatrix} 1 \\ 1 \end{bmatrix} | \mathbf{v}_{t|1} = \begin{bmatrix} 1 \\ 0 \end{bmatrix}))$$

$$+ p_\phi(\mathbf{v}_{t-1|0} = \begin{bmatrix} 1 \\ 1 \end{bmatrix} | \mathbf{v}_{t|0} = \begin{bmatrix} 0 \\ 1 \end{bmatrix})(p_\phi(\mathbf{v}_{t|0} = \begin{bmatrix} 0 \\ 1 \end{bmatrix}) - p_\phi(\mathbf{v}_{t|1} = \begin{bmatrix} 0 \\ 1 \end{bmatrix}))$$

$$+ p_\phi(\mathbf{v}_{t|1} = \begin{bmatrix} 0 \\ 1 \end{bmatrix})(p_\phi(\mathbf{v}_{t-1|0} = \begin{bmatrix} 1 \\ 1 \end{bmatrix} | \mathbf{v}_{t|0} = \begin{bmatrix} 0 \\ 1 \end{bmatrix}) - p_\phi(\mathbf{v}_{t-1|1} = \begin{bmatrix} 1 \\ 1 \end{bmatrix} | \mathbf{v}_{t|1} = \begin{bmatrix} 0 \\ 1 \end{bmatrix}))$$

$$(210)$$

Since $p_\phi(\mathbf{v}_{t|0} = \begin{bmatrix} 1 \\ 1 \end{bmatrix}) - p_\phi(\mathbf{v}_{t|1} = \begin{bmatrix} 1 \\ 1 \end{bmatrix}) = -\sum_{\mathbf{v}_t \neq [1,1]^T}(p_\phi(\mathbf{v}_{t|0} = \mathbf{v}_t) - p_\phi(\mathbf{v}_{t|1} = \mathbf{v}_t))$ and $\max\Big\{ p_\phi(\mathbf{v}_{t-1|0} = \begin{bmatrix} 1 \\ 1 \end{bmatrix} | \mathbf{v}_{t|0} = \begin{bmatrix} 0 \\ 0 \end{bmatrix}), p_\phi(\mathbf{v}_{t-1|0} = \begin{bmatrix} 1 \\ 1 \end{bmatrix} | \mathbf{v}_{t|0} = \begin{bmatrix} 1 \\ 0 \end{bmatrix}), p_\phi(\mathbf{v}_{t-1|0} = \begin{bmatrix} 1 \\ 1 \end{bmatrix} | \mathbf{v}_{t|0} = \begin{bmatrix} 0 \\ 1 \end{bmatrix})\Big\} = p_\phi(\mathbf{v}_{t-1|0} = \begin{bmatrix} 1 \\ 1 \end{bmatrix} | \mathbf{v}_{t|0} = \begin{bmatrix} 1 \\ 0 \end{bmatrix}) = p_\phi(\mathbf{v}_{t-1|0} = \begin{bmatrix} 1 \\ 1 \end{bmatrix} | \mathbf{v}_{t|0} = \begin{bmatrix} 0 \\ 1 \end{bmatrix})$, we have

$$H_{t-1} \geq \left[ p_\phi(\mathbf{v}_{t-1|0} = \begin{bmatrix} 1 \\ 1 \end{bmatrix} | \mathbf{v}_{t|0} = \begin{bmatrix} 1 \\ 1 \end{bmatrix}) - p_\phi(\mathbf{v}_{t-1|0} = \begin{bmatrix} 1 \\ 1 \end{bmatrix} | \mathbf{v}_{t|0} = \begin{bmatrix} 1 \\ 0 \end{bmatrix}) \right] \cdot H_t$$

$$+ \min_{\mathbf{v}_t} \left[ p_\phi(\mathbf{v}_{t-1|0} = \begin{bmatrix} 1 \\ 1 \end{bmatrix} | \mathbf{v}_{t|0} = \mathbf{v}_t) - p_\phi(\mathbf{v}_{t-1|1} = \begin{bmatrix} 1 \\ 1 \end{bmatrix} | \mathbf{v}_{t|1} = \mathbf{v}_t) \right] \quad (211)$$

For $\min_{\mathbf{v}_t} \left[ p_\phi(\mathbf{v}_{t-1|0} = \begin{bmatrix} 1 \\ 1 \end{bmatrix} | \mathbf{v}_{t|0} = \mathbf{v}_t) - p_\phi(\mathbf{v}_{t-1|1} = \begin{bmatrix} 1 \\ 1 \end{bmatrix} | \mathbf{v}_{t|1} = \mathbf{v}_t) \right]$, we have

$$p_\phi(\mathbf{v}_{t-1|0} = \begin{bmatrix} 1 \\ 1 \end{bmatrix} | \mathbf{v}_{t|0} = \begin{bmatrix} 0 \\ 0 \end{bmatrix}) - p_\phi(\mathbf{v}_{t-1|1} = \begin{bmatrix} 1 \\ 1 \end{bmatrix} | \mathbf{v}_{t|1} = \begin{bmatrix} 0 \\ 0 \end{bmatrix}) \quad (212)$$

$$= \frac{[(s-1)(\tilde{C}_{2,t} - C_{2,t})(\bar{\mu}_t^+)^2(\bar{\mu}_t^-)^2][2(s-1)^2 C_{2,t}(\bar{\mu}_t^+)^4 + 3(s-1)(C_{2,t} + \tilde{C}_{2,t})(\bar{\mu}_t^+)^2(\bar{\mu}_t^-)^2 + 4\tilde{C}_{2,t}(\bar{\mu}_t^-)^4]}{[(s-1)(\bar{\mu}_t^+)^2 + 2(\bar{\mu}_t^-)^2]^2[(s-1)(\bar{\mu}_t^+)^2 + (\bar{\mu}_t^-)^2]^2}$$

$$(213)$$

$$p_\phi(\mathbf{v}_{t-1|0} = \begin{bmatrix} 1 \\ 1 \end{bmatrix} | \mathbf{v}_{t|0} = \begin{bmatrix} 1 \\ 1 \end{bmatrix}) - p_\phi(\mathbf{v}_{t-1|1} = \begin{bmatrix} 1 \\ 1 \end{bmatrix} | \mathbf{v}_{t|1} = \begin{bmatrix} 1 \\ 1 \end{bmatrix}) \quad (214)$$

$$= \frac{[(s-1)(C_{1,t} - \tilde{C}_{1,t})(\bar{\mu}_t^+)^2(\bar{\mu}_t^-)^2][2(s-1)^2 C_{1,t}(\bar{\mu}_t^+)^4 + 3(s-1)(C_{1,t} + \tilde{C}_{1,t})(\bar{\mu}_t^+)^2(\bar{\mu}_t^-)^2 + 4\tilde{C}_{1,t}(\bar{\mu}_t^+)^4]}{[(s-1)(\bar{\mu}_t^-)^2 + 2(\bar{\mu}_t^+)^2]^2[(s-1)(\bar{\mu}_t^-)^2 + (\bar{\mu}_t^+)^2]^2}$$

$$(215)$$

$$p_\phi(\mathbf{v}_{t-1|0} = \begin{bmatrix} 1 \\ 1 \end{bmatrix} | \mathbf{v}_{t|0} = \begin{bmatrix} 1 \\ 1 \end{bmatrix}) - p_\phi(\mathbf{v}_{t-1|0} = \begin{bmatrix} 1 \\ 1 \end{bmatrix} | \mathbf{v}_{t|1} = \begin{bmatrix} 1 \\ 0 \end{bmatrix}) \quad (216)$$

$$= \frac{(s-1)\{(s^2 - 2s - 1)[(C_{1,t} \cdot C_{2,t} - \tilde{C}_{1,t} \cdot C_{2,t}) + (\tilde{C}_{1,t} \cdot \tilde{C}_{2,t} - \tilde{C}_{1,t} \cdot C_{2,t})] + (3s+1)(C_{1,t} \cdot \tilde{C}_{2,t} - \tilde{C}_{1,t} \cdot \tilde{C}_{2,t})\}}{s^2(s+1)^2}$$

(217)

From detailed calculation, we can show that

$$\min_{\mathbf{v}_t} \left[ p_\phi(\mathbf{v}_{t-1|0} = \begin{bmatrix} 1 \\ 1 \end{bmatrix} | \mathbf{v}_{t|0} = \mathbf{v}_t) - p_\phi(\mathbf{v}_{t-1|1} = \begin{bmatrix} 1 \\ 1 \end{bmatrix} | \mathbf{v}_{t|1} = \mathbf{v}_t) \right]$$

(218)

$$= \frac{[(s-1)(\tilde{C}_{2,t} - C_{2,t})(\bar{\mu}_t^+)^2(\bar{\mu}_t^-)^2][2(s-1)^2 C_{2,t}(\bar{\mu}_t^+)^4 + 3(s-1)(C_{2,t} + \tilde{C}_{2,t})(\bar{\mu}_t^+)^2(\bar{\mu}_t^-)^2 + 4\tilde{C}_{2,t}(\bar{\mu}_t^-)^4]}{[(s-1)(\bar{\mu}_t^+)^2 + 2(\bar{\mu}_t^-)^2]^2[(s-1)(\bar{\mu}_t^+)^2 + (\bar{\mu}_t^-)^2]^2}$$

(219)

$$:= G_t$$

(220)

For $t = 0$, we have

$$p_\phi(\mathbf{v}_{1|1} = \begin{bmatrix} 1 \\ 1 \end{bmatrix})(p_\phi(\mathbf{v}_{0|0} = \begin{bmatrix} 1 \\ 1 \end{bmatrix} | \mathbf{v}_{1|0} = \begin{bmatrix} 1 \\ 1 \end{bmatrix}) - p_\phi(\mathbf{v}_{0|1} = \begin{bmatrix} 1 \\ 1 \end{bmatrix} | \mathbf{v}_{1|1} = \begin{bmatrix} 1 \\ 1 \end{bmatrix}))$$

$$+ p_\phi(\mathbf{v}_{1|1} = \begin{bmatrix} 0 \\ 0 \end{bmatrix})(p_\phi(\mathbf{v}_{0|0} = \begin{bmatrix} 1 \\ 1 \end{bmatrix} | \mathbf{v}_{1|0} = \begin{bmatrix} 0 \\ 0 \end{bmatrix}) - p_\phi(\mathbf{v}_{0|1} = \begin{bmatrix} 1 \\ 1 \end{bmatrix} | \mathbf{v}_{1|1} = \begin{bmatrix} 0 \\ 0 \end{bmatrix}))$$

$$+ p_\phi(\mathbf{v}_{1|1} = \begin{bmatrix} 1 \\ 0 \end{bmatrix})(p_\phi(\mathbf{v}_{0|0} = \begin{bmatrix} 1 \\ 1 \end{bmatrix} | \mathbf{v}_{1|0} = \begin{bmatrix} 1 \\ 0 \end{bmatrix}) - p_\phi(\mathbf{v}_{0|1} = \begin{bmatrix} 1 \\ 1 \end{bmatrix} | \mathbf{v}_{1|1} = \begin{bmatrix} 1 \\ 0 \end{bmatrix}))$$

$$+ p_\phi(\mathbf{v}_{1|1} = \begin{bmatrix} 0 \\ 1 \end{bmatrix})(p_\phi(\mathbf{v}_{0|0} = \begin{bmatrix} 1 \\ 1 \end{bmatrix} | \mathbf{v}_{1|0} = \begin{bmatrix} 0 \\ 1 \end{bmatrix}) - p_\phi(\mathbf{v}_{0|1} = \begin{bmatrix} 1 \\ 1 \end{bmatrix} | \mathbf{v}_{1|1} = \begin{bmatrix} 0 \\ 1 \end{bmatrix}))$$

(221)

$$= p_\phi(\mathbf{v}_{1|1} = \begin{bmatrix} 0 \\ 0 \end{bmatrix}) \cdot \frac{1}{s\bar{R}_1^2} + p_\phi(\mathbf{v}_{1|1} = \begin{bmatrix} 0 \\ 0 \end{bmatrix}) \cdot \frac{2}{s} + p_\phi(\mathbf{v}_{1|1} = \begin{bmatrix} 1 \\ 1 \end{bmatrix}) \cdot \frac{\bar{R}_1^2}{s}$$

(222)

$$\geq \frac{1}{s\bar{R}_1^2} =: G_0$$

(223)

Let $F_t := p_\phi(\mathbf{v}_{t-1|0} = \begin{bmatrix} 1 \\ 1 \end{bmatrix} | \mathbf{v}_{t|0} = \begin{bmatrix} 1 \\ 1 \end{bmatrix}) - p_\phi(\mathbf{v}_{t-1|0} = \begin{bmatrix} 1 \\ 1 \end{bmatrix} | \mathbf{v}_{t|0} = \begin{bmatrix} 1 \\ 0 \end{bmatrix})$, we have

$$F_t = \frac{4(\bar{\mu}_t^+)^4 A + 4(s-1)(\bar{\mu}_t^+)^2(\bar{\mu}_t^-)^2 B + (s-1)^2(\bar{\mu}_t^-)^4 C}{(s+1)^2[2(\bar{\mu}_t^+)^2 + (s-1)(\bar{\mu}_t^-)^2]^2}$$

(224)

where

$$A = (s+1)^2 C_{1,t}^2 - [(s-1)\tilde{C}_{1,t} + 2C_{1,t}][(s-1)C_{1,t} + 2\tilde{C}_{2,t}],$$

(225)

$$B = (s+1)^2 \tilde{C}_{1,t} \cdot C_{1,t} - [(s-1)\tilde{C}_{1,t} + 2C_{1,t}][(s-1)C_{1,t} + 2\tilde{C}_{2,t}],$$

(226)

$$C = (s+1)^2 \tilde{C}_{1,t}^2 - [(s-1)\tilde{C}_{1,t} + 2C_{1,t}][(s-1)C_{1,t} + 2\tilde{C}_{2,t}]$$

(227)

Given that $H_T = 0$, and by iteratively applying the inequality $H_{t-1} \geq G_t + F_t H_t$, we arrive at the desired conclusion. $\square$

## J.8 Proof of Lemma D.2

*Proof.* Since $q(\mathbf{v}_{T|1} = \begin{bmatrix} 0 \\ 0 \end{bmatrix}) = q(\mathbf{v}_{T|1} = \begin{bmatrix} 1 \\ 1 \end{bmatrix}) = q(\mathbf{v}_{T|1} = \begin{bmatrix} 1 \\ 0 \end{bmatrix}) = q(\mathbf{v}_{T|1} = \begin{bmatrix} 0 \\ 1 \end{bmatrix}) = \frac{1}{4}$. From induction and symmetric properties, we obtain that for any $t$, we have

$$q(\mathbf{v}_{t|1} = \begin{bmatrix} 0 \\ 0 \end{bmatrix}) \geq q(\mathbf{v}_{t|1} = \begin{bmatrix} 1 \\ 0 \end{bmatrix}) = q(\mathbf{v}_{t|1} = \begin{bmatrix} 0 \\ 1 \end{bmatrix}) \geq q(\mathbf{v}_{t|1} = \begin{bmatrix} 1 \\ 1 \end{bmatrix})$$

(228)

On the other hand,

$$p_\phi(\mathbf{v}_{t-1|1} = \begin{bmatrix} 1 \\ 1 \end{bmatrix} | \mathbf{v}_{t|1} = \begin{bmatrix} 0 \\ 0 \end{bmatrix}) < p_\phi(\mathbf{v}_{t-1|1} = \begin{bmatrix} 1 \\ 1 \end{bmatrix} | \mathbf{v}_{t|1} = \begin{bmatrix} 1 \\ 0 \end{bmatrix}) = p_\phi(\mathbf{v}_{t-1|1} = \begin{bmatrix} 1 \\ 1 \end{bmatrix} | \mathbf{v}_{t|1} = \begin{bmatrix} 0 \\ 1 \end{bmatrix})$$

$$< p_\phi(\mathbf{v}_{t-1|1} = \begin{bmatrix} 1 \\ 1 \end{bmatrix} \bigg| \mathbf{v}_{t|1} = \begin{bmatrix} 1 \\ 1 \end{bmatrix}) \tag{229}$$

Thus, using Chebyshev's inequality,

$$p_\phi(\mathbf{v}_{0|1} = \begin{bmatrix} 1 \\ 1 \end{bmatrix}) = \sum_{\mathbf{v}_1} p_\phi(\mathbf{v}_{0|1} = \begin{bmatrix} 1 \\ 1 \end{bmatrix} \bigg| \mathbf{v}_{1|1} = \mathbf{v}_1) p_\phi(\mathbf{v}_{1|1} = \mathbf{v}_1) \tag{230}$$

$$\leq p_\phi(\mathbf{v}_{0|1} = \begin{bmatrix} 1 \\ 1 \end{bmatrix} \bigg| \mathbf{v}_{1|1} = \begin{bmatrix} 1 \\ 0 \end{bmatrix}) + p_\phi(\mathbf{v}_{0|1} = \begin{bmatrix} 1 \\ 1 \end{bmatrix} \bigg| \mathbf{v}_{1|1} = \begin{bmatrix} 1 \\ 1 \end{bmatrix}) p_\phi(\mathbf{v}_{1|1} = \begin{bmatrix} 1 \\ 1 \end{bmatrix}) \tag{231}$$

$$\leq \frac{1}{s} + \frac{\bar{R}_1^2}{\bar{R}_1^2 + s} \cdot \underbrace{(\frac{1}{4} \min_t \{(\sum_{\mathbf{v}_1} p_\phi(\mathbf{v}_{t-1|1} = \begin{bmatrix} 1 \\ 1 \end{bmatrix} \bigg| \mathbf{v}_{t|1} = \mathbf{v}_1))\})}_{\Delta} \tag{232}$$

$$= \Theta_s(\frac{1}{s}) \tag{233}$$

where from proof of Lemma D.1, we have

$$\Delta = \min_t \frac{1}{4} [\frac{C_{2,t} \cdot s \cdot \bar{R}_t^2}{s \cdot \bar{R}_t^2 + 1} + \frac{\tilde{C}_{2,t}}{s\bar{R}_t^2 + 1} + \frac{\tilde{C}_{1,t} \cdot s}{s + \bar{R}_t^2} + \frac{C_{1,t} \cdot \bar{R}_t^2}{s + \bar{R}_t^2} + 2(\frac{\tilde{C}_{1,t} \cdot (s-1)}{s} + \frac{C_{1,t\cdot}}{s})(\frac{C_{2,t} \cdot (s-1)}{s} + \frac{\tilde{C}_{2,t}}{s})]. \tag{234}$$

$\square$

## K   Additional Lemmas and Proofs

**Lemma K.1** (Monotonicity of KL Divergence). Let $P_{X_1,X_2,\ldots,X_T}, Q_{X_1,X_2,\ldots,X_T}$ be probability measures over random variables $X_1, X_2, \ldots, X_T$, where $P_{X_1}, Q_{X_1}$ are the marginal measures of $P_{X_1,X_2,\ldots,X_T}$ and $Q_{X_1,X_2,\ldots,X_T}$, respectively. Then, by the monotonicity property of KL divergence, we have:

$$\mathcal{D}_{\mathrm{KL}}(P_{X_1} \| Q_{X_1}) \leq \mathcal{D}_{\mathrm{KL}}(P_{X_1,X_2,X_3,\ldots,X_T} \| Q_{X_1,X_2,\ldots,X_T}) \tag{235}$$

*Proof of Lemma K.1.* The proof refer to (Polyanskiy, 2020). $\square$

**Lemma K.2** (Boundedness of Posterior Distribution). The posterior distribution in discrete diffusion process is given as $q(\mathbf{v}_{t-1}|\mathbf{v}_t, \mathbf{v}_0) = \mathrm{Cat}(\mathbf{v}_{t-1}; \mathbf{v}_t Q_t^T \odot \mathbf{v}_0 \overline{Q}_{t-1} / \mathbf{v}_0 \overline{Q}_t \mathbf{v}_t^T)$ such that

$$\max_{\mathbf{v}_0, \mathbf{v}_{t-1}, \mathbf{v}_t} q(\mathbf{v}_{t-1}|\mathbf{v}_t, \mathbf{v}_0) = \frac{\mu_t^+ \cdot \bar{\mu}_{t-1}^+}{\bar{\mu}_t^+}, \quad \min_{\mathbf{v}_0, \mathbf{v}_{t-1}, \mathbf{v}_t} q(\mathbf{v}_{t-1}|\mathbf{v}_t, \mathbf{v}_0) = \frac{\mu_t^- \cdot \bar{\mu}_{t-1}^-}{\bar{\mu}_t^+}. \tag{236}$$

*Proof of Lemma K.2.* Since $\overline{Q}_t = \overline{\alpha}_t I + (1 - \overline{\alpha}_t)\frac{\mathbb{1}\mathbb{1}^T}{k}$, from definition, given $l, j \in \{1, 2, ..., k\}$

$$q(\mathbf{v}_{t-1}|\mathbf{v}_t = l, \mathbf{v}_0 = j) = \frac{[\alpha_t I + (1 - \alpha_t)\frac{\mathbb{1}\mathbb{1}^T}{k}]_{l,\cdot} \odot [\overline{\alpha}_{t-1} I + (1 - \overline{\alpha}_{t-1})\frac{\mathbb{1}\mathbb{1}^T}{k}]_{j,\cdot}}{[\overline{\alpha}_t I + (1 - \overline{\alpha}_t)\frac{\mathbb{1}\mathbb{1}^T}{k}]_{j,l}} \tag{237}$$

$$[\alpha_t I + (1 - \alpha_t)\frac{\mathbb{1}\mathbb{1}^T}{k}]_{l,\cdot} = [\mu_t^-, \mu_t^-, ..., \mu_t^-, \mu_t^+, \mu_t^-, ..., \mu_t^-] \tag{238}$$

$$[\overline{\alpha}_{t-1} I + (1 - \overline{\alpha}_{t-1})\frac{\mathbb{1}\mathbb{1}^T}{k}]_{l,\cdot} = [\bar{\mu}_{t-1}^-, \bar{\mu}_{t-1}^-, ..., \bar{\mu}_{t-1}^-, \bar{\mu}_{t-1}^+, \bar{\mu}_{t-1}^-, ..., \bar{\mu}_{t-1}^-] \tag{239}$$

1. When $l = j$, we have

$$[\alpha_t I + (1 - \alpha_t)\frac{\mathbb{1}\mathbb{1}^T}{k}]_{l,\cdot} \odot [\overline{\alpha}_{t-1} I + (1 - \overline{\alpha}_{t-1})\frac{\mathbb{1}\mathbb{1}^T}{k}]_{j,\cdot}$$

$$= [\underbrace{\mu_t^- \cdot \bar{\mu}_{t-1}^-, ..., \mu_t^- \cdot \bar{\mu}_{t-1}^-}_{l-1 \text{ terms}}, \mu_t^+ \cdot \bar{\mu}_{t-1}^+, \mu_t^- \cdot \bar{\mu}_{t-1}^-, ...] \tag{240}$$

Besides,

$$[\bar{\alpha}_t I + (1 - \bar{\alpha}_t) \frac{\mathbb{1}\mathbb{1}^T}{k}]_{l,j} = \bar{\mu}_t^+ \tag{241}$$

Therefore, we have the explicit form for $q(\mathbf{v}_{t-1}|\mathbf{v} = l, \mathbf{v}_0 = j)$

$$q(\mathbf{v}_{t-1}|\mathbf{v}_t = l, \mathbf{v}_0 = j) = [\frac{\mu_t^- \cdot \bar{\mu}_{t-1}^-}{\bar{\mu}_t^+}, ..., \frac{\mu_t^- \cdot \bar{\mu}_{t-1}^-}{\bar{\mu}_t^+}, \frac{\mu_t^+ \cdot \bar{\mu}_{t-1}^+}{\bar{\mu}_t^+}, \frac{\mu_t^- \cdot \bar{\mu}_{t-1}^-}{\bar{\mu}_t^+}, ..., \frac{\mu_t^- \cdot \bar{\mu}_{t-1}^-}{\bar{\mu}_t^+}] \tag{242}$$

2. When $l \neq j$, we have

$$[\alpha_t I + (1 - \alpha_t) \frac{\mathbb{1}\mathbb{1}^T}{k}]_{l,\cdot} \odot [\bar{\alpha}_{t-1} I + (1 - \bar{\alpha}_{t-1}) \frac{\mathbb{1}\mathbb{1}^T}{k}]_{j,\cdot}$$
$$= [\underbrace{\mu_t^- \cdot \bar{\mu}_{t-1}^-, ..., \mu_t^- \cdot \bar{\mu}_{t-1}^-}_{l-1 \text{ terms}}, \mu_t^+ \cdot \bar{\mu}_t^-, \underbrace{..., \mu_t^- \cdot \bar{\mu}_{t-1}^-}_{j-l-1 \text{ terms}}, ..., \mu_t^- \cdot \bar{\mu}_{t-1}^+, ..., \mu_t^- \cdot \bar{\mu}_{t-1}^-] \tag{243}$$

$$[\bar{\alpha}_t I + (1 - \bar{\alpha}_t) \frac{\mathbb{1}\mathbb{1}^T}{k}]_{l,j} = \bar{\mu}_t^- \tag{244}$$

The explicit form of $q(\mathbf{v}_{t-1}|\mathbf{v}_t = l, \mathbf{v}_0 = j)$ is given as:

$$q(\mathbf{v}_{t-1}|\mathbf{v}_t = l, \mathbf{v}_0 = j) = [\frac{\mu_t^- \cdot \bar{\mu}_{t-1}^-}{\bar{\mu}_t^-}, ..., \frac{\mu_t^+ \cdot \bar{\mu}_{t-1}^-}{\bar{\mu}_t^-}, ..., \frac{\mu_t^- \cdot \bar{\mu}_{t-1}^+}{\bar{\mu}_t^-}, ...] \tag{245}$$

From above derivation, we can show that

$$\max_{\mathbf{v}_0, \mathbf{v}_{t-1}, \mathbf{v}_t} q(\mathbf{v}_{t-1}|\mathbf{v}_t, \mathbf{v}_0) = \frac{\mu_t^+ \cdot \bar{\mu}_{t-1}^+}{\bar{\mu}_t^+}, \quad \min_{\mathbf{v}_0, \mathbf{v}_{t-1}, \mathbf{v}_t} q(\mathbf{v}_{t-1}|\mathbf{v}_t, \mathbf{v}_0) = \frac{\mu_t^- \cdot \bar{\mu}_{t-1}^-}{\bar{\mu}_t^+}. \tag{246}$$

$\square$

**Lemma K.3.** Given a positive bounded trivariate kernel $\mathcal{K}(x, \tilde{x}, z)$ such that $\mathcal{K}(x, \tilde{x}, z) \in [c_1, c_2]$ and the conditional probabilities $\tilde{q}_i(\tilde{x}|z)$ and $\tilde{p}_i(\tilde{x}|z)$ satisfy

$$\mathcal{D}_{\text{KL}}(\tilde{q}_i(\tilde{x}|z) \| \tilde{p}_i(\tilde{x}|z)) \leq \gamma, i \in \{0, 1\}, \forall z.$$

Define $q_i(x|z)$ and $p_i(x|z)$ as

$$q_i(x|z) = \int_{\tilde{x}} \mathcal{K}(x, \tilde{x}, z) \tilde{q}_i(\tilde{x}|z) d\tilde{x}, \quad p_i(x|z) = \int_{\tilde{x}} \mathcal{K}(x, \tilde{x}, z) \tilde{p}_i(\tilde{x}|z) d\tilde{x}, \quad i \in \{0, 1\}. \tag{247}$$

we have

$$\mathbb{E}_z [\mathcal{D}_{\text{KL}}(p_0(x|z) \| p_1(x|z))] \leq \mathbb{E}_z [\mathcal{D}_{\text{KL}}(q_0(x|z) \| q_1(x|z))] + \frac{c_2^2 \cdot k}{c_1} \sqrt{\frac{\gamma}{2}} \tag{248}$$

*Proof of Lemma K.3.* From Pinsker's inequality, we have

$$\int_{\tilde{x}} |\tilde{q}_i(\tilde{x}|z) - \tilde{p}_i(\tilde{x}|z)| d\tilde{x} \leq \sqrt{\frac{\mathcal{D}_{\text{KL}}(\tilde{q}_i(\tilde{x}|z) \| \tilde{p}_i(\tilde{x}|z))}{2}} \leq \sqrt{\frac{\gamma}{2}}, \quad i \in \{0, 1\} \tag{249}$$

Since positive kernel $\mathcal{K}(x, \tilde{x}, z) \in [c_1, c_2]$, we have

$$|q_i(x|z) - p_i(x|z)| = \int_{\tilde{x}} \mathcal{K}(x, \tilde{x}, z)(\tilde{q}_i(\tilde{x}|z) - \tilde{p}_i(\tilde{x}|z)) \leq c_2 \sqrt{\frac{\gamma}{2}}$$

Thus, $\|q_i(x|z) - p_i(x|z)\|_{\text{TV}} \leq c_2 k \sqrt{\frac{\gamma}{2}}$. Now, for any given $z$, we consider $\int_x p_0(x|z) \log \frac{p_0(x|z)}{p_1(x|z)} - \int_x q_0(x|z) \log \frac{q_0(x|z)}{q_1(x|z)}$:

$$
\begin{aligned}
&\int_x p_0(x|z) \log \frac{p_0(x|z)}{p_1(x|z)} - \int_x q_0(x|z) \log \frac{q_0(x|z)}{q_1(x|z)} \\
&\leq \int \frac{(p_0(x|z) - q_0(x|z))(p_0(x|z) - p_1(x|z))}{p_1(x|z)} + \int q_0(x|z)(\log \frac{p_0(x|z)}{q_0(x|z)} + \log \frac{q_1(x|z)}{p_1(x|z)}) \\
&\leq \frac{c_2 - c_1}{c_1} \int |p_0(x|z) - q_0(x|z)| + \int |p_0(x|z) - q_0(x|z)| + \frac{c_2}{c_1} \int |q_1(x|z) - p_1(x|z)| \\
&= \frac{c_2}{c_1} (\int |p_0(x|z) - q_0(x|z)| + \int |p_1(x|z) - q_1(x|z)|) \\
&\leq \frac{c_2^2 \cdot k}{c_1} \sqrt{\frac{\gamma}{2}}
\end{aligned}
$$

We take expectation on both sides and we obtain the result. $\qquad\square$

**Lemma K.4.** If $\sum_i b_i = \sum_i a_i$, we have

$$
\sum_i (b_i - a_i) c_i \leq \frac{1}{2} \|b - a\|_{\ell_1} (\max c_i - \min c_i)
$$

*Proof of Lemma K.4.* First, WLOG we can assume $b_1 - a_1 \geq b_2 - a_2 \geq \ldots \geq b_r - a_r \geq 0 \geq b_{r+1} - a_{r+1} \geq \ldots \geq b_n - a_n$. Since $\sum_i b_i = \sum_i a_i$, we have $\sum_{j=1}^r (b_j - a_j) = -\sum_{s=r+1}^n (b_j - a_j) = \frac{1}{2} \|b - a\|_{l_1}$. Further, we have

$$
\sum_i (b_i - a_i) \cdot c_i = \sum_{j=1}^r (b_j - a_j) \cdot c_j + \sum_{s=r+1}^n (b_j - a_j) \cdot c_s \tag{250}
$$

$$
= \sum_{j=1}^r (b_j - a_j) \cdot c_j + \sum_{s=r+1}^n (-(b_j - a_j)) \cdot (-c_s) \tag{251}
$$

$$
\leq \max c_i \cdot \frac{1}{2} \|b - a\|_{l_1} + (-\min c_i) \cdot \frac{1}{2} \|b - a\|_{l_1} \tag{252}
$$

$$
= \frac{1}{2} \|b - a\|_{l_1} (\max c_i - \min c_i) \tag{253}
$$

$$\square$$

**Lemma K.5.** For $x, y, z \geq 0$,

$$
\frac{(a+1)y + z}{(ax + y + z)(ay + x + z + 1)} \leq \frac{a+1}{(a^2+1)x + ay + az + 1} \tag{254}
$$

*Proof of Lemma K.5.* We reformulate the above inequality into a polynomial division problem. Define a multivariate linear function $f(x, y, z) = M_1 x + M_2 y + M_3 z + M_4$ such that

$$
[(a+1)y + z] \cdot f(x, y, z) \leq (ax + y + z) \cdot (ay + x + z + 1) \tag{255}
$$

We expand the terms on both sides, we have

$$
\text{R.H.S} - \text{L.H.S} \tag{256}
$$

$$
\begin{aligned}
= &[a^2 - M_1(a+1)]xy + [a - M_2(a+1)]y^2 + ((a+1) - M_2 - M_3(a+1))yz \\
&+ ax^2 + xy + (a+1-M_1)xz + (1-M_3)z^2 + ax + [1 - M_4(a+1)]y + (1-M_4)z
\end{aligned} \tag{257}
$$

Let $M_1 = \frac{a^2+1}{a=1}$, $M_2 = \frac{a}{a+1}$, $M_3 = \frac{a}{a+1}$ and $M_4 = \frac{1}{a+1}$, we further have

$$\text{R.H.S} - \text{L.H.S} = \frac{1}{a+1}yz + ax^2 + \frac{2a}{a+1}xy + \frac{1}{a+1}z^2 + ax + \frac{a}{a+1} \geq 0 \tag{258}$$

Thus, we prove the inequality. $\qquad\square$

**Lemma K.6.** We upper bound the total variation of conditional probability gap as follows:

- If $\mathbf{v}_t^i = \mathbf{v}^{*i}$, we have

$$\|q(\mathbf{v}_{t-1|0}^i|\mathbf{v}_{t|0} = \mathbf{v}_t) - q(\mathbf{v}_{t-1|1}^i|\mathbf{v}_{t|1} = \mathbf{v}_t)\|_{l_1} \leq \frac{\mu_t^+ \cdot \frac{\bar{\mu}_{t-1}^+}{\bar{\mu}_t^+} \cdot (1 - \frac{\bar{R}_t}{\bar{R}_{t-1}})}{1 + \zeta(\mathcal{V}_1, \mathbf{v}^*, \mathbf{v}_t, t)} \tag{259}$$

- If $\mathbf{v}_t^i \neq \mathbf{v}^{*i}$, we have

$$\|q(\mathbf{v}_{t-1|0}^i|\mathbf{v}_{t|0} = \mathbf{v}_t) - q(\mathbf{v}_{t-1|1}^i|\mathbf{v}_{t|1} = \mathbf{v}_t)\|_{l_1} \leq \frac{\mu_t^+ \cdot \frac{\bar{\mu}_{t-1}^-}{\bar{\mu}_t^-} \cdot (\frac{\bar{R}_{t-1}}{\bar{R}_t} - 1)}{1 + \zeta(\mathcal{V}_1, \mathbf{v}^*, \mathbf{v}_t, t)} \tag{260}$$

*Proof of Lemma K.6.* 1. When $\mathbf{v}_t^i = \mathbf{v}^{*i}$, we have

$$\frac{1}{2}\|q(\mathbf{v}_{t-1|0}^i|\mathbf{v}_{t|0} = \mathbf{v}_t) - q(\mathbf{v}_{t-1|1}^i|\mathbf{v}_{t|1} = \mathbf{v}_t)\|_{l_1} \tag{261}$$

$$= \sum_{\mathbf{v}_{t-1}^i} (q(\mathbf{v}_{t-1|0}^i|\mathbf{v}_{t|0} = \mathbf{v}_t) - q(\mathbf{v}_{t-1|1}^i|\mathbf{v}_{t|1} = \mathbf{v}_t))_+ \tag{262}$$

$$= \sum_{\mathbf{v}_{t-1}^i=1}^k \frac{\sum_{\mathbf{v}_0 \in \mathcal{V}_1} \tau(\mathbf{v}^{*i}, \mathbf{v}_{t-1}^i, \mathbf{v}_t^i) \cdot (1 - \frac{\tau(\mathbf{v}_0^i, \mathbf{v}_{t-1}^i, \mathbf{v}_t^i)}{\tau(\mathbf{v}^{*i}, \mathbf{v}_{t-1}^i, \mathbf{v}_t^i)})_+ (\bar{R}_t)^{\bar{\omega}(\mathbf{v}^*, \mathbf{v}_t) - \bar{\omega}(\mathbf{v}_0, \mathbf{v}_t)}}{(\sum_{\mathbf{v}_0 \in \mathcal{V}_1}(\bar{R}_t)^{\bar{\omega}(\mathbf{v}^*, \mathbf{v}_t) - \bar{\omega}(\mathbf{v}_0, \mathbf{v}_t)})(1 + \sum_{\mathbf{v}_0 \in \mathcal{V}_1}(\bar{R}_t)^{\bar{\omega}(\mathbf{v}^*, \mathbf{v}_t) - \bar{\omega}(\mathbf{v}_0, \mathbf{v}_t)})} \tag{263}$$

$$\overset{(i)}{=} \tau(\mathbf{v}^{*i}, \mathbf{v}^{*i}, \mathbf{v}_t^i) \cdot \frac{\sum_{\mathbf{v}_0 \in \mathcal{V}_1}(1 - \frac{\tau(\mathbf{v}_0^i, \mathbf{v}^{*i}, \mathbf{v}_t^i)}{\tau(\mathbf{v}^{*i}, \mathbf{v}^{*i}, \mathbf{v}_t^i)})(\bar{R}_t)^{\bar{\omega}(\mathbf{v}^*, \mathbf{v}_t) - \bar{\omega}(\mathbf{v}_0, \mathbf{v}_t)}}{(\sum_{\mathbf{v}_0 \in \mathcal{V}_1}(\bar{R}_t)^{\bar{\omega}(\mathbf{v}^*, \mathbf{v}_t) - \bar{\omega}(\mathbf{v}_0, \mathbf{v}_t)})(1 + \sum_{\mathbf{v}_0 \in \mathcal{V}_1}(\bar{R}_t)^{\bar{\omega}(\mathbf{v}^*, \mathbf{v}_t) - \bar{\omega}(\mathbf{v}_0, \mathbf{v}_t)})} \tag{264}$$

$$= \frac{\mu_t^+ \cdot \frac{\bar{\mu}_{t-1}^+}{\bar{\mu}_t^+} \cdot (1 - \frac{\bar{R}_t}{\bar{R}_{t-1}}) \cdot \sum_{\mathbf{v}_0 \in \mathcal{V}_1, \mathbf{v}_0^i \neq \mathbf{v}^{*i}}(\bar{R}_t)^{\bar{\omega}(\mathbf{v}^*, \mathbf{v}_t) - \bar{\omega}(\mathbf{v}_0, \mathbf{v}_t)}}{(\sum_{\mathbf{v}_0 \in \mathcal{V}_1}(\bar{R}_t)^{\bar{\omega}(\mathbf{v}^*, \mathbf{v}_t) - \bar{\omega}(\mathbf{v}_0, \mathbf{v}_t)})(1 + \sum_{\mathbf{v}_0 \in \mathcal{V}_1}(\bar{R}_t)^{\bar{\omega}(\mathbf{v}^*, \mathbf{v}_t) - \bar{\omega}(\mathbf{v}_0, \mathbf{v}_t)})} \tag{265}$$

$$\leq \mu_t^+ \cdot \frac{\bar{\mu}_{t-1}^+}{\bar{\mu}_t^+} \cdot (1 - \frac{\bar{R}_t}{\bar{R}_{t-1}}) \cdot \frac{1}{1 + \zeta(\mathcal{V}_1, \mathbf{v}^*, \mathbf{v}_t, t)} \tag{266}$$

where $(i)$ is from that $\tau(\mathbf{v}_0^i, \mathbf{v}_{t-1}^i, \mathbf{v}_t^i)/\tau(\mathbf{v}^{*i}, \mathbf{v}_{t-1}^i, \mathbf{v}_t^i) < 1$ if and only if $\mathbf{v}_{t-1}^i = \mathbf{v}^{*i}$.

2. When $\mathbf{v}_t^i \neq \mathbf{v}^{*i}$, to simplify the notation, we abbreviate the terms as:

$$\sum_{\mathbf{v}_{t-1}^i}|\sum_{\mathbf{v}_0 \in \mathcal{V}_1}| := \sum_{\mathbf{v}_{t-1}^i=1}^k \left|\sum_{\mathbf{v}_0 \in \mathcal{V}_1} \frac{\tau(\mathbf{v}^{*i}, \mathbf{v}_{t-1}^i, \mathbf{v}_t^i) \cdot (1 - \frac{\tau(\mathbf{v}_0^i, \mathbf{v}_{t-1}^i, \mathbf{v}_t^i)}{\tau(\mathbf{v}^{*i}, \mathbf{v}_{t-1}^i, \mathbf{v}_t^i)})(\bar{R}_t)^{\bar{\omega}(\mathbf{v}^*, \mathbf{v}_t) - \bar{\omega}(\mathbf{v}_0, \mathbf{v}_t)}}{\zeta(\mathcal{V}_1, \mathbf{v}^*, \mathbf{v}_t, t)(1 + \zeta(\mathcal{V}_1, \mathbf{v}^*, \mathbf{v}_t, t))}\right|$$

$$\sum_{\mathbf{v}_{t-1}^i=\mathbf{v}^{*i}}|\sum_{\mathbf{v}_0 \in \mathcal{V}_1}| := \sum_{\mathbf{v}_{t-1}^i=\mathbf{v}^{*i}} \left|\sum_{\mathbf{v}_0 \in \mathcal{V}_1} \frac{\tau(\mathbf{v}^{*i}, \mathbf{v}_{t-1}^i, \mathbf{v}_t^i) \cdot (1 - \frac{\tau(\mathbf{v}_0^i, \mathbf{v}_{t-1}^i, \mathbf{v}_t^i)}{\tau(\mathbf{v}^{*i}, \mathbf{v}_{t-1}^i, \mathbf{v}_t^i)})(\bar{R}_t)^{\bar{\omega}(\mathbf{v}^*, \mathbf{v}_t) - \bar{\omega}(\mathbf{v}_0, \mathbf{v}_t)}}{\zeta(\mathcal{V}_1, \mathbf{v}^*, \mathbf{v}_t, t)(1 + \zeta(\mathcal{V}_1, \mathbf{v}^*, \mathbf{v}_t, t))}\right|$$

$$\sum_{\mathbf{v}_{t-1}^i \neq \mathbf{v}^{*i}, \mathbf{v}_t^i}|\sum_{\mathbf{v}_0 \in \mathcal{V}_1}| := \sum_{\mathbf{v}_{t-1}^i \neq \mathbf{v}^{*i}, \mathbf{v}_t^i} \left|\sum_{\mathbf{v}_0 \in \mathcal{V}_1} \frac{\tau(\mathbf{v}^{*i}, \mathbf{v}_{t-1}^i, \mathbf{v}_t^i) \cdot (1 - \frac{\tau(\mathbf{v}_0^i, \mathbf{v}_{t-1}^i, \mathbf{v}_t^i)}{\tau(\mathbf{v}^{*i}, \mathbf{v}_{t-1}^i, \mathbf{v}_t^i)})(\bar{R}_t)^{\bar{\omega}(\mathbf{v}^*, \mathbf{v}_t) - \bar{\omega}(\mathbf{v}_0, \mathbf{v}_t)}}{\zeta(\mathcal{V}_1, \mathbf{v}^*, \mathbf{v}_t, t)(1 + \zeta(\mathcal{V}_1, \mathbf{v}^*, \mathbf{v}_t, t))}\right|$$

$$\sum_{\mathbf{v}_{t-1}^i} |\sum_{\mathbf{v}_0 = \mathbf{v}_t^i} | := \sum_{\mathbf{v}_{t-1}^i = 1}^{k} \left| \sum_{\mathbf{v}_0 \mathbf{v}_t^i} \frac{\tau(\mathbf{v}^{*i}, \mathbf{v}_{t-1}^i, \mathbf{v}_t^i) \cdot (1 - \frac{\tau(\mathbf{v}_0^i, \mathbf{v}_{t-1}^i, \mathbf{v}_t^i)}{\tau(\mathbf{v}^{*i}, \mathbf{v}_{t-1}^i, \mathbf{v}_t^i)})(\bar{R}_t)^{\bar{\omega}(\mathbf{v}^*, \mathbf{v}_t) - \bar{\omega}(\mathbf{v}_0, \mathbf{v}_t)}}{\zeta(\mathcal{V}_1, \mathbf{v}^*, \mathbf{v}_t, t)(1 + \zeta(\mathcal{V}_1, \mathbf{v}^*, \mathbf{v}_t, t))} \right|$$

We have

$$\|q(\mathbf{v}_{t-1|0}^i | \mathbf{v}_{t|0} = \mathbf{v}_t) - q(\mathbf{v}_{t-1|1}^i | \mathbf{v}_{t|1} = \mathbf{v}_t)\| = \sum_{\mathbf{v}_{t-1}^i} |\sum_{\mathbf{v}_0 \in \mathcal{V}_1} | \tag{267}$$

$$\leq \sum_{\mathbf{v}_{t-1}^i = \mathbf{v}^{*i}} |\sum_{\mathbf{v}_0 \in \mathcal{V}_1} | + \sum_{\mathbf{v}_{t-1}^i \neq \mathbf{v}^{*i}, \mathbf{v}_t^i} |\sum_{\mathbf{v}_0 \neq \mathbf{v}_{t-1}^i} | + \sum_{\mathbf{v}_{t-1}^i \neq \mathbf{v}^{*i}, \mathbf{v}_t^i} |\sum_{\mathbf{v}_0 = \mathbf{v}_{t-1}^i} | + \sum_{\mathbf{v}_{t-1}^i = \mathbf{v}_t^i} |\sum_{\mathbf{v}_0 \in \mathcal{V}_1} | \tag{268}$$

$$= \sum_{\mathbf{v}_{t-1}^i = \mathbf{v}^{*i}} \sum_{\mathbf{v}_0 \in \mathcal{V}_1} + \sum_{\mathbf{v}_{t-1}^i \neq \mathbf{v}^{*i}, \mathbf{v}_t^i} \sum_{\mathbf{v}_0 \neq \mathbf{v}_{t-1}^i} - \sum_{\mathbf{v}_{t-1}^i \neq \mathbf{v}^{*i}, \mathbf{v}_t^i} \sum_{\mathbf{v}_0 = \mathbf{v}_{t-1}^i} - \sum_{\mathbf{v}_{t-1}^i = \mathbf{v}_t^i} \sum_{\mathbf{v}_0 \in \mathcal{V}_1} \tag{269}$$

$$= \sum_{\mathbf{v}_{t-1}^i = \mathbf{v}^{*i}} \sum_{\mathbf{v}_0 \in \mathcal{V}_1} + \sum_{\mathbf{v}_{t-1}^i \neq \mathbf{v}^{*i}, \mathbf{v}_t^i} \sum_{\mathbf{v}_0 \neq \mathbf{v}_{t-1}^i} + \sum_{\mathbf{v}_{t-1}^i \neq \mathbf{v}^{*i}, \mathbf{v}_t^i} \sum_{\mathbf{v}_0 = \mathbf{v}_{t-1}^i}$$

$$- (2 \sum_{\mathbf{v}_{t-1}^i \neq \mathbf{v}^{*i}, \mathbf{v}_t^i} \sum_{\mathbf{v}_0 = \mathbf{v}_{t-1}^i} + \sum_{\mathbf{v}_{t-1}^i = \mathbf{v}_t^i} \sum_{\mathbf{v}_0 \in \mathcal{V}_1}) \tag{270}$$

$$= \sum_{\mathbf{v}_{t-1}^i \neq \mathbf{v}_t^i} \sum_{\mathbf{v}_0 \in \mathcal{V}_1} -2 \sum_{\mathbf{v}_{t-1}^i \neq \mathbf{v}^{*i}, \mathbf{v}_t^i} \sum_{\mathbf{v}_0 = \mathbf{v}_{t-1}^i} - \sum_{\mathbf{v}_{t-1}^i = \mathbf{v}_t^i} \sum_{\mathbf{v}_0 \in \mathcal{V}_1} \tag{271}$$

$$= -2 \sum_{\mathbf{v}_{t-1}^i = \mathbf{v}^{*i}} \sum_{\mathbf{v}_0 \in \mathcal{V}_1} -2 \sum_{\mathbf{v}_{t-1}^i = \mathbf{v}_t^i} \sum_{\mathbf{v}_0 = \mathbf{v}_t^i} \tag{272}$$

Therefore, from above we have

$$\frac{1}{2}\|q(\mathbf{v}_{t-1|0}^i | \mathbf{v}_{t|0} = \mathbf{v}_t) - q(\mathbf{v}_{t-1|1}^i | \mathbf{v}_{t|1} = \mathbf{v}_t)\|_{l_1} \tag{273}$$

$$\leq -\tau(\mathbf{v}^{*i}, \mathbf{v}_t^i, \mathbf{v}_t^i) \frac{\sum_{\mathbf{v}_0 \in \mathcal{V}_1}(1 - \frac{\tau(\mathbf{v}_0^i, \mathbf{v}_t^i, \mathbf{v}_t^i)}{\tau(\mathbf{v}^{*i}, \mathbf{v}_t^i, \mathbf{v}_t^i)})(\bar{R}_t)^{\bar{\omega}(\mathbf{v}^*, \mathbf{v}_t) - \bar{\omega}(\mathbf{v}_0, \mathbf{v}_t)}}{(\zeta(\mathcal{V}_1, \mathbf{v}^*, \mathbf{v}_t, t))(1 + \zeta(\mathcal{V}_1, \mathbf{v}^*, \mathbf{v}_t, t))}$$

$$- \sum_{\mathbf{v}_{t-1}^i \neq \mathbf{v}^{*i}, \mathbf{v}_t^i} \tau(\mathbf{v}^{*i}, \mathbf{v}_{t-1}^i, \mathbf{v}_t^i) \frac{\sum_{\mathbf{v}_0 \in \mathcal{V}_1, \mathbf{v}_0 = \mathbf{v}_{t-1}^i}(1 - \frac{\tau(\mathbf{v}_0^i, \mathbf{v}_{t-1}^i, \mathbf{v}_t^i)}{\tau(\mathbf{v}^{*i}, \mathbf{v}_{t-1}^i, \mathbf{v}_t^i)})(\bar{R}_t)^{\bar{\omega}(\mathbf{v}^*, \mathbf{v}_t) - \bar{\omega}(\mathbf{v}_0, \mathbf{v}_t)}}{(\zeta(\mathcal{V}_1, \mathbf{v}^*, \mathbf{v}_t, t))(1 + \zeta(\mathcal{V}_1, \mathbf{v}^*, \mathbf{v}_t, t))} \tag{274}$$

$$= \mu_t^+ \frac{\bar{\mu}_{t-1}^-}{\bar{\mu}_t^-}(\frac{\bar{R}_{t-1}}{\bar{R}_t} - 1) \frac{\sum_{\mathbf{v}_0 \in \mathcal{V}_1, \mathbf{v}_0^i = \mathbf{v}_t^i}(\bar{R}_t)^{\bar{\omega}(\mathbf{v}^*, \mathbf{v}_t) - \bar{\omega}(\mathbf{v}_0, \mathbf{v}_t)}}{(\zeta(\mathcal{V}_1, \mathbf{v}^*, \mathbf{v}_t, t))(1 + \zeta(\mathcal{V}_1, \mathbf{v}^*, \mathbf{v}_t, t))} \tag{275}$$

$$+ \frac{\mu_t^- \bar{\mu}_{t-1}^-}{\bar{\mu}_t^-}(\bar{R}_t - 1) \frac{\sum_{\mathbf{v}_0 \in \mathcal{V}_1, \mathbf{v}_0 \neq \mathbf{v}_t^i, \mathbf{v}^{*i}}(\bar{R}_t)^{\bar{\omega}(\mathbf{v}^*, \mathbf{v}_t) - \bar{\omega}(\mathbf{v}_0, \mathbf{v}_t)}}{(\zeta(\mathcal{V}_1, \mathbf{v}^*, \mathbf{v}_t, t))(1 + \zeta(\mathcal{V}_1, \mathbf{v}^*, \mathbf{v}_t, t))} \tag{276}$$

$$\overset{(i)}{\leq} \mu_t^+ \cdot \frac{\bar{\mu}_{t-1}^-}{\bar{\mu}_t^-} \cdot (\frac{\bar{R}_{t-1}}{\bar{R}_t} - 1) \cdot \frac{1}{1 + \zeta(\mathcal{V}_1, \mathbf{v}^*, \mathbf{v}_t, t)} \tag{277}$$

$(i)$ is from the fact that

$$\mu_t^+ \cdot \frac{\bar{\mu}_{t-1}^-}{\bar{\mu}_t^-} \cdot (\frac{\bar{R}_{t-1}}{\bar{R}_t} - 1) \Big/ \frac{\mu_t^- \cdot \bar{\mu}_{t-1}^-}{\bar{\mu}_t^-} \cdot (\bar{R}_t - 1) \tag{278}$$

$$= \frac{\mu_t^+}{\mu_t^-}(\frac{\bar{R}_{t-1}}{\bar{R}_t} - 1) \frac{1}{\bar{R}_t - 1} = \frac{1 + (k-1)\alpha_t}{1 + (k-1)\bar{\alpha}_t} \frac{1 - \bar{\alpha}_t}{1 - \bar{\alpha}_{t-1}} \frac{1}{\alpha_t} > 1 \tag{279}$$

$\square$

**Lemma K.7.** Given $\eta \in [n]$, when $ep \leq \frac{4}{5}$, we have the following results

$$\eta + 1 = \operatorname*{argmax}_{h;h \in \{\eta+1,...,n\}} \left(\frac{eh}{\eta}\right)^{\eta} q^{h-\eta} \text{ and } \left(\frac{eh}{\eta}\right)^{\eta} q^{h-\eta} \text{ is monotonically decreasing,} \tag{280}$$

$$2\eta = \operatorname*{argmax}_{h;h \in \{2\eta,2\eta+1,...,n\}} \left(\frac{eqh}{h-\eta}\right)^{h-\eta} \text{ and } \left(\frac{eqh}{h-\eta}\right)^{h-\eta} \text{ is monotonically decreasing.} \tag{281}$$

In the second equation, we assume $\eta \leq \lfloor \frac{n}{2} \rfloor$.

*Proof of Lemma K.7.* We first show $\eta + 1 = \operatorname{argmax}_{h;h \in \{\eta+1,...,n\}} \left(\frac{eh}{\eta}\right)^{\eta} q^{h-\eta}$:

Let $x = \eta + h$, and $f(x) = ((1 + \frac{x}{\eta})e)^x$. Easy to show that $f(x) = \left(\frac{eh}{\eta}\right)^{\eta} q^{h-\eta}$. Now consider $f(x)/f(x+1)$:

$$\frac{f(x)}{f(x+1)} = \frac{((1+\frac{x}{\eta})e)^{\eta} q^x}{((1+\frac{x+1}{\eta})e)^{\eta} q^{x+1}} = \left(\frac{\eta+x}{(\eta+x+1)q^{\frac{1}{\eta}}}\right)^{\eta} \tag{282}$$

Now compare $(\eta + x)$ and $(\eta + x + 1)q^{\frac{1}{\eta}}$, since

$$\left(\frac{\eta+x}{\eta+x+1}\right)^{\eta} \geq \left(\frac{\eta}{\eta+1}\right)^{\eta} \geq \frac{1}{e} > q \tag{283}$$

Thus, when $eq \leq \frac{4}{5}$, $f(x)$ is monotonically decreasing. Therefore,

$$1 = \operatorname*{argmax}_{x;x \in \{1,2,...,n-\eta\}} f(x) \Leftrightarrow \eta + 1 = \operatorname*{argmax}_{h;h \in \{\eta+1,...,n\}} \left(\frac{eh}{\eta}\right)^{\eta} q^{k-\eta} \tag{284}$$

We now show $2\eta = \operatorname{argmax}_{h;h \in \{2\eta,2\eta+1,...,n\}} \left(\frac{eqh}{h-\eta}\right)^{h-\eta}$:

Let $x = \eta + h, x \geq \eta, x \leq n - \eta$, and $f(x) = (eq(1 + \frac{\eta}{x}))^x$. Consider the monotonicity of $f(x)$.

$$f'(x) = (eq(1 + \frac{\eta}{x}))^x \cdot \left[\log(eq(1 + \frac{\eta}{x})) - \frac{\eta}{x+\eta}\right] \tag{285}$$

Further consider $\log(eq(1 + \frac{\eta}{x})) - \frac{\eta}{x+\eta}$. Let $\xi = \frac{x}{\eta}, \xi \geq 1, \xi \leq \frac{n}{\eta} - 1$, $\varphi(\xi) = \log(eq(1 + \frac{1}{\xi})) - \frac{1}{1+\xi}$.

$$\varphi(\xi) = -\frac{\xi}{(1+\xi)^2} < 0 \tag{286}$$

Hence, $\varphi$ is monotonically decreasing. When $eq \leq \frac{4}{5}$, $\varphi(1) = \log(2) + \log(eq) - \frac{1}{2} < 0$. Thus, $f'(x) < 0$ for $x \in \{\eta, \eta+1, ..., n-\eta\}$. $f(x)$ therefore is monotonically decreasing.

$$\eta = \operatorname*{argmax}_{x;x \in \{\eta,\eta+1,...,n-\eta\}} f(x) \Leftrightarrow 2\eta = \operatorname*{argmax}_{h;h \in \{2\eta,2\eta+1,...,n\}} \left(\frac{eqh}{h-\eta}\right)^{h-\eta} \tag{287}$$

$\square$

**Corollary K.1.** Given $\eta \in [n]$, when $ep \leq \frac{4}{5}$. Define

$$f(h) = \min\left\{\left(\frac{eh}{\eta}\right)^{\eta} q^{h-\eta}, \left(\frac{eqh}{h-\eta}\right)^{h-\eta}\right\} \tag{288}$$

We have

$$f(h) = \begin{cases} \text{When } h \in \{\eta+1, \eta+2, ..., \min\{2\eta-1, n\}\}, f(h) = \left(\frac{eh}{\eta}\right)^{\eta} q^{h-\eta} \\ \\ \text{When } \eta \leq \lfloor \frac{n}{2} \rfloor \text{ and } h \in \{2\eta, ..., n\}, f(h) = \left(\frac{eqh}{h-\eta}\right)^{h-\eta} \end{cases} \tag{289}$$

**Lemma K.8** (Optimality of $p_\phi(\mathbf{v}_0^i|\mathbf{v}_t)$). Let $q(\cdot)$ and $p_\phi$ denote the probability measure in the forward diffusion and backward denoising process respectively. In the training procedure, we are actually solving the following optimization problem:

$$\text{minimize } \mathcal{D}_{\mathrm{KL}}\bigg(q(\mathbf{v}_0)\|\prod_{i=1}^{n}\sum_{\mathbf{v}_t}p_\phi(\mathbf{v}_0^i|\mathbf{v}_t)\cdot q(\mathbf{v}_t)\bigg) \tag{290}$$

and the optimal solution is obtained when $p_\phi(\mathbf{v}_0^i|\mathbf{v}_t) = q(\mathbf{v}_0^i|\mathbf{v}_t)$.

*Proof of Lemma K.8.* Reformulate the KL divergence as follows:

$$\text{minimize } \mathcal{D}_{\mathrm{KL}}\bigg(q(\mathbf{v}_0)\|\prod_{i=1}^{n}\sum_{\mathbf{v}_t}p_\phi(\mathbf{v}_0^i|\mathbf{v}_t)\cdot q(\mathbf{v}_t)\bigg) \tag{291}$$

$$\Leftrightarrow\text{maximize } \sum_{\mathbf{v}_0}q(\mathbf{v}_0)\log\prod_{i=1}^{n}\sum_{\mathbf{v}_t}p_\phi(\mathbf{v}_0^i|\mathbf{v}_t)\cdot q(\mathbf{v}_t) \tag{292}$$

Let $p_i(\mathbf{v}_0^i) := \sum_{\mathbf{v}_t}p_\phi(\mathbf{v}_0^i|\mathbf{v}_t)\cdot q(\mathbf{v}_t)$. The above objective is $\sum_{\mathbf{v}_0}q(\mathbf{v}_0)\log p_i(\mathbf{v}_0^i)$. Further define $q_i(\mathbf{v}_0^i) := \sum_{\mathbf{v}_t}q(\mathbf{v}_0^i|\mathbf{v}_t)\cdot q(\mathbf{v}_t)$.

$$\sum_{\mathbf{v}_0}q(\mathbf{v}_0)\log\prod_{i=1}^{n}\sum_{\mathbf{v}_t}q(\mathbf{v}_0^i|\mathbf{v}_t)\cdot q(\mathbf{v}_t) - \sum_{\mathbf{v}_0}q(\mathbf{v}_0)\log\prod_{i=1}^{n}\sum_{\mathbf{v}_t}p_\phi(\mathbf{v}_0^i|\mathbf{v}_t)\cdot q(\mathbf{v}_t) \tag{293}$$

$$= \sum_{\mathbf{v}_0}q(\mathbf{v}_0)\log\prod_{i=1}^{n}q_i(\mathbf{v}_0^i) - \sum_{\mathbf{v}_0}q(\mathbf{v}_0)\log\prod_{i=1}^{n}p_i(\mathbf{v}_0^i) \tag{294}$$

$$= \sum_{\mathbf{v}_0}q(\mathbf{v}_0^1,\mathbf{v}_0^2,...,\mathbf{v}_0^n)\sum_{i=1}^{n}\log\frac{q_i(\mathbf{v}_0^i)}{p_i(\mathbf{v}_0^i)} \tag{295}$$

$$= \sum_{i=1}^{n}\mathcal{D}_{\mathrm{KL}}(q_i(\mathbf{v}_0^i)\|p_i(\mathbf{v}_0^i)) \geq 0 \tag{296}$$

Therefore, the equality hold when $p_\phi(\mathbf{v}_0^i|\mathbf{v}_t) = q(\mathbf{v}_0^i|\mathbf{v}_t)$. $\qquad\square$

# L  Specific Discussion on $\tau(\mathbf{v}_0^i, \mathbf{v}_{t-1}^i, \mathbf{v}_t^i)$

Recall that

$$\tau(\mathbf{v}_0^i, \mathbf{v}_{t-1}^i, \mathbf{v}_t^i) = \frac{\left[(\mu_t^+)^{1-\mathbb{1}_{\mathbf{v}_t^i\neq\mathbf{v}_{t-1}^i}}(\mu_t^-)^{\mathbb{1}_{\mathbf{v}_t^i\neq\mathbf{v}_{t-1}^i}}\right]\cdot\left[(\bar\mu_{t-1}^+)^{1-\mathbb{1}_{\mathbf{v}_{t-1}^i\neq\mathbf{v}_0^i}}(\bar\mu_{t-1}^-)^{\mathbb{1}_{\mathbf{v}_{t-1}^i\neq\mathbf{v}_0^i}}\right]}{(\bar\mu_t^+)^{1-\mathbb{1}_{\mathbf{v}_t^i\neq\mathbf{v}_0^i}}(\bar\mu_t^-)^{\mathbb{1}_{\mathbf{v}_t^i\neq\mathbf{v}_0^i}}} \tag{297}$$

There are in total five possible conditions:

- If $\mathbf{v}_{t-1}^i = \mathbf{v}_t^i$ and $\mathbf{v}_0^i = \mathbf{v}_t^i$, $\tau(\mathbf{v}_0^i, \mathbf{v}_{t-1}^i, \mathbf{v}_t^i) = \frac{\mu_t^+\bar\mu_{t-1}^+}{\bar\mu_t^+}$.

- If $\mathbf{v}_{t-1}^i = \mathbf{v}_t^i$ and $\mathbf{v}_0^i \neq \mathbf{v}_t^i$, $\tau(\mathbf{v}_0^i, \mathbf{v}_{t-1}^i, \mathbf{v}_t^i) = \frac{\mu_t^+\bar\mu_{t-1}^-}{\bar\mu_t^-}$.

- If $\mathbf{v}_{t-1}^i \neq \mathbf{v}_t^i$ and $\mathbf{v}_0^i = \mathbf{v}_t^i$, $\tau(\mathbf{v}_0^i, \mathbf{v}_{t-1}^i, \mathbf{v}_t^i) = \frac{\mu_t^-\bar\mu_{t-1}^-}{\bar\mu_t^+}$.

- If $\mathbf{v}_{t-1}^i \neq \mathbf{v}_t^i$ and $\mathbf{v}_0^i = \mathbf{v}_{t-1}^i$, $\tau(\mathbf{v}_0^i, \mathbf{v}_{t-1}^i, \mathbf{v}_t^i) = \frac{\mu_t^-\bar\mu_{t-1}^+}{\bar\mu_t^-}$.

- If $\mathbf{v}_{t-1}^i \neq \mathbf{v}_t^i$ and $\mathbf{v}_0^i \neq \mathbf{v}_t^i$ and $\mathbf{v}_0^i \neq \mathbf{v}_{t-1}^i$, $\tau(\mathbf{v}_0^i, \mathbf{v}_{t-1}^i, \mathbf{v}_t^i) = \frac{\mu_t^-\bar\mu_{t-1}^-}{\bar\mu_t^-}$.

Since in the derivation, we encounter the ratio $\tau(\mathbf{v}_0^i, \mathbf{v}_{t-1}^i, \mathbf{v}_t^i)/\tau(\mathbf{v}^{*i}, \mathbf{v}_{t-1}^i, \mathbf{v}_t^i)$. Therefore, we discuss the ratio in detail here. Define $\kappa(\mathbf{v}^{*i}, \mathbf{v}_0^i, \mathbf{v}_{t-1}^i, \mathbf{v}_t^i) = \frac{\tau(\mathbf{v}_0^i, \mathbf{v}_{t-1}^i, \mathbf{v}_t^i)}{\tau(\mathbf{v}^{*i}, \mathbf{v}_{t-1}^i, \mathbf{v}_t^i)} = \frac{q(\mathbf{v}_{t-1|0}^i = \mathbf{v}_{t-1}^i | \mathbf{v}_{0|0}^i = v_0^i)}{q(\mathbf{v}_{t|0}^i = v_t^i | \mathbf{v}_{0|0}^i = v_0^i)} \frac{q(\mathbf{v}_{t|0}^i = v_t^i | \mathbf{v}_{0|0}^i = \mathbf{v}^{*i})}{q(\mathbf{v}_{t-1|0}^i = \mathbf{v}_{t-1}^i | \mathbf{v}_{0|0}^i = \mathbf{v}^{*i})}$. In the following, we will denote it simply as $\kappa$.

- When $\mathbf{v}_t^i = \mathbf{v}^{*i}$
  - If $\mathbf{v}_{t-1}^i = \mathbf{v}_t^i$ and $\mathbf{v}_0^i = \mathbf{v}_t^i$, $\kappa = 1$
  - If $\mathbf{v}_{t-1}^i = \mathbf{v}_t^i$ and $\mathbf{v}_0^i \neq \mathbf{v}_t^i$, $\kappa = \frac{\bar{\mu}_t^+ \bar{\mu}_{t-1}^-}{\bar{\mu}_t^- \bar{\mu}_{t-1}^+} = \bar{R}_t/\bar{R}_{t-1} < 1$
  - If $\mathbf{v}_{t-1}^i \neq \mathbf{v}_t^i$ and $\mathbf{v}_0^i = \mathbf{v}_t^i$, $\kappa = 1$
  - If $\mathbf{v}_{t-1}^i \neq \mathbf{v}_t^i$ and $\mathbf{v}_0^i \neq \mathbf{v}_t^i$ and $\mathbf{v}_0^i = \mathbf{v}_{t-1}^i$, $\kappa = \frac{\bar{\mu}_t^+ \bar{\mu}_{t-1}^+}{\bar{\mu}_t^- \bar{\mu}_{t-1}^-} = \bar{R}_t \bar{R}_{t-1} > 1$
  - If $\mathbf{v}_{t-1}^i \neq \mathbf{v}_t^i$ and $\mathbf{v}_0^i \neq \mathbf{v}_t^i$ and $\mathbf{v}_0^i \neq \mathbf{v}_{t-1}^i$, $\kappa = \frac{\bar{\mu}_t^+}{\bar{\mu}_t^-} = \bar{R}_t > 1$

- When $\mathbf{v}_t^i \neq \mathbf{v}^{*i}$
  - If $\mathbf{v}_{t-1}^i = \mathbf{v}_t^i$ and $\mathbf{v}_0^i = \mathbf{v}_t^i$, $\kappa = \frac{\bar{\mu}_{t-1}^+ \bar{\mu}_t^-}{\bar{\mu}_{t-1}^- \bar{\mu}_t^+} = \bar{R}_{t-1}/\bar{R}_t > 1$
  - If $\mathbf{v}_{t-1}^i = \mathbf{v}_t^i$ and $\mathbf{v}_0^i \neq \mathbf{v}_t^i$, $\kappa = 1$
  - If $(\mathbf{v}_{t-1}^i \neq \mathbf{v}_t^i$ and $\mathbf{v}_{t-1}^i = \mathbf{v}^{*i})$ and $\mathbf{v}_0^i = \mathbf{v}_t^i$, $\kappa = \frac{\bar{\mu}_t^- \bar{\mu}_{t-1}^-}{\bar{\mu}_t^+ \bar{\mu}_{t-1}^+} = 1/(\bar{R}_{t-1} \bar{R}_t) < 1$
  - If $(\mathbf{v}_{t-1}^i \neq \mathbf{v}_t^i$ and $\mathbf{v}_{t-1}^i \neq \mathbf{v}^{*i})$ and $\mathbf{v}_0^i = \mathbf{v}_t^i$, $\kappa = \frac{\bar{\mu}_t^-}{\bar{\mu}_t^+} = 1/\bar{R}_t < 1$
  - If $(\mathbf{v}_{t-1}^i \neq \mathbf{v}_t^i$ and $\mathbf{v}_{t-1}^i = \mathbf{v}^{*i})$ and $\mathbf{v}_0^i \neq \mathbf{v}_t^i$ and $\mathbf{v}_0^i = \mathbf{v}_{t-1}^i$, $\kappa = 1$
  - If $(\mathbf{v}_{t-1}^i \neq \mathbf{v}_t^i$ and $\mathbf{v}_{t-1}^i \neq \mathbf{v}^{*i})$ and $\mathbf{v}_0^i \neq \mathbf{v}_t^i$ and $\mathbf{v}_0^i = \mathbf{v}_{t-1}^i$, $\kappa = \frac{\bar{\mu}_{t-1}^+}{\bar{\mu}_{t-1}^-} = \bar{R}_{t-1} > 1$
  - If $(\mathbf{v}_{t-1}^i \neq \mathbf{v}_t^i$ and $\mathbf{v}_{t-1}^i = \mathbf{v}^{*i})$ and $\mathbf{v}_0^i \neq \mathbf{v}_t^i$ and $\mathbf{v}_0^i \neq \mathbf{v}_{t-1}^i$, $\kappa = \frac{\bar{\mu}_{t-1}^-}{\bar{\mu}_{t-1}^+} = 1/\bar{R}_{t-1} < 1$
  - If $(\mathbf{v}_{t-1}^i \neq \mathbf{v}_t^i$ and $\mathbf{v}_{t-1}^i \neq \mathbf{v}^{*i})$ and $\mathbf{v}_0^i \neq \mathbf{v}_t^i$ and $\mathbf{v}_0^i \neq \mathbf{v}_{t-1}^i$, $\kappa = 1$

Further, we define $\bar{\kappa}(\mathbf{v}^{*i}, \mathbf{v}_0^i, \mathbf{v}_{t-1}^i, \mathbf{v}_t^i) = \frac{\tau(\mathbf{v}_0^i, \mathbf{v}_{t-1}^i, \mathbf{v}_t^i)}{\tau(\mathbf{v}^{*i}, \mathbf{v}_{t-1}^i, \mathbf{v}_t^i)} \cdot \left(\frac{\bar{\mu}_t^+}{\bar{\mu}_t^-}\right)^{1_{v^* \neq \mathbf{v}_t^i} - 1_{\mathbf{v}_0^i \neq \mathbf{v}_t^i}}$.

- When $\mathbf{v}_t^i = \mathbf{v}^{*i}$
  - If $\mathbf{v}_{t-1}^i = \mathbf{v}_t^i$ and $\mathbf{v}_0^i = \mathbf{v}_t^i$, $\bar{\kappa} = 1$
  - If $\mathbf{v}_{t-1}^i = \mathbf{v}_t^i$ and $\mathbf{v}_0^i \neq \mathbf{v}_t^i$, $\bar{\kappa} = \frac{\bar{\mu}_{t-1}^-}{\bar{\mu}_{t-1}^+} = 1/\bar{R}_{t-1} < 1$
  - If $\mathbf{v}_{t-1}^i \neq \mathbf{v}_t^i$ and $\mathbf{v}_0^i = \mathbf{v}_t^i$, $\bar{\kappa} = 1$
  - If $\mathbf{v}_{t-1}^i \neq \mathbf{v}_t^i$ and $\mathbf{v}_0^i = \mathbf{v}_{t-1}^i$, $\bar{\kappa} = \frac{\bar{\mu}_{t-1}^+}{\bar{\mu}_{t-1}^-} = \bar{R}_{t-1} > 1$
  - If $\mathbf{v}_{t-1}^i \neq \mathbf{v}_t^i$ and $\mathbf{v}_0^i \neq \mathbf{v}_t^i$ and $\mathbf{v}_0^i \neq \mathbf{v}_{t-1}^i$, $\bar{\kappa} = 1$

- When $\mathbf{v}_t^i \neq \mathbf{v}^{*i}$
  - If $\mathbf{v}_{t-1}^i = \mathbf{v}_t^i$ and $\mathbf{v}_0^i = \mathbf{v}_t^i$, $\bar{\kappa} = \frac{\bar{\mu}_{t-1}^+}{\bar{\mu}_{t-1}^-} = \bar{R}_{t-1} > 1$
  - If $\mathbf{v}_{t-1}^i = \mathbf{v}_t^i$ and $\mathbf{v}_0^i \neq \mathbf{v}_t^i$, $\bar{\kappa} = 1$
  - If $(\mathbf{v}_{t-1}^i \neq \mathbf{v}_t^i$ and $\mathbf{v}_{t-1}^i = \mathbf{v}^{*i})$ and $\mathbf{v}_0^i = \mathbf{v}_t^i$, $\bar{\kappa} = \frac{\bar{\mu}_{t-1}^-}{\bar{\mu}_{t-1}^+} = 1/\bar{R}_{t-1} < 1$
  - If $(\mathbf{v}_{t-1}^i \neq \mathbf{v}_t^i$ and $\mathbf{v}_{t-1}^i \neq \mathbf{v}^{*i})$ and $\mathbf{v}_0^i = \mathbf{v}_t^i$, $\bar{\kappa} = 1$
  - If $(\mathbf{v}_{t-1}^i \neq \mathbf{v}_t^i$ and $\mathbf{v}_{t-1}^i = \mathbf{v}^{*i})$ and $\mathbf{v}_0^i \neq \mathbf{v}_t^i$ and $\mathbf{v}_0^i = \mathbf{v}_{t-1}^i$, $\bar{\kappa} = 1$

– If $(\mathbf{v}_{t-1}^i \neq \mathbf{v}_t^i$ and $\mathbf{v}_{t-1}^i \neq \mathbf{v}^{*i})$ and $\mathbf{v}_0^i \neq \mathbf{v}_t^i$ and $\mathbf{v}_0^i = \mathbf{v}_{t-1}^i$, $\bar{\kappa} = \frac{\bar{\mu}_{t-1}^+}{\bar{\mu}_{t-1}} = \bar{R}_{t-1} > 1$

– If $(\mathbf{v}_{t-1}^i \neq \mathbf{v}_t^i$ and $\mathbf{v}_{t-1}^i = \mathbf{v}^{*i})$ and $\mathbf{v}_0^i \neq \mathbf{v}_t^i$ and $\mathbf{v}_0^i \neq \mathbf{v}_{t-1}^i$, $\bar{\kappa} = \frac{\bar{\mu}_{t-1}^-}{\bar{\mu}_{t-1}^+} = 1/\bar{R}_{t-1} < 1$

– If $(\mathbf{v}_{t-1}^i \neq \mathbf{v}_t^i$ and $\mathbf{v}_{t-1}^i \neq \mathbf{v}^{*i})$ and $\mathbf{v}_0^i \neq \mathbf{v}_t^i$ and $\mathbf{v}_0^i \neq \mathbf{v}_{t-1}^i$, $\bar{\kappa} = 1$

