# OpenReview forum: "On the Inherent Privacy Properties of Discrete Denoising Diffusion Models"
_TMLR — Accepted by TMLR_

### Review · Reviewer_UWmy · 2024-02-19

**Summary Of Contributions:**

This paper studies the inherent privacy guarantee of discrete diffusion model (DDM), providing an upper bound on the privacy parameter $\delta$, namely, failure property, under the definition of per-instance differential privacy (p-DP). Through empirical verification on synthetic and real-world datasets, the research underscores the role of diffusion models in mitigating privacy risks associated with data sharing. It leads to the conclusion that privacy-preserving techniques should be applied along with the training of diffusion model.

**Audience:**

Yes

**Broader Impact Concerns:**

I do not think it is necessary to add more discussion on ethical or societal impacts.

**Claims And Evidence:**

Yes

**Requested Changes:**

According the discussion above, the paper might benefit from repositioning Theorem E.1 and relevant discussion to the main body of paper.

**Strengths And Weaknesses:**

Strength:
1.	The paper focuses on a timely question: the relationship between privacy and diffusion model.
2.	The experiments are carefully designed and they provide strong evidence that the theoretical results well align with empirical findings.
3.	The paper's emphasis on the nuanced relationship between data size, diffusion coefficients, and privacy leakage bounds contributes to a better understanding of the nature of this problem.

Weakness:
I am not an expert in diffusion model but it seems to me that the theoretical results within the context of p-DP are equally important as that of within the context of p-DP. It might be of better readers’ interest to reveal the relationship between privacy and diffusion model under the original notion of DP, which is more widely acknowledged.

---

> ### Author Response · Authors · 2024-03-19
> **Response to Reviewer UWmy**
>
> We extend our gratitude to Reviewer UWmy for acknowledging the value of our work.
>
> >It might be of better readers’ interest to reveal the relationship between privacy and diffusion model under the original notion of DP, which is more widely acknowledged. The paper might benefit from repositioning Theorem E.1 and relevant discussion to the main body of paper.
>
> Thank you for the valuable suggestion. We have incorporated a new paragraph (highlighted in red textbox on Page 8) in Section 3.1, summarizing our results on the DP guarantees of DDMs. To maintain the flow of the main text and minimize reader disruption, we have detailed the conditions for quantities mentioned in the theorem in Appendix E.

---

### Review · Reviewer_hAZk · 2024-02-19

**Summary Of Contributions:**

This paper studies the per-instance DP (pDP) property of discrete denoising diffusion models.

**Audience:**

Yes

**Broader Impact Concerns:**

Nothing within my scope.

**Claims And Evidence:**

Yes

**Requested Changes:**

1. I suggest adding a highlight to discuss why Theorem 1 is not true for continuous diffusion models. Even though $k$ appeared in equation (8) of Theorem 1, it's not quite clear (at least to me) what if $k\to\infty$, which corresponds to the continuous diffusion model (more general and has more application scenarios).

2. In the abstract, $s$ needs to be defined before using. At the bottom of page 2, $m$ needs to be defined before using.

3. I suggest moving the algorithm to remove sensitive data to the main text. I think the algorithm is very important because it can increase both privacy and accuracy if removing ~2% according to the experiment. I'm also curious about the time complexity of the algorithm. Can the authors provide any analysis?

**Strengths And Weaknesses:**

In general, I think this is a good paper.

Strength:

1. Diffusion model is a very important model in computer vision and many other tasks. Any privacy guarantee on those models has great potential to help the applications of diffusion models.

2. The theoretical analysis looks promising to me, which analyzes the per-instance DP property of DDM

3. The presentation of the main contribution is great! I like Figure 2 and Figure 3, great job!

Weakness:

1. The notations need to be defined before use, especially the abstract and introduction

2. Minor things on presentation (see the requested changes below)

---

> ### Author Response · Authors · 2024-03-19
> **Response to Reviewer hAZk**
>
> We greatly thank Reviewer hAZk for appreciating our theory contributions. Here, we will address questions raised by Reviewer hAZk.
>
> >The notations need to be defined before use, especially the abstract and introduction (Requested Changes: In the abstract, s needs to be defined before using. At the bottom of page 2, m needs to be defined before using.)
>
> Thank you for pointing out the notation issue. In the revised version, we have ensured that all terms are clearly defined at their initial introduction to enhance readability. We have highlighted the definition of m and s in the revised version.
>
> >I suggest adding a highlight to discuss why Theorem 1 is not true for continuous diffusion models.
>
> Thank you for the insightful suggestion. Regarding the extension to discrete-time / continuous-time continuous feature space diffusion models, our approach shifts from discrete (finite set) to continuous (uncountable set) feature spaces, such as those in image datasets. Our current analysis relies on some of the properties of discrete distribution. We think the high-level logic of our proof idea can be extended to continuous diffusion models, while the current statement in Theorem 1 cannot be directly extended due to the leverage of properties of discrete distributions. Potential changes need to be made when analyzing continuous data: as outlined in the Proof Sketch of Theorem 1 (Appendix A.2), we leverage the discrete distribution for radius selection and measure partition to balance the privacy leakage in each subset, for continuous features, the radius selection will depend on the integration of expected privacy leakage across partitioned subsets in continuous space.
>
> In our uploaded revised manuscript, we add Section 6 Limitation and Future Work where we include the discussion of the potential extension of our current analysis.
>
> >I suggest moving the algorithm to remove sensitive data to the main text. I think the algorithm is very important because it can increase both privacy and accuracy if removing ~2% according to the experiment. I'm also curious about the time complexity of the algorithm. Can the authors provide any analysis?
>
> Thank you for the valuable suggestion. We have relocated our pDP algorithm to Section 3.4 of the main text (in the revised manuscript). Regarding time complexity, our pDP algorithm has a total time complexity of $O(s \times T \times n \log n)$, where $s$ represents the number of samples for which we aim to assess privacy, $T$ denotes the total diffusion steps, and $n$ is the feature dimension. The most time-intensive part of our algorithm involves selecting the radius (lines 9), where determining the radius coefficient for each sample at each diffusion step for a given $\eta$ requires $O(\log n)$ time through binary search. Further, we add this time complexity analysis of our algorithm in Appendix C.

---

### Review · Reviewer_eTtK · 2024-03-13

**Summary Of Contributions:**

The paper introduces the first theoretical exploration into how discrete diffusion models (DDMs), used for creating discrete datasets, preserve privacy. Specifically, it uses the concept of per-instance differential privacy (pDP) to assess potential privacy risks for individual data points within a training dataset. This involves understanding how the privacy loss of each data point is influenced by the overall distribution of the dataset. It finds that privacy risks increase as the model transitions from generating pure noise to producing synthetic data. Additionally, it notes that a quicker decrease in diffusion coefficients can strengthen privacy protections. The authors validate their theoretical insights by applying them to both synthetic and real-world datasets, thereby confirming their practical applicability.

**Audience:**

Yes

**Broader Impact Concerns:**

Nothing to report.

**Claims And Evidence:**

Yes

**Requested Changes:**

Could you please assess whether the ''Onion Effect' occurs in your case? For evaluation, I believe that employing membership inference attacks, as you did in Section 4.2.2, could be an effective approach.

**Strengths And Weaknesses:**

**Strengths**

1. The authors have provided very interesting theoretical results based on the pDP concept.

2. The authors have empirically verified their theoretical findings on both synthetic and real-world datasets

3. The paper is well-written and presented.

**Weaknesses**

1. From the paper: "We evaluate the data-dependent part by removing the most sensitive data points (according to our data-dependent privacy parameters) from the dataset to train a DDM, and then evaluating the ML models trained based on the synthetic dataset generated by the DDM. Interestingly, we observe that the ML models obtained after a part of data removal can even outperform others without such data removal. We attribute this to the fact that the removed data points are likely outliers which may be actually not good for ML models to learn from. This illustrates another potentially valuable usage of our data-dependent analysis." : I believe simply removing data points identified as outliers may not suffice. Intriguingly, the authors in [1] have introduced a phenomenon they term the 'Onion Effect' This effect emerges when, upon the deletion of an initial group of outliers, a new set of outliers becomes apparent. Additionally, your analysis implies—and I concur—that increasing the training dataset size tends to mitigate risk. Therefore, removing data points and retraining the DDM on this now reduced dataset could potentially intensify the issue of privacy leakage.

[1] Carlini, N., Jagielski, M., Zhang, C., Papernot, N., Terzis, A., & Tramer, F. (2022). The privacy onion effect: Memorization is relative. Advances in Neural Information Processing Systems, 35, 13263-13276.

---

> ### Author Response · Authors · 2024-03-19
> **Response to Reviewer eTtK**
>
> We are deeply grateful to Reviewer eTtK for the insightful feedback and appreciating our contributions to the theory. Here, we are to respond to the questions proposed by Reviewer eTtK.
>
> >Could you please assess whether the ''Onion Effect' occurs in your case?
>
> Thank you for the thought-provoking question. We agree with the reviewer that the mere exclusion of the most sensitive data points from the dataset does not guarantee a reduction in privacy risks, as certain inliers may become outliers post-removal, a phenomenon we recognize as the “Onion Effect.” In our empirical analysis, we did not observe the "Onion Effect" in the Adult dataset, but we indeed noted it in both the German Credit and Loan datasets at varying data removal ratios. Specifically, for the German Credit dataset, eliminating up to 50% of sensitive data and doing recalculation cease to reduce privacy leakage as indicated by our pDP; for the Loan dataset, this threshold is around 8%. This can be attributed to the fact that an individual data point's privacy risk is determined by its similarity to the current dataset distribution. Simply removing certain data points does not necessarily enhance this similarity, and thus privacy leakage may not decrease (illustrated in Figure 6, Second Row).
>
> Nonetheless, our goal is not to assert that data removal, guided by our pDP bounds, will invariably strengthen privacy safeguards. Rather, our intent is to illustrate that our pDP bound can detect sensitive points, enables data curators to better evaluate the privacy risk of individual data points within training sets and to understand the relationship between data similarity and privacy exposure. In practice, data curators should perform pDP recalculations on modified datasets with varying data removal ratios to potentially achieve better privacy-utility trade-offs.
>
> To address potential misunderstandings and provide clear guidance, we incorporate additional explanations at the conclusion of Section 4.2.1 in our uploaded revised manuscript, delineating this concept more explicitly.

---

### Author Response · Authors · 2024-03-19
**Response Summary to Reviews**

We are grateful for the valuable feedback from all reviewers. It is recognized that our paper introduces the first theoretical analysis of diffusion models' inherent privacy. Here we summarize some key questions from reviewers: (1) whether the 'Onion Effect' exists in our data removal experiments and the computational time complexity of our algorithm; (2) the challenge in extending our theory to continuous diffusion models. We are dedicated to thoroughly addressing these questions to each reviewer and upload a revised manuscript according to the suggestions from reviewers (We have highlighted inline changes and indicated paragraph modifications with red text boxes.).

---

### Decision · Action_Editor_7Suc · 2024-05-01

**Recommendation:** Accept with minor revision

**Comment:**

There are couple of minor things I think need to be fixed before the final version:
- Fig 1 "Diffuion"
- "evaluate how the potential privacy leakage if one uses DDMs to generate synthetic datasets.", is the "how" there a typo, or is something else missing from the sentence?
- Is the $\overline{\lambda}$ used in definition of $d^{(t)}(v)$ defined somewhere?
- Maybe you could add the linear lines for Figures 5 LEFT and MIDDLE, to illustrate how close the bounds are to the $1/s$ and $1/s^2$.

**Audience:**

As the privacy of generative models is an interesting topic with great practical importance, I believe the paper is interesting for the broad audience of TMLR. This assessment was also shared by all the reviewers.

**Claims And Evidence:**

This paper studies the privacy properties of a deep learning based generative model. Authors show theoretically, that a particular type generative model, a discrete diffusion model, provides only weak privacy guarantees for the generated data. This is interesting result extends the earlier works that have show how GANs also satisfy only a weak privacy guarantee. The result further highlights the need for formal privacy preserving algorithms to be used when training a large deep learning based generative model.

The main claim of the paper is the privacy leakage, characterized in Theorem 1. The theorem presents a rather complicated bound, which author later simplify to a function of data size. This simplification is validated by empirical study. Furthermore, the privacy leakage of the algorithm is empirically studied with membership inference attacks (MIA). The MIA results provide further evidence to the implications of Theorem 1. All the reviewers, as well as I, found both the theoretical and empirical evidence convincing.